# Doubly Robust Bayesian Inference for Non-Stationary Streaming Data with $\beta$-Divergences

**Jeremias Knoblauch**
The Alan Turing Institute
Department of Statistics
University of Warwick
Coventry, CV4 7AL
j.knoblauch@warwick.ac.uk

**Jack Jewson**
Department of Statistics
University of Warwick
Coventry, CV4 7AL
j.e.jewson@warwick.ac.uk

**Theodoros Damoulas**
The Alan Turing Institute
Department of Computer Science & Department of Statistics
University of Warwick
Coventry, CV4 7AL
t.damoulas@warwick.ac.uk

## Abstract

We present the first robust Bayesian Online Changepoint Detection algorithm through General Bayesian Inference (GBI) with $\beta$-divergences. The resulting inference procedure is doubly robust for both the parameter and the changepoint (CP) posterior, with linear time and constant space complexity. We provide a construction for exponential models and demonstrate it on the Bayesian Linear Regression model. In so doing, we make two additional contributions: Firstly, we make GBI scalable using Structural Variational approximations that are exact as $\beta \to 0$. Secondly, we give a principled way of choosing the divergence parameter $\beta$ by minimizing expected predictive loss on-line. Reducing False Discovery Rates of CPs from over 90% to 0% on real world data, this offers the state of the art.

## 1  Introduction

Modeling non-stationary time series with changepoints (CPs) is popular [23, 50, 33] and important in a wide variety of research fields, including genetics [8, 16, 42], finance [27], oceanography [24], brain imaging and cognition [13, 20], cybersecurity [37] and robotics [2, 26]. For streaming data, a particularly important subclass are Bayesian On-line Changepoint Detection (BOCPD) methods that can process data sequentially [1, 11, 43, 47, 46, 41, 8, 34, 44, 40, 25] while providing full probabilistic uncertainty quantification. These algorithms declare CPs if the posterior predictive computed from $\boldsymbol{y}_{1:t}$ at time $t$ has low density for the value of the observation $\boldsymbol{y}_{t+1}$ at time $t+1$. Naturally, this leads to a high false CP discovery rate in the presence of outliers and as they run on-line, pre-processing is not an option. In this work, we provide the first robust on-line CP detection method that is applicable to multivariate data, works with a class of scalable models and quantifies model, CP and parameter uncertainty in a principled Bayesian fashion.

Standard Bayesian inference minimizes the Kullback-Leibler divergence (KLD) between the fitted model and the Data Generating Mechanism (DGM), but is not robust under outliers or model mis-specification due to its strictly increasing influence function. We remedy this by instead minimizing the $\beta$-divergence ($\beta$-D) whose influence function has a unique maximum, allowing us to deal with outliers effectively. Fig. 1 **A** illustrates this: Under the $\beta$-D, the influence of observations first

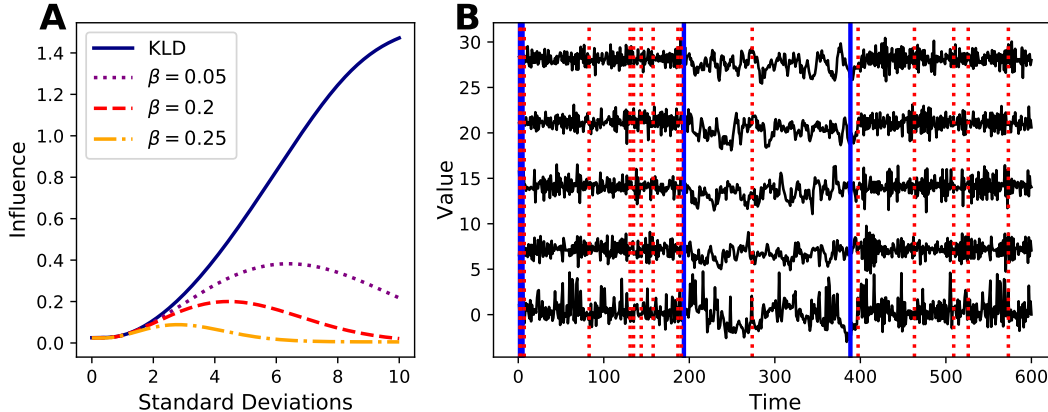

Figure 1: **A:** Influence of $\boldsymbol{y}_t$ on inference as function of distance to the posterior expectation in Standard Deviations for $\beta$-divergences with different $\beta$s. **B:** Five jointly modeled Simulated Autoregressions (ARs) with true CPs at $t = 200, 400$; bottom-most AR injected with $t_4$-noise. Maximum A Posteriori CPs of robust (standard) BOCPD shown as solid (dashed) vertical lines.

increases as they move away from the posterior mean, mimicking the KLD. However, once they move far enough, their influence decreases again. This can be interpreted to mean that they are (increasingly) treated as outliers. As $\beta$ increases, observations are registered as outliers closer to the posterior mean. Conversely, as $\beta \to 0$, one recovers the KLD which cannot treat any observation as an outlier. In addressing misspecification and outliers this way, our approach builds on the principles of General Bayesian Inference (GBI) [see 6, 21] and robust divergences [e.g. 4, 15]. This paper presents three contributions in separate domains that are also illustrated in Figs. 1 and 3:

(1) **Robust BOCPD**: We construct the very first robust BOCPD inference. The procedure is applicable to a wide class of (multivariate) models and is demonstrated on Bayesian Linear Regression (BLR). Unlike standard BOCPD, it discerns outliers and CPs, see Fig. 1 **B**.

(2) **Scalable GBI:** Due to intractable posteriors, GBI has received little attention in machine learning so far. We remedy this with a Structural Variational approximation which preserves parameter dependence and is exact as $\beta \to 0$, providing a near-perfect fit, see Fig. 3.

(3) **Choosing $\beta$:** While Fig. 1 **A** shows that $\beta$ regulates the degree of robustness [see also 21, 15], it is unclear how to set its magnitude. For the first time, we provide a principled way of initializing $\beta$. Further, we show how to refine it on-line by minimizing predictive losses.

The remainder of the paper is structured as follows: In Section 2, we summarize standard BOCPD and show how to extend it to robust inference using the $\beta$-D. We quantify the degree of robustness and show that inference under the $\beta$-D can be designed so that a single outlier never results in false declaration of a CP, which is impossible under the KLD. Section 3 motivates efficient Structural Variational Inference (SVI) with the $\beta$-D posterior. Within BOCPD, we propose to scale SVI using variance-reduced Stochastic Gradient Descent. Next, Section 4 expands on how $\beta$ can be initialized before the algorithm is run and then optimized on-line during execution time. Lastly, Section 5 showcases the substantial gains in performance of robust BOCPD when compared to its standard version on real world data in terms of both predictive error and CP detection.

## 2 Using Bayesian On-line Changepoint Detection with $\beta$-Divergences

BOCPD is based on the Product Partition Model [3] and introduced independently in Adams and MacKay [1] and Fearnhead and Liu [11]. Recently, both formulations have been unified in Knoblauch and Damoulas [25]. The underlying algorithm has extensions ranging from Gaussian Processes [41] and on-line hyperparameter optimization [8] to non-exponential families [44, 34].

To formulate BOCPD probabilistically, define the run-length $r_t$ as the number of observations at time $t$ since the most recent CP and $m_t$ as the best model in the set $\mathcal{M}$ for the observations since that CP. Then, given a real-valued multivariate process $\{\boldsymbol{y}_t\}_{t=1}^{\infty}$ of dimension $d$, a model universe $\mathcal{M}$, a

run-length prior $h$ defined over $\mathbb{N}_0$ and a model prior $q$ over $\mathcal{M}$, the BOCPD model is

$$r_t|r_{t-1} \sim H(r_t, r_{t-1}) \qquad m_t|m_{t-1}, r_t \sim q(m_t|m_{t-1}, r_t) \tag{1a}$$

$$\boldsymbol{\theta}_m|m_t \sim \pi_{m_t}(\boldsymbol{\theta}_{m_t}) \qquad \boldsymbol{y}_t|m_t, \boldsymbol{\theta}_{m_t} \sim f_{m_t}(\boldsymbol{y}_t|\boldsymbol{\theta}_{m_t}) \tag{1b}$$

where $q(m_t|m_{t-1}, r_t) = 1_{m_{t-1}}(m_t)$ for $r_t > 0$ and $q(m_t)$ otherwise, and where $H$ is the conditional run-length prior so that $H(0, r) = h(r+1)$, $H(r+1, r) = 1-h(r+1)$ for any $r \in \mathbb{N}_0$ and $H(r, r') = 0$ otherwise. For example, Bayesian Linear Regression (BLR) with the $d \times p$ regressor matrix $\boldsymbol{X}_t$ and prior covariance $\Sigma_0$ is given by $\boldsymbol{\theta}_m = (\sigma^2, \boldsymbol{\mu})$, $f_m(\boldsymbol{y}_t|\boldsymbol{\theta}_m) = \mathcal{N}_d(\boldsymbol{y}_t; \boldsymbol{X}_t\boldsymbol{\mu}, I_d)$ and $\pi_m(\boldsymbol{\theta}_m) = \mathcal{N}_d(\boldsymbol{\mu}; \boldsymbol{\mu}_0, \sigma^2\Sigma_0)\mathcal{IG}(\sigma^2; a_0, b_0)$. If the computations of the parameter posterior $\pi_m(\boldsymbol{\theta}_m|\boldsymbol{y}_{1:t}, r_t)$ and the posterior predictive $f_m(\boldsymbol{y}_t|\boldsymbol{y}_{1:(t-1)}, r_t) = \int_{\Theta_m} f_m(\boldsymbol{y}_t|\boldsymbol{\theta}_m)\pi_m(\boldsymbol{\theta}_m|\boldsymbol{y}_{1:(t-1)}, r_t)d\boldsymbol{\theta}_m$ are efficient for all models $m \in \mathcal{M}$, then so is the recursive computation given by

$$p(\boldsymbol{y}_1, r_1 = 0, m_1) = q(m_1) \cdot \int_{\Theta_{m_1}} f_{m_1}(\boldsymbol{y}_1|\boldsymbol{\theta}_{m_1})\pi_{m_1}(\boldsymbol{\theta}_{m_1})d\boldsymbol{\theta}_{m_1} = q(m_1) \cdot f_{m_1}(\boldsymbol{y}_1|\boldsymbol{y}_0), \tag{2a}$$

$$p(\boldsymbol{y}_{1:t}, r_t, m_t) = \sum_{m_{t-1}, r_{t-1}} \Big\{ f_{m_t}(\boldsymbol{y}_t|\mathcal{F}_{t-1})q(m_t|\mathcal{F}_{t-1}, m_{t-1})H(r_t, r_{t-1})p(\boldsymbol{y}_{1:(t-1)}, r_{t-1}, m_{t-1}) \Big\} \tag{2b}$$

where $\mathcal{F}_{t-1} = \{\boldsymbol{y}_{1:(t-1)}, r_{t-1}\}$ and $p(\boldsymbol{y}_{1:t}, r_t, m_t)$ is the joint density of $\boldsymbol{y}_{1:t}$, $m_t$ and $r_t$. The run-length and model posteriors are then available exactly at time $t$, as $p(r_t, m_t|\boldsymbol{y}_{1:t}) = p(\boldsymbol{y}_{1:t}, r_t, m_t)/\sum_{m_t, r_t} p(\boldsymbol{y}_{1:t}, r_t, m_t)$. For a full derivation and the resulting inference see [25].

## 2.1 General Bayesian Inference (GBI) with $\beta$-Divergences ($\beta$-D)

Standard Bayesian inference minimizes the KLD between the Data Generating Mechanism (DGM) and its probabilistic model (see Section 2.1 of [6] for a clear illustration). In the M-closed world where one assumes that the DGM and model coincide, the KLD is the most efficient way of updating posterior beliefs. However, this is no longer the case in the M-open world [5] where they match only approximately [21], e.g. in the presence of outliers. GBI [6, 21] generalizes standard Bayesian updating based on the KLD to a family of divergences. In particular, it uses the relationship between losses $\ell$ and divergences $D$ to deduce for $D$ a corresponding loss $\ell^D$. It can then be shown that for model $m$, the posterior update optimal for $D$ yields the distribution

$$\pi_m^D(\boldsymbol{\theta}_m|\boldsymbol{y}_{(t-r_t):t}) \propto \pi_m(\theta) \exp\left\{ -\sum_{i=t-r_t}^{t} \ell^D(\boldsymbol{\theta}_m|\boldsymbol{y}_i) \right\}. \tag{3}$$

For parameter inference with the KLD and $\beta$-D, these losses are the log score and the Tsallis score:

$$\ell^{\text{KLD}}(\boldsymbol{\theta}_m|\boldsymbol{y}_t) = -\log\left(f_m(\boldsymbol{y}_t|\boldsymbol{\theta}_m\right) \tag{4}$$

$$\ell^\beta(\boldsymbol{\theta}_m|\boldsymbol{y}_t) = -\left( \frac{1}{\beta_{\text{p}}} f_m(\boldsymbol{y}_t|\boldsymbol{\theta}_m)^{\beta_{\text{p}}} - \frac{1}{1+\beta_{\text{p}}} \int_{\mathcal{Y}} f_m(\boldsymbol{z}|\boldsymbol{\theta}_m)^{1+\beta_{\text{p}}}d\boldsymbol{z} \right). \tag{5}$$

Eq. (5) shows why the $\beta$-D excels at robust inference: Similar to tempering, $\ell^\beta$ exponentially downweights the density, attaching less influence to observations in the tails of the model. This phenomenon is depicted with influence functions $I(\boldsymbol{y}_t)$ in Figure 1 **A**. $I(\boldsymbol{y}_t)$ is a divergence between the posterior with and without an observation $\boldsymbol{y}_t$ [28].

GBI with the $\beta$-D yields robust inference without the need to specify a heavy-tailed or otherwise robustified model. Hence, one estimates the same model parameters as in standard Bayesian inference while down-weighting the influence of observations that are overly inconsistent with the model. Accordingly, GBI provides robust inference for a much wider class of models and situations than the ones illustrated here. Though other divergences such as $\alpha$-Divergences [e.g. 19] also accommodate robust inference, we restrict ourselves to the $\beta$-D. We do this because unlike other divergences, it does not require estimation of the DGM's density. Density estimation increases estimation error, is computationally cumbersome and works poorly for small run-lengths (i.e. sample sizes). Note that versions of GBI have been proposed before [14, 32, 38, 10], but have framed the procedure as alternative to Variational Bayes instead.

Apart from the computational gains of Section 3.1, we tackle robust inference via the $\beta$-D rather than via Student's $t$ errors for three reasons: Firstly, robust run-length posteriors need robustness in *ratios* rather than *tails* (see Section 2.3 and the simulation results for Student's $t$ errors in the Appendix). Secondly, Student's $t$ errors model outliers as part of the DGM, which compromises

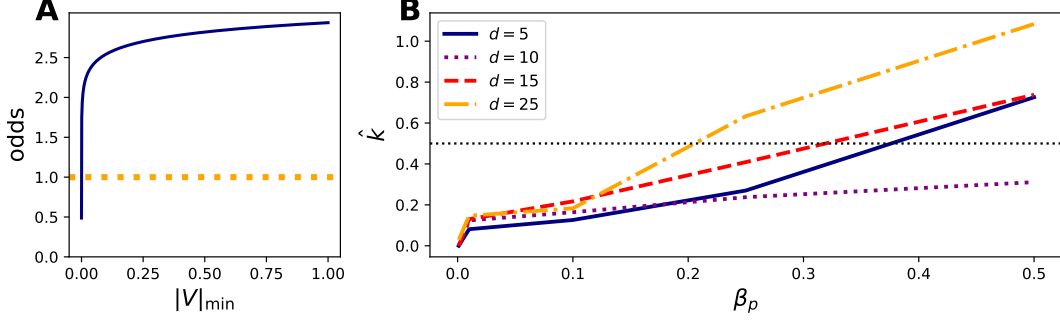

Figure 2: **A**: Lower bound on the odds of Thm. 1 for priors used for Figure 1 **B** and $h(r) = 1/100$. **B**: $\hat{k}$ for different choices of $\beta_{\mathrm{p}}$ and output (input) dimensions $d$ ($2d$) in an autoregressive BLR

the inference target: Consider a BLR with error $e_t = \varepsilon_t + w_t \nu_t$, where $w_t \sim \mathrm{Ber}(p)$ for $p = 0.01$, $\varepsilon_t \sim \mathcal{N}(0, \sigma^2)$ with outliers $\nu_t \sim t_1(0, \gamma)$. Appropriate choices of $\beta_{\mathrm{p}}$ give most influence to the $(1 - p) \cdot 100\% = 99\%$ of typical observations one can explain well with the BLR model. In contrast, modeling $e_t$ as Student's $t$ under the KLD lets $\nu_t$ dominate parameter inference and lets 1% of observations inflate the predictive variance substantially. Thirdly, using Student's $t$ errors is a technique only applicable to symmetric, continuous models. In contrast, GBI with the $\beta$-D is valid for any setting, e.g. for asymmetric errors as well as point and count processes.

## 2.2 Robust BOCPD

The literature on robust on-line CP detection so far is sparse and covers limited settings without Bayesian uncertainty quantification [e.g. 36, 7, 12]. For example, the method in Fearnhead and Rigaill [12] only produces point estimates and is limited to fitting a piecewise constant function to univariate data. In contrast, BOCPD can be applied to multivariate data and a set of models $\mathcal{M}$ while quantifying uncertainty about these models, their parameters and potential CPs, but is not robust. Noting that for standard BOCPD the posterior expectation is given by

$$\mathbb{E}\left(\boldsymbol{y}_t | \boldsymbol{y}_{1:(t-1)}\right) = \sum_{r_t, m_t} \mathbb{E}\left(\boldsymbol{y}_t | \boldsymbol{y}_{1:(t-1)}, r_{t-1}, m_{t-1}\right) p(r_{t-1}, m_{t-1} | \boldsymbol{y}_{1:(t-1)}), \qquad (6)$$

the key observation is that prediction is driven by two probability distributions: The run-length and model posterior $p(r_t, m_t | \boldsymbol{y}_{1:t})$ and parameter posterior distributions $\pi_m(\boldsymbol{\theta}_m | \boldsymbol{y}_{1:t})$. Thus, we make BOCPD robust by using $\beta$-D posteriors $p^{\beta_{\mathrm{rlm}}}(r_t, m_t | \boldsymbol{y}_{1:t})$, $\pi_m^{\beta_{\mathrm{p}}}(\boldsymbol{\theta}_m | \boldsymbol{y}_{1:t})$ for $\boldsymbol{\beta} = (\beta_{\mathrm{rlm}}, \beta_{\mathrm{p}}) > 0^1$.

$\beta_{\mathrm{rlm}}$ prevents abrupt changes in $p^{\beta_{\mathrm{rlm}}}(r_t, m_t | \boldsymbol{y}_{1:t})$ caused by a small number of observations, see section 2.3. This form of robustness is easy to implement and retains the closed forms of BOCPD: In Eqs. (2a) and (2b), one simply replaces $f_{m_t}(\boldsymbol{y}_t | \boldsymbol{y}_0)$ and $f_{m_t}(\boldsymbol{y}_t | \mathcal{F}_{t-1})$ by their $\beta$-D-counterparts $\exp\{\ell^{\beta_{\mathrm{rlm}}}(\boldsymbol{\theta}_{m_t} | \boldsymbol{y}_t)\}$, where

$$\ell^{\beta_{\mathrm{rlm}}}(\boldsymbol{\theta}_{m_t} | \boldsymbol{y}_t) = -\left( \frac{1}{\beta_{\mathrm{rlm}}} f_m(\boldsymbol{y}_t | \mathcal{F}_{t-1})^{\beta_{\mathrm{rlm}}} - \frac{1}{1 + \beta_{\mathrm{rlm}}} \int_{\mathcal{Y}} f_m(\boldsymbol{z} | \mathcal{F}_{t-1})^{1+\beta_{\mathrm{rlm}}} d\boldsymbol{z} \right). \qquad (7)$$

While the posterior $p^{\beta_{\mathrm{rlm}}}(r_t, m_t | \boldsymbol{y}_{1:t})$ is only available up to a constant, it is discrete and thus easy to normalize. Complementing this, $\beta_{\mathrm{p}}$ regulates the robustness of $\pi_m^{\beta_{\mathrm{p}}}(\boldsymbol{\theta} | \boldsymbol{y}_{1:t})$ by preventing it from being dominated by tail events. Section 3.1 overcomes the intractability of $\pi_m^{\beta_{\mathrm{p}}}(\boldsymbol{\theta} | \boldsymbol{y}_{1:t})$ using Structural Variational Inference (SVI) that recovers the approximated distribution exactly as $\beta_{\mathrm{p}} \to 0$.

## 2.3 Quantifying robustness

The algorithm of Fearnhead and Rigaill [12] is robust because hyperparameters enforce that a single outlier is insufficient for declaring a CP. Analogously, we investigate conditions under which a single (outlying) observation $\boldsymbol{y}_{t+1}$ is able to force a CP. An intuitive way of achieving this is by studying the odds of $r_{t+1} \in \{0, r+1\}$ conditional on $r_t = r$:

$$\frac{p(r_{t+1} = r + 1 | \boldsymbol{y}_{1:t+1}, r_t = r, m_t)}{p(r_{t+1} = 0 | \boldsymbol{y}_{1:t+1}, r_t = r, m_t)} = \frac{p(\boldsymbol{y}_{1:t}, r_t = r, m_t) \cdot (1 - H(r_{t+1}, r_t)) f_{m_t}^D(\boldsymbol{y}_{t+1} | \mathcal{F}_t)}{p(\boldsymbol{y}_{1:t}, r_t = r, m_t) \cdot H(r_{t+1}, r_t) f_{m_t}^D(\boldsymbol{y}_{t+1} | \boldsymbol{y}_0)}. \qquad (8)$$

Here, $f_{m_t}^D$ denotes the negative exponential of the score under divergence $D$. In particular, $f_{m_t}^{\text{KLD}}(\boldsymbol{y}_{t+1}|\mathcal{F}_t) = f_{m_t}(\boldsymbol{y}_{t+1}|\mathcal{F}_t)$ and $f_{m_t}^{\beta_{\text{rlm}}}(\boldsymbol{y}_{t+1}|\mathcal{F}_t) = \exp\left\{-\ell^{\beta_{\text{rlm}}}(\boldsymbol{\theta}_m|\boldsymbol{y}_t)\right\}$ as in Eq. (7). Taking a closer look at Eq. (8), if $\boldsymbol{y}_{t+1}$ is an outlier with low density under $f_{m_t}^D(\boldsymbol{y}_{t+1}|\mathcal{F}_t)$, the odds will move in favor of a CP provided that the prior is sufficiently uninformative to make $f_{m_t}^D(\boldsymbol{y}_{t+1}|\boldsymbol{y}_0) > f_{m_t}^D(\boldsymbol{y}_{t+1}|\mathcal{F}_t)$. In fact, even very small differences have a substantial impact on the odds. This is why using the Student's $t$ error for the BLR model with standard Bayes will not provide robust run-length posteriors: While an outlying observation $\boldsymbol{y}_{t+1}$ will have greater density $f_{m_t}^{\text{KLD}}(\boldsymbol{y}_{t+1}|\mathcal{F}_t)$ under a Student's $t$ error model than under a normal error model, $f_{m_t}^{\text{KLD}}(\boldsymbol{y}_{t+1}|\boldsymbol{y}_0)$ (the density under the prior) will also be larger under the Student's $t$ error model. As a result, changing the tails of the model only has a very limited effect on the ratio in Eq. (8). In fact, the perhaps unintuitive consequence is that Student's $t$ error models will yield CP inference that very closely resembles that of the corresponding normal model. A range of numerical examples in the Appendix illustrate this surprising fact. In contrast, CP inference robustified via the $\beta$-D does not suffer from this phenomenon. In fact, Theorem 1 provides very mild conditions for the $\beta$-D robustified BLR model ensuring that the odds never favor a CP after *any* single outlying observation $\boldsymbol{y}_{t+1}$.

**Theorem 1.** If $m_t$ in Eq. (8) is the Bayesian Linear Regression (BLR) model with $\boldsymbol{\mu} \in \mathbb{R}^p$ and priors $a_0, b_0, \mu_0, \Sigma_0$; and if the posterior predictive's variance determinant is larger than $|V|_{\min} > 0$, then one can choose any $(\beta_{\text{rlm}}, H(r_t, r_{t+1})) \in S(p, \beta_{\text{rlm}}, a_0, b_0, \mu_0, \Sigma_0, |V|_{\min})$ to guarantee that

$$\frac{(1 - H(r_{t+1}, r_t))f_{m_t}^{\beta_{\text{rlm}}}(\boldsymbol{y}_{t+1}|\mathcal{F}_t)}{H(r_{t+1}, r_t)f_{m_t}^{\beta_{\text{rlm}}}(\boldsymbol{y}_{t+1}|\boldsymbol{y}_0)} \geq 1, \tag{9}$$

where the set $S(p, \beta_{\text{rlm}}, a_0, b_0, \mu_0, \Sigma_0, |V|_{\min})$ is defined by an inequality given in the Appendix.

Thm. 1 says that one can bound the odds for a CP independently of $\boldsymbol{y}_{t+1}$. The requirement for a lower bound $|V|_{\min}$ results from the integral term in Eq. (5), which dominates $\beta$-D-inference if $|V|$ is extremely small. In practice, this is not restrictive: E.g. for $p = 5$, $h(r) = \frac{1}{\lambda}$, $a_0 = 3, b_0 = 5, \Sigma_0 = \text{diag}(100, 5)$ used in Fig. 1 **B**, Thm. 1 holds for $(\beta_{\text{rlm}}, \lambda) = (0.15, 100)$ used for inference if $|V|_{\min} \geq 8.12 \times 10^{-6}$. Fig. 2 **A** plots the lower bound (see Appendix) as function of $|V|_{\min}$.

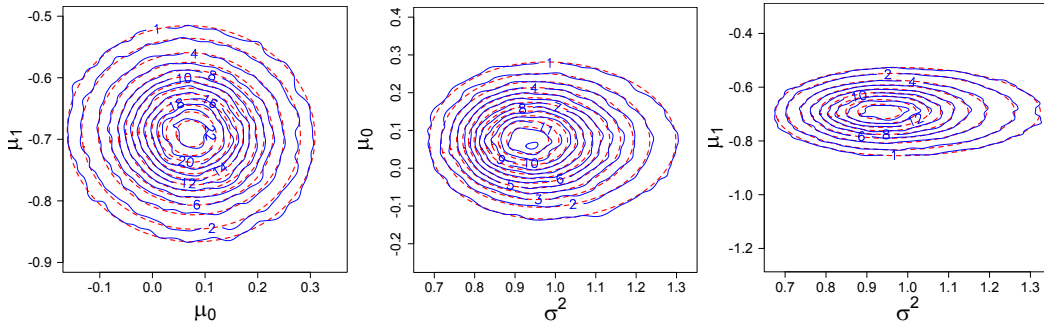

Figure 3: Exemplary contour plots of bivariate marginals for the approximation $\widehat{\pi}_m^{\beta_{\text{p}}}(\boldsymbol{\theta}_m)$ of Eq. (11) (dashed) and the target $\pi_m^{\beta_{\text{p}}}(\boldsymbol{\theta}_m|\boldsymbol{y}_{(t-r_t):t})$ (solid) estimated and smoothed from $95,000$ Hamiltonian Monte Carlo samples for the $\beta$-D posterior of BLR with $d = 1$, two regressors and $\beta_{\text{p}} = 0.25$.

## 3 On-line General Bayesian Inference (GBI)

### 3.1 Structural Variational Approximations for Conjugate Exponential Families

While there has been a recent surge in theoretical work on GBI [6, 15, 21, 14], applications have been sparse, in large part due to intractability. While sampling methods have been used successfully for GBI [21, 15], it is not easy to scale these for the robust BOCPD setting. Thus, most work on BOCPD has focused on conjugate distributions [1, 43, 11] and approximations [44, 34]. We extend the latter branch of research by deploying Structural Variational Inference (SVI). Unlike mean-field approximations, this preserves parameter dependence in the posterior, see Figure 3. While it is in principle possible to solve the inference task by sampling, this is computationally burdensome and makes the algorithm on-line in name only: Any sampling approach needs to (I) sample from $\pi_m^{\beta_p}(\boldsymbol{\theta}_m|\boldsymbol{y}_{t-r_t:t})$ in Eq. (3), (II) numerically integrate to obtain $f_m(\boldsymbol{y}_t|\boldsymbol{y}_{1:(t-1)}, r_t)$ and lastly (III)

sample and numerically integrate the integral in Eq. (7) which no longer has a closed form. Moreover, this has to be performed for each $(r_t, m)$ at times $t = 1, 2, \ldots$. On top of this increased computational cost, it creates three sources of approximation error propagated forward through time via Eqs. (2a) and (2b). Since $\pi_m^{\text{KLD}}$ is available in closed form and as $\beta\text{-D} \to \text{KLD}$ as $\beta \to 0$ [4], there is an especially compelling way of doing SVI for conjugate models using the $\beta$-D based on the fact that

$$\pi_m^{\beta_{\text{p}}}(\boldsymbol{\theta}_m | \boldsymbol{y}_{(t-r_t):t}) \approx \pi_m^{\text{KLD}}(\boldsymbol{\theta}_m | \boldsymbol{y}_{(t-r_t):t}) \tag{10}$$

is exact as $\beta \to 0$. Thus we approximate the $\beta$-D posterior for model $m$ and run-length $r_t$ as

$$\widehat{\pi}_m^{\beta_{\text{p}}}(\boldsymbol{\theta}_m) = \operatorname*{argmin}_{\pi_m^{\text{KLD}}(\boldsymbol{\theta}_m)} \left\{ \text{KL}\left( \pi_m^{\text{KLD}}(\boldsymbol{\theta}_m) \,\Big\|\, \pi_m^{\beta_{\text{p}}}(\boldsymbol{\theta}_m | \boldsymbol{y}_{(t-r_t):t}) \right) \right\}. \tag{11}$$

While this ensures that the densities $\widehat{\pi}_m^{\beta_{\text{p}}}$ and $\pi_m^{\text{KLD}}$ belong to the same family, the variational parameters can be very different from those implied by the KLD-posterior. This approximation mitigates multiple issues that would arise with sampling approaches: By forcing $\pi_m^{\beta_{\text{p}}}(\theta_m | \boldsymbol{y}_{1:t})$ into the conjugate closed form, steps (II) and (III) are solved analytically. Thus, inference is orders of magnitude faster, while the resulting approximation error remains negligible (see Figs 2B, 3).

Moreover, for many models, the Evidence Lower Bound (ELBO) associated with the optimization in Eq. (11) is available in closed form. As a result, off-the-shelf optimizers are sufficient and no black-box or sampling-based techniques are required to efficiently tackle the problem. Theorem 2 provides the conditions for a conjugate exponential family to admit such a closed form ELBO. The proof alongside the derivation of the ELBO for BLR can be found in the Appendix

**Theorem 2.** The ELBO objective corresponding to the $\beta$-D posterior approximation in Eq. (11) of an exponential family likelihood model $f_m(\boldsymbol{y}; \theta_m) = \exp\left(\eta(\theta_m)^T T(\boldsymbol{y})\right) g(\eta(\theta_m)) A(x)$ with conjugate prior $\pi_0(\theta_m | \nu_0, \mathcal{X}_0) = g(\eta(\theta_m))^{\nu_0} \exp\left(\nu_0 \eta(\theta_m)^T \mathcal{X}_0\right) h(\mathcal{X}_0, \nu_0)$ and variational posterior $\widehat{\pi}_m^{\beta_{\text{p}}}(\theta_m | \nu_m, \mathcal{X}_m) = g(\eta(\theta_m))^{\nu_m} \exp\left(\nu_m \eta(\theta_m)^T \mathcal{X}_m\right) h(\mathcal{X}_m, \nu_m)$ within the same conjugate family is analytically available iff the following three quantities have closed form:

$$\mathbb{E}_{\widehat{\pi}_m^{\beta_{\text{p}}}}\left[\eta(\theta_m)\right], \; \mathbb{E}_{\widehat{\pi}_m^{\beta_{\text{p}}}}\left[\log g(\eta(\theta_m))\right], \; \int A(z)^{1+\beta_{\text{p}}}\left[h\left(\frac{(1+\beta_{\text{p}})T(z) + \nu_m \mathcal{X}_m}{1 + \beta_{\text{p}} + \nu_m}, 1 + \beta + \nu_m\right)\right]^{-1} dz.$$

The conditions of Theorem 2 are met by many exponential models, e.g. the Normal-Inverse-Gamma, the Exponential-Gamma, and the Gamma-Gamma. For a simulated autoregressive BLR, we assess the quality of $\widehat{\pi}^{\beta_{\text{p}}}$ following Yao et al. [48], who estimate a difference $\hat{k}$ between $\pi_m^{\beta_{\text{p}}}$ and $\widehat{\pi}_m^{\beta_{\text{p}}}$ relative to a posterior expectation. We use this on the posterior predictive, which is an expectation relative to $\pi_m^{\beta_{\text{p}}}$ and drives the CP detection. Yao et al. [48] rate $\widehat{\pi}_m^{\beta_{\text{p}}}$ as *close* to $\pi_m^{\beta_{\text{p}}}$ if $\hat{k} < 0.5$. Figs 3 and 2 **B** show that our approximation lies well below this threshold for choices of $\beta_{\text{p}}$ decreasing reasonably fast with the dimension. Note that these are exactly the values of $\beta_{\text{p}}$ one will want to select for inference: As $d$ increases, the magnitude of $f_{m_t}(\boldsymbol{y}_t | \mathcal{F}_{t-1})$ decreases rapidly. Hence, $\beta_{\text{p}}$ needs to decrease as $d$ increases to prevent the $\beta$-D inference from being dominated by the integral in Eq. (5) and disregarding $\boldsymbol{y}_t$ [21]. This is also reflected in our experiments in section 5, for which we initialize $\beta_{\text{p}} = 0.05$ and $\beta_{\text{p}} = 0.005$ for $d = 1$ and $d = 29$, respectively. However, as Figs. 3 and 2 **B** illustrate, the approximation is still excellent for values of $\beta_{\text{p}}$ that are much larger than that.

### 3.2 Stochastic Variance Reduced Gradient (SVRG) for BOCPD

While highest predictive accuracy within BOCPD is achieved using full optimization of the variational parameters at each of $T$ time periods, this has space and time complexity of $\mathcal{O}(T)$ and $\mathcal{O}(T^2)$. In comparison, Stochastic Gradient Descent (SGD) has space and time complexity of $\mathcal{O}(1)$ and $\mathcal{O}(T)$, but yields a loss in accuracy, substantially so for small run-lengths. In the BOCPD setting, there is an obvious trade-off between accuracy and scalability: Since the posterior predictive distributions $f_{m_t}(\boldsymbol{y}_t | \boldsymbol{y}_{1:(t-1)}, r_t)$ for all run-lengths $r_t$ drive CP detection, SGD estimates are insufficiently accurate for small run-lengths $r_t$. On the other hand, once $r_t$ is sufficiently large, the variational parameter estimates only need minor adjustments and computing an optimum is costly.

Recently, a new generation of algorithms interpolating SGD and global optimization have addressed this trade-off. They achieve substantially better convergence rates by anchoring the stochastic gradient to a point near an optimum [22, 9, 35, 18, 29]. We propose a memory-efficient two-stage variation of

**Stochastic Variance Reduced Gradient (SVRG) inference for BOCPD**

---

**Input at time** $0$**:** Window & batch sizes $W$, $B^*$, $b^*$; frequency $m$, prior $\boldsymbol{\theta}_0$, #steps $K$, step size $\eta$
   s.t. $W > B^* > b^*$; and $\sim$ denotes sampling without replacement

**for**  next observation $\boldsymbol{y}_t$ at time $t$ **do**
    **for** retained run-lengths $r \in R(t)$  **do**
        **if** $\tau_r = 0$ **then**
            **if** $r < W$ **then**
                $\boldsymbol{\theta}_r \leftarrow \boldsymbol{\theta}_r^* \leftarrow \text{FullOpt}\left(\text{ELBO}(\boldsymbol{y}_{t-r:t})\right); \tau_r \leftarrow m$
            **else if**  $r \geq W$ **then**
                $\boldsymbol{\theta}_r^* \leftarrow \boldsymbol{\theta}_r; \tau_r \leftarrow \text{Geom}\left(B^*/(B^* + b^*)\right)$
            $B \leftarrow \min(B^*, r)$
            $g_r^{\text{anchor}} \leftarrow \frac{1}{B} \sum_{i \in \mathcal{I}} \nabla\text{ELBO}(\boldsymbol{\theta}_r^*, \boldsymbol{y}_{t-i})$, where $\mathcal{I} \sim \text{Unif}\{0, \ldots, \min(r, W)\}, |\mathcal{I}| = B$
            **for**  $j = 1, 2, \ldots, K$ **do**
                $b \leftarrow \min(b^*, r)$ and $\widetilde{\mathcal{I}} \sim \text{Unif}\{0, \ldots, \min(r, W)\}$ and $|\widetilde{\mathcal{I}}| = b$
                $g_r^{\text{old}} \leftarrow \frac{1}{b} \sum_{i \in \widetilde{\mathcal{I}}} \nabla\text{ELBO}(\boldsymbol{\theta}_r^*, \boldsymbol{y}_{t-i}), \; g_r^{\text{new}} \leftarrow \frac{1}{b} \sum_{i \in \widetilde{\mathcal{I}}} \nabla\text{ELBO}(\boldsymbol{\theta}_r, \boldsymbol{y}_{t-i})$
                $\boldsymbol{\theta}_r \leftarrow \boldsymbol{\theta}_r + \eta \cdot \left(g_r^{\text{new}} - g_r^{\text{old}} + g_r^{\text{anchor}}\right); \tau_r \leftarrow \tau_r - 1$
    $r \leftarrow r + 1$ for all $r \in R(t)$; $R(t) \leftarrow R(t) \cup \{0\}$

---

these methods tailored to BOCPD. First, the variational parameters are moved close to their global optimum using a variant of [22, 35]. Unlike standard versions, we anchor the gradient estimates to a (local) optimum by calling a convex optimizer FullOpt every $m$ steps for the first $W$ iterations. While our implementation uses Python scipy's L-BFSG-B optimization routine, any convex optimizer could be used for this step. Compared to standard SGD or SVRG, full optimization substantially decreases variance and increases accuracy for small $r_t$. Second, once $r_t > W$ we do not perform full optimization anymore. Instead, we anchor optimization to the current value as in standard SVRG, by updating the anchor at stochastic time intervals determined by a geometric random variable with success probability $B^*/(B^* + b^*)$. Whether the anchor is based on global optimization or not, the next step consists in sampling $B = \min(r_t, B^*)$ observations without replacement from a window with the $\min(r_t, W)$ most recent observations to initiate the SVRG procedure. Following this, for the next $K$ observations, we incrementally refine the estimates while keeping their variance low using a stochastic-batch variant of [29, 30] by sampling a batch of size $b = \min(r_t, b^*)$ without replacement from the $\min(r_t, W)$ most recent observations. The resulting on-line inference has constant space and linear time complexity like SGD, but produces good estimates for small $r_t$ and converges faster [22, 29, 30]. We provide a detailed complexity analysis of the procedure in the Appendix, where we also demonstrate numerically that it is orders of magnitude faster than MCMC-based inference.

## 4   Choice of $\beta$

**Initializing** $\beta_{\text{p}}$: The $\beta$-D has been used in a variety of settings [15, 4, 14, 49], but there is no principled framework for selecting $\beta$. We remedy this by minimizing the expected predictive loss with respect to $\beta$ on-line. As the losses need not be convex in $\beta_{\text{p}}$, initial values can matter for the optimization. A priori, we pick $\beta_{\text{p}}$ maximizing the $\beta$-D influence for a given Mahalanobis Distance (MD) $x^*$ under $\pi(\theta_m)$. As Figure 1 **A** shows, $\beta_{\text{p}} > 0$ induces a point of maximum influence $\text{MD}(\beta_{\text{p}}, \pi_m(\theta_m))$: Points further in the tails are treated as outliers, while points closer to the mode receive similar influence as under the KLD. A Monte Carlo estimate of $\text{MD}(\beta_{\text{p}}, \pi_m(\theta_m))$ is found via $\widehat{\text{MD}}(\beta_{\text{p}}, \pi_m(\theta_m)) = \text{argmax}_{x \in \mathbb{R}_+} \hat{I}(\beta_{\text{p}}, \pi_m(\theta_m))(x)$ [28]. We initialize $\beta_{\text{p}}$ by solving the inverse problem: For $x^*$, we seek $\beta_{\text{p}}$ such that $\widehat{\text{MD}}(\beta_{\text{p}}, \pi_m(\theta_m)) = x^*$. (The Appendix contains a pictorial illustration of this procedure.) The $k$-th standard deviation under the prior is a good choice of $x^*$ for low dimensions [see also 12], but not appropriate as delimiter for high density regions even in moderate dimensions $d$. Thus, we propose $x^* = \sqrt{d}$ for larger values of $d$, inspired by the fact that under normality, $\text{MD} \to \sqrt{d}$ as $d \to \infty$ [17]. One then finds $\beta_{\text{p}}$ by approximating the gradient of $\widehat{\text{MD}}(\beta_{\text{p}}, \pi_m(\theta_m))$ with respect to $\beta_{\text{p}}$. As $\beta_{\text{rlm}}$ does not affect $\pi_m^{\beta_{\text{p}}}$, its initialization matters less and generally, initializing $\beta_{\text{rlm}} \in [0, 1]$ produces reasonable results.

**Optimizing $\boldsymbol{\beta}$ on-line**: For $\boldsymbol{\beta} = (\beta_{\text{rlm}}, \beta_{\text{p}})$ and prediction $\widehat{\boldsymbol{y}}_t(\boldsymbol{\beta})$ of $\boldsymbol{y}_t$ obtained as posterior expectation via Eq. (6), define $\boldsymbol{\varepsilon}_t(\boldsymbol{\beta}) = \boldsymbol{y}_t - \widehat{\boldsymbol{y}}_t(\boldsymbol{\beta})$. For predictive loss $L : \mathbb{R} \to \mathbb{R}_+$, we target $\boldsymbol{\beta}^* = \text{argmin}_{\boldsymbol{\beta}} \{\mathbb{E}(L(\boldsymbol{\varepsilon}_t(\boldsymbol{\beta})))\}$. Replacing expected by empirical loss and deploying SGD, we seek to find the partial derivatives of $\nabla_{\boldsymbol{\beta}} L(\boldsymbol{\varepsilon}_t(\boldsymbol{\beta}))$. Noting that $\nabla_{\boldsymbol{\beta}} L(\boldsymbol{\varepsilon}_t(\boldsymbol{\beta}))) = L'(\boldsymbol{\varepsilon}_t(\boldsymbol{\beta})) \cdot \nabla_{\boldsymbol{\beta}} \widehat{\boldsymbol{y}}_t(\boldsymbol{\beta})$, the issue reduces to finding the partial derivatives $\nabla_{\beta_{\text{rlm}}} \widehat{\boldsymbol{y}}_t(\boldsymbol{\beta})$ and $\nabla_{\beta_{\text{p}}} \widehat{\boldsymbol{y}}_t(\boldsymbol{\beta})$. Remarkably, $\nabla_{\beta_{\text{rlm}}} \widehat{\boldsymbol{y}}_t(\boldsymbol{\beta})$ can be updated sequentially and efficiently by differentiating the recursion in Eq. (2b). The derivation is provided in the Appendix. The gradient $\nabla_{\beta_{\text{p}}} \widehat{\boldsymbol{y}}_t(\boldsymbol{\beta})$ on the other hand is not available analytically and thus is approximated numerically. Now, $\beta$ can be updated on-line via

$$\boldsymbol{\beta}_t = \boldsymbol{\beta}_{t-1} - \eta \cdot \begin{bmatrix} \nabla_{\beta_{\text{rlm},t}} L\left(\varepsilon_t(\boldsymbol{\beta}_{1:(t-1)})\right) \\ \nabla_{\beta_{\text{p},t}} L\left(\varepsilon_t(\boldsymbol{\beta}_{1:(t-1)})\right) \end{bmatrix} \tag{12}$$

In spirit, this procedure resembles existing approaches for model hyperparameter optimization [8]. For robustness, $L$ should be chosen appropriately. In our experiments $L$ is a bounded absolute loss.

## 5 Results

Next, we illustrate the most important improvements this paper makes to BOCPD. First, we show how robust BOCPD deals with outliers on the well-log data set. Further, we show that standard BOCPD breaks down in the M-open world whilst $\beta$-D yields useful inference by analyzing noisy measurements of Nitrogen Oxide (NOX) levels in London. In both experiments, we use the methods in section 4, on-line hyperparameter optimization [8] and pruning for $p(r_t, m_t|\boldsymbol{y}_{1:t})$ [1]. Detailed information is provided in the Appendix. Software and simulation code is available as part of a reproducibility award at `https://github.com/alan-turing-institute/rbocpdms/`.

### 5.1 Well-log

The well-log data set was first studied in Ruanaidh et al. [39] and has become a benchmark data set for univariate CP detection. However, except in Fearnhead and Rigaill [12] its outliers have been removed before CP detection algorithms are run [e.g. 1, 31, 40]. With $\mathcal{M}$ containing one BLR model of form $y_t = \mu + \varepsilon_t$, Figure 4 shows that robust BOCPD deals with outliers on-line. The maximum of the run-length distribution for standard BOCPD is zero 145 times, so declaring CPs based on the run-length distribution's maximum [see e.g. 41] yields a False Discovery Rate (FDR) > 90%. This problem persists even with non-parametric, Gaussian Process, models [p. 186, 45]. Even using Maximum A Posteriori (MAP) segmentation [11], standard BOCPD mislabels 8 outliers as CPs, making for a FDR > 40%. In contrast, the segmentation of the $\beta$-D version does not mislabel any outliers. Morevoer and in accordance with Thm. 1, its run-length distribution's maximum never drops to zero in response to outliers. Further, a natural byproduct of the robust segmentation is a reduction in squared (absolute) prediction error by 10% (6%) compared to the standard version. The

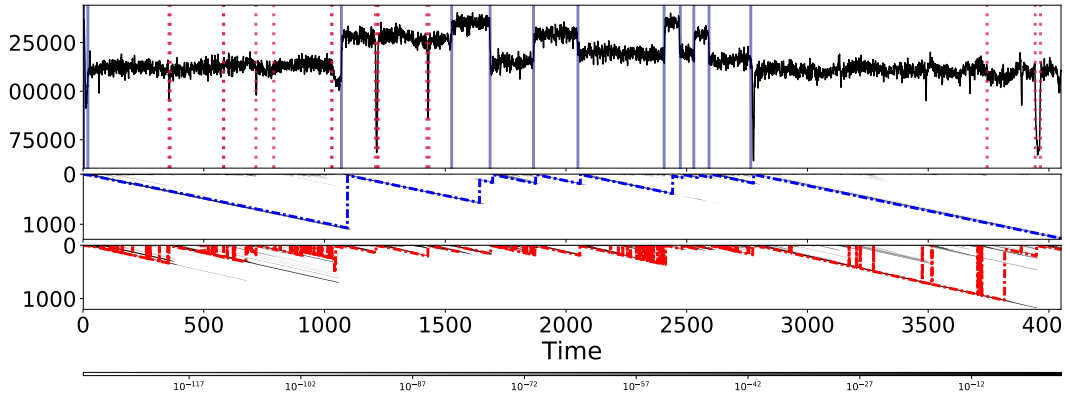

Figure 4: Maximum A Posteriori (MAP) segmentation and run-length distributions of the well-log data. Robust segmentation depicted using solid lines, CPs additionally declared under standard BOCPD with dashed lines. The corresponding run-length distributions for robust (middle) and standard (bottom) BOCPD are shown in grayscale. The most likely run-lengths are dashed.

robust version has more computational overhead than standard BOCPD, but still needs less than 0.5 seconds per observation using a 3.1 GHz Intel i7 and 16GB RAM.

Not only does robust BOCPD's segmentation in Figure 4 match that in Fearnhead and Rigaill [12], but it also offers three additional on-line outputs: Firstly, it produces probabilistic (rather than point) forecasts and parameter inference. Secondly, it self-regulates its robustness via $\beta$. Thirdly, it can compare multiple models and produce model posteriors (see section 5.2). Further, unlike Fearnhead and Rigaill [12], it is not restricted to fitting univariate data with piecewise constant functions.

## 5.2 Air Pollution

The example in Fig. 1 **B** gives an illustration of the importance of robustness in medium-dimensional (BOCPD) problems: It suffices for a *single* dimension of the problem to be misspecified or outlier-prone for inference to fail. Moreover, the presence of misspecification or outliers in this plot can hardly be spotted – and this effect will worsen with increasing dimensionality. To illustrate this point on a multivariate real world data set, we also analyze Nitrogen Oxide (NOX) levels across 29 stations in London using spatially structured Bayesian Vector Autoregressions [see 25]. Previous robust on-line methods [e.g. 36, 7, 12] cannot be applied to this problem because they assume univariate data or do not allow for dependent observations. As Figure 5 shows, robust BOCPD finds one CP corresponding to the introduction of the congestion charge, while standard BOCPD produces an FDR >90%. Both methods find a change in dynamics (i.e. models) after the congestion charge introduction, but variance in the model posterior is substantially lower for the robust algorithm. Further, it increases the average one-step-ahead predictive likelihood by 10% compared to standard BOCPD.

## 6 Conclusion

This paper has presented the very first robust Bayesian on-line changepoint (CP) detection algorithm and the first ever scalable General Bayesian Inference (GBI) method. While CP detection is a particularly salient example of unaddressed heterogeneity and outliers leading to poor inference, the capabilities of GBI and the Structural Variational approximations presented extend far beyond this setting. With an ever increasing interest in the field of machine learning to efficiently and reliably quantify uncertainty, robust probabilistic inference will only become more relevant. In this paper, we give a particularly striking demonstration of the inferential power that can be unlocked through divergence-based General Bayesian inference.

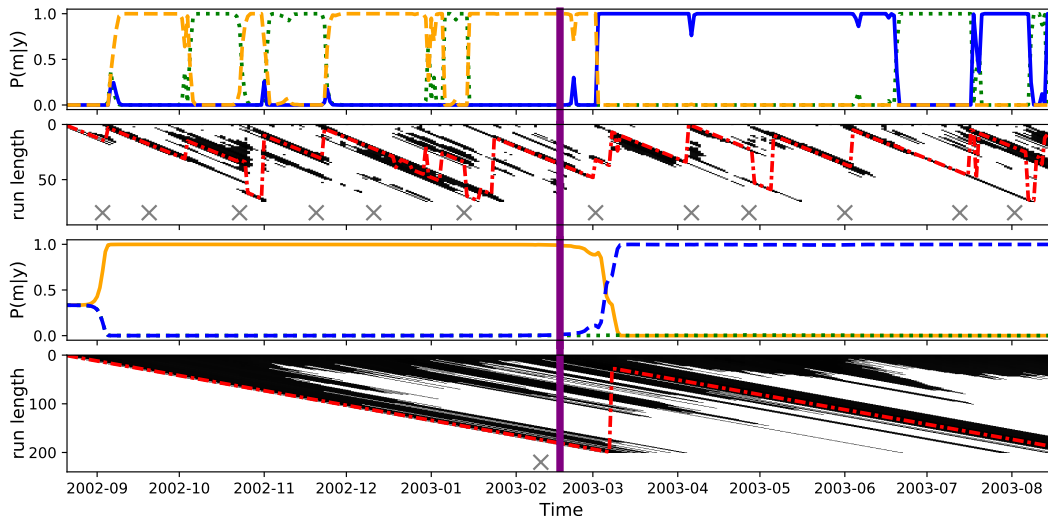

Figure 5: On-line model posteriors for three different VAR models (solid, dashed, dotted) and run-length distributions in grayscale with most likely run-lengths dashed for standard (top two panels) and robust (bottom two panels) BOCPD. Also marked are the congestion charge introduction, 17/02/2003 (solid vertical line) and the MAP segmentations (crosses)

## Acknowledgements

We would like to cordially thank both Jim Smith and Chris Holmes for fruitful discussions and help with some of the theoretical results. JK and JJ are funded by EPSRC grant EP/L016710/1 as part of the Oxford-Warwick Statistics Programme (OxWaSP). TD is funded by the Lloyds Register Foundation programme on Data Centric Engineering through the London Air Quality project. This work was supported by The Alan Turing Institute for Data Science and AI under EPSRC grant EP/N510129/1. In collaboration with the Greater London Authority.

## Footnotes

[1] In fact, $\beta_{\mathrm{p}} = \beta_p^m$, i.e. the robustness is model-specific, but this is suppressed for readability

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
