[Supplementary Material]

# Appendix: Doubly Robust Bayesian Inference for Non-Stationary Streaming Data with $\beta$-Divergences

## Contents

Figure 1: **Top:** Plot of the *well-log* data between $t = 300$ and $t = 500$ with one obvious outlying period. **Middle Top:** KLD run length posterior under the Gaussian error model with the MAP of the run length posterior at each time point overlayed in red. **Middle Bottom:** KLD run length posterior under the Student's $t$ with 5 degrees of freedom error model with the MAP of the run length posterior at each time point overlayed in red. **Bottom:** $\beta$-D run length posterior under the Gaussian error model with $\beta_p = 0.25$ and $\beta_{rlm} = 0.5$ with the MAP of the run length posterior at each time point overlayed in blue.

# 1 Student-t Experiments

In Section 2.3 (Quantifying Robustness) of the paper we argue that substituting the Gaussian error model in the BLR setting for a Student's $t$ error model – a traditional solution for robust parameter inference – will be insufficient to ensure that standard Bayesian run-length posteriors are robust. Here, the type of robustness we refer to is defined in Theorem 1. To demonstrate this, we implement a version of BOCPD using both the Gaussian error model and the Student's $t$ error model on two subsets of the well-log data. The Student's $t$ distribution is no longer an exponential family and thus cannot be implemented in analytical form or via our structural variational approximation. Hence, we used *stan* [Carpenter et al., 2016] for MCMC sampling from the parameter posterior under the Student's $t$ error model. For comparability, hyperparameters were fixed for both the Gaussian and Student's $t$ error models at $\mu_0 = 0$, $\Sigma_0 = \sqrt{5}$, $a_0 = 0.5$, $b_0 = 2$, $h(r_{t+1}) = 0.01 \ \forall r_{t+1}$, where $N = 1000$ values were sampled from the parameter posterior, $M = 25$ run lengths were stored and the degrees of freedom of the Student's $t$ error model were set to be $\nu = 5$. Figures 1 and 2 plot the KLD run-length posteriors of the Gaussian and Student's-$t$ error models as well as the $\beta$-D run-length

posteriors of the Gaussian error models for the two subsets of the *well-log* data. In both examples, the KLD run-length posteriors favor declaring a CP under both the Gaussian and Student's $t$ error model at the first sign of an outlier. In the second example, the outlier is severe enough to permanently disrupt the run-length inference for both KLD-based methods, while the $\beta$-D-based method remains robust. Theorem 1 outlines situations were this desireable behaviour of $\beta$-D-based inference can be guaranteed to happen when it would not happen under the KLD with *any* error model.

Figure 2: **Top:** Plot of the *well-log* data between $t = 1100$ and $t = 1300$ with one obvious outlying period. **Middle:** KLD run length posterior under the Gaussian error model with the MAP of the run length posterior at each time point overlayed in red. **Bottom:** KLD run length posterior under the Student's $t$ with 5 degrees of freedom error model with the MAP of the run length posterior at each time point overlayed in red. **Bottom:** $\beta$-D run length posterior under the Gaussian error model with $\beta_p = 0.5$ and $\beta_{\mathrm{rlm}} = 1$ with the MAP of the run length posterior at each time point overlayed in blue.

## 2 Proof of Theorem 1

*Proof.* This proof looks at the run length posterior parameterised by $\beta_{\text{rlm}}$, however to ease notation we refer to $\beta_{\text{rlm}} = \beta$ throughout. Condition on the event $r_t = r$ then after one time step either $r_{t+1} = r + 1$ or $r_{t+1} = 0$. The odds of these two possibilities are as in Thm. 1. Now substituting the definitions of $f_{m_t}^\beta(\boldsymbol{y}_{t+1}|\mathcal{F}_t)$ and $f_{m_t}^\beta(\boldsymbol{y}_{t+1}|\boldsymbol{y}_0)$ leaves

$$\frac{f_{m_t}^\beta(\boldsymbol{y}_{t+1}|\mathcal{F}_t)}{f_{m_t}^\beta(\boldsymbol{y}_{t+1}|\boldsymbol{y}_0)} = \frac{\exp\left(\frac{1}{\beta}p(\boldsymbol{y}_{t+1}|\boldsymbol{y}_{1:t})^\beta - \frac{1}{1+\beta}\int p(\boldsymbol{z}|\boldsymbol{y}_{1:t})^{1+\beta}d\boldsymbol{z}\right)}{\exp\left(\frac{1}{\beta}p(\boldsymbol{y}_{t+1}|\boldsymbol{y}_0)^\beta - \frac{1}{1+\beta}\int p(\boldsymbol{z}|\boldsymbol{y}_0)^{1+\beta}d\boldsymbol{z}\right)} \tag{1}$$

$$= \exp\left(\frac{1}{\beta}\left(p(\boldsymbol{y}_{t+1}|\boldsymbol{y}_{1:t})^\beta - p(\boldsymbol{y}_{t+1}|\boldsymbol{y}_0)^\beta\right) - \frac{1}{1+\beta}\int p(\boldsymbol{z}|\boldsymbol{y}_{1:t})^{1+\beta} - p(\boldsymbol{z}|\boldsymbol{y}_0)^{1+\beta}d\boldsymbol{z}\right). \tag{2}$$

This proof first seeks a lower bound for this ratio. A lower bound on $\frac{1}{\beta}p(\boldsymbol{y}_{t+1}|\boldsymbol{y}_{1:t})^\beta$ is 0, while the maximal value of $\frac{1}{\beta}p(\boldsymbol{y}_{t+1}|x_0)^\beta$ will occur at the prior mode. For the multivariate $t$-distribution prior predictive with NIG hyperparameters $a_0, b_0, \mu_0, \boldsymbol{\Sigma}_0$ of dimensions $p$ the prior mode has density

$$p(\boldsymbol{\mu}_0|\nu_0, \boldsymbol{\mu}_0, \boldsymbol{V}_0, p) = \frac{\Gamma((\nu_0 + p)/2)}{\Gamma(\nu_0/2)\nu_0^{p/2}\pi^{p/2}|\boldsymbol{V}_0|^{1/2}}\left[1 + \frac{1}{\nu_0}(\boldsymbol{\mu}_0 - \boldsymbol{\mu}_0)\boldsymbol{\Sigma}_0^{-1}(\boldsymbol{\mu}_0 - \boldsymbol{\mu}_0)\right]^{-(\nu_0+p)/2} \tag{3}$$

$$= \frac{\Gamma((\nu_0 + p)/2)}{\Gamma(\nu_0/2)\nu_0^{p/2}\pi^{p/2}|\boldsymbol{V}_0|^{1/2}} \tag{4}$$

$$= \frac{\Gamma(a_0 + p/2)}{\Gamma(a_0)(2b_0\pi)^{p/2}|\boldsymbol{I} + \boldsymbol{X}\boldsymbol{\Sigma}_0\boldsymbol{X}^T|^{1/2}}. \tag{5}$$

As a result the only term in the lower bound of $f_{m_t}^\beta(\boldsymbol{y}_{t+1}|\mathcal{F}_t)/f_{m_t}^\beta(\boldsymbol{y}_{t+1}|\boldsymbol{y}_0)$ that does not solely depend on the prior parameters is $\frac{1}{1+\beta}\int p(\boldsymbol{z}|\boldsymbol{y}_{1:t})^{1+\beta}d\boldsymbol{z}$. This term appears in the negative and thus to lower bound $f_{m_t}^\beta(\boldsymbol{y}_{t+1}|\mathcal{F}_t)/f_{m_t}^\beta(\boldsymbol{y}_{t+1}|\boldsymbol{y}_0)$, an upper bound for $\frac{1}{1+\beta}\int p(\boldsymbol{z}|\boldsymbol{y}_{1:t})^{1+\beta}d\boldsymbol{z}$ must be found. The multivariate t-distribution can be integrated as

$$\frac{1}{1+\beta}\int \text{MVSt}_\nu(\boldsymbol{z}|\boldsymbol{\mu}, \boldsymbol{V})^{1+\beta}d\boldsymbol{z} = \frac{\Gamma((\nu+p)/2)^{\beta+1}\Gamma((\beta\nu+\beta p+\nu)/2)}{\Gamma(\nu/2)^{\beta+1}\Gamma((\beta\nu+\beta p+\nu+p)/2)}\frac{1}{(1+\beta)(\nu\pi)^{(\beta p)/2}|\boldsymbol{V}|^{\beta/2}} \tag{6}$$

$$= \frac{\Gamma((\nu+p)/2)^\beta\Gamma((\nu+p)/2)\Gamma((\beta\nu+\beta p+\nu)/2)}{\Gamma(\nu/2)^\beta\Gamma(\nu/2)\Gamma((\beta\nu+\beta p+\nu+p)/2)}\frac{1}{(1+\beta)(\pi\nu)^{(\beta p)/2}|\boldsymbol{V}|^{\beta/2}} \tag{7}$$

$$\leqslant \frac{\Gamma((\nu+p)/2)^\beta}{\Gamma(\nu/2)^\beta}\frac{1}{(1+\beta)(\pi\nu)^{(\beta p)/2}|\boldsymbol{V}|^{\beta/2}}. \tag{8}$$

The inequality is derived from the fact that $\frac{\Gamma\left(x+\frac{p}{2}\right)}{\Gamma(x)}$ is increasing in $x$ and as $\beta \geqslant 0$ and $\nu \geqslant 0$ then $(\beta\nu + \beta p + \nu)/2) \geqslant \nu/2$ which implies $\frac{\Gamma((\nu+p)/2)\Gamma((\beta\nu+\beta p+\nu)/2)}{\Gamma(\nu/2)\Gamma((\beta\nu+\beta p+\nu+p)/2)} \leqslant 1$.

Now employing the well-known result using Stirling's formula to bound the gamma function

$$(2\pi)^{1/2}x^{x-1/2}\exp(-x) \leqslant \Gamma(x) \leqslant (2\pi)^{1/2}x^{x-1/2}\exp(1/(12x) - x) \tag{9}$$

we can therefore rewrite the ratio of gamma functions leaving

$$\frac{1}{1+\beta}\int \mathrm{MVSt}-t_\nu(\boldsymbol{z}|\boldsymbol{\mu},\boldsymbol{V})^{1+\beta}d\boldsymbol{z} \leqslant \frac{\Gamma((\nu+p)/2)^\beta}{\Gamma(\nu/2)^\beta}\frac{1}{(1+\beta)(\pi\nu)^{(\beta p)/2}|\boldsymbol{V}|^{\beta/2}} \tag{10}$$

$$\leqslant \frac{\left(\sqrt{2\pi}((\nu+p)/2)^{(\nu+p-1)/2}\exp(-(\nu+p)/2+1/6(\nu+p))\right)^\beta}{\left(\sqrt{2\pi}(\nu/2)^{(\nu-1)/2}\exp(-\nu/2)\right)^\beta(1+\beta)(\pi\nu)^{(\beta p)/2}|\boldsymbol{V}|^{\beta/2}} \tag{11}$$

$$= ((1+\frac{p}{\nu})^{\beta(\nu+p-1)/2}\exp(\beta(1/(6(\nu+p))-p/2))\frac{1}{(1+\beta)(\pi)^{(\beta p)/2}|\boldsymbol{V}|^{\beta/2}}. \tag{12}$$

Clearly $\exp\left(\beta(1/(6(\nu+p))-p/2)\right)$ is decreasing in $\nu$ for all $p$ and to demonstrate when $((1+\frac{p}{\nu})^{\beta(\nu+p-1)/2}$ is decreasing in $\nu$ we examine its derivative

$$w = \left(1+\frac{p}{\nu}\right)^{\beta(\nu+p-1)/2} \tag{13}$$

$$= \exp\left((\beta(\nu+p-1)/2)\log\left(\left(1+\frac{p}{\nu}\right)\right)\right) \tag{14}$$

$$\frac{dw}{d\nu} = \frac{\beta}{2}\left(\log\left(1+\frac{p}{\nu}\right)-(\nu+p-1)\frac{\frac{p}{\nu^2}}{1+\frac{p}{\nu}}\right)\left(1+\frac{p}{\nu}\right)^{\beta(\nu+p-1)/2)}. \tag{15}$$

The sign of $\frac{dw}{d\nu}$ is dictated by $\left(\log\left(1+\frac{p}{\nu}\right)-(\nu+p-1)\frac{\frac{p}{\nu^2}}{1+\frac{p}{\nu}}\right)$, which can be demonstrated to be positive always if $p=1$ and negative always if $p>1$.

**Case 1**: when $p>1$, $\frac{1}{1+\beta}\int p(\boldsymbol{z}|\boldsymbol{y}_{1:t})^{1+\beta}d\boldsymbol{z}$ is decreasing in $\nu$ and thus we can upper bound it by substituting the smallest value of $\nu$. Here we bound $\nu$ above 1 in order to enforce that the mean of the predictive $t$-distribution exists. Under the KLD posterior it is clear that $a_0$ rises as more data is seen and while we do not have closed forms associated with the variational approximation to the $\beta$-D posterior we expect this to be the case here. As more data is seen the finite sampling uncertainty, represented by $\nu$ in the NIG case, should be decreasing. Therefore provided $a_0$ is set such that $2a_0>1$, then this lower bound should never be violated.

**Case 2**: when $p=1$, Stirling's formula has failed to provide a decreasing upper bound for $\frac{1}{1+\beta}\int p(\boldsymbol{z}|\boldsymbol{y}_{1:t})^{1+\beta}d\boldsymbol{z}$. However in the univariate case

$$\frac{1}{1+\beta}\int \mathrm{St}_\nu(\boldsymbol{z}|\boldsymbol{\mu},\boldsymbol{V})^{1+\beta}d\boldsymbol{z} \leqslant \frac{\Gamma((\nu+1)/2)^\beta}{\Gamma(\nu/2)^\beta}\frac{1}{(1+\beta)(\nu|\boldsymbol{V}|)^{(\beta)/2}\pi^{(\beta)/2}} \tag{16}$$

$$\leqslant \frac{1}{(1+\beta)|\boldsymbol{V}|^{(\beta)/2}\pi^{(\beta)/2}} \tag{17}$$

Where $p=1$ is substituted into the bound from equation (8) and the inequality comes from that fact that $\frac{\Gamma((x+1)/2)}{\Gamma(x/2)}\leqslant\sqrt{x}$. This bound conveniently does not depend on the degrees of freedom $\nu$ at all.

We can therefore lower bound $f_{m_t}^\beta(\boldsymbol{y}_{t+1}|\mathcal{F}_t)/f_{m_t}^\beta(\boldsymbol{y}_{t+1}|\boldsymbol{y}_0)$ as

$$\frac{f_{m_t}^\beta(\boldsymbol{y}_{t+1}|\mathcal{F}_t)}{f_{m_t}^\beta(\boldsymbol{y}_{t+1}|\boldsymbol{y}_0)} \geqslant \begin{cases} \exp\left\{-\frac{1}{\beta}\left(\frac{\Gamma(a_0+1/2)}{\Gamma(a_0)(2b_0\pi)^{1/2}|\boldsymbol{I}+\boldsymbol{X}\boldsymbol{\Sigma}_0\boldsymbol{X}^T|^{1/2}}\right)^\beta-\frac{1}{(1+\beta)|\boldsymbol{V}|^{(\beta)/2}\pi^{(\beta)/2}}+\right. \\ \left.\frac{\Gamma(a_0+1/2)^{\beta+1}\Gamma(\beta a_0+\beta/2+a_0)}{\Gamma(a_0)^{\beta+1}\Gamma(\beta a_0+\beta/2+a_0+1/2)}\frac{1}{(1+\beta)(2\pi b_0)^{(\beta)/2}|\boldsymbol{I}+\boldsymbol{X}\boldsymbol{\Sigma}_0\boldsymbol{X}^T|^{\beta/2}}\right\} \text{ if } p=1 \\ \exp\left\{-\frac{1}{\beta}\left(\frac{\Gamma(a_0+p/2)}{\Gamma(a_0)(2b_0\pi)^{p/2}|\boldsymbol{I}+\boldsymbol{X}\boldsymbol{\Sigma}_0\boldsymbol{X}^T|^{1/2}}\right)^\beta+\right. \\ \frac{\Gamma(a_0+p/2)^{\beta+1}\Gamma(\beta a_0+\beta p/2+a_0)}{\Gamma(a_0)^{\beta+1}\Gamma(\beta a_0+\beta p/2+a_0+p/2)}\frac{1}{(1+\beta)(2\pi b_0)^{(\beta p)/2}|\boldsymbol{I}+\boldsymbol{X}\boldsymbol{\Sigma}_0\boldsymbol{X}^T|^{\beta/2}}- \\ \left.((1+p)^{\beta p/2)}\exp(\beta(1/(6(1+p))-p/2))\frac{1}{(1+\beta)(\pi)^{(\beta p)/2}|\boldsymbol{V}|^{\beta/2}}\right\} \text{ if } p>1 \end{cases} \tag{18}$$

Now fixing $p,a_0,b_0,\mu_0,\Sigma_0$ and $|V|_{\min}$ which values of $\beta$ and $H(r_t,r_{t+1})$ would leave

$$\frac{1-H(r_t,r_{t+1}))}{H(r_t,r_{t+1})}\frac{f_{m_t}^\beta(\boldsymbol{y}_{t+1}|\mathcal{F}_t)}{f_{m_t}^\beta(\boldsymbol{y}_{t+1}|\boldsymbol{y}_0)} \geqslant 1? \tag{19}$$

We demonstrate this for $p > 1$ but it is straightforward to see that it extends to when $p = 1$. Rearranging the inequality in equation (18) gives us that (19) holds providing

$$\frac{1}{|\boldsymbol{V}|^{\beta/2}} \leqslant$$

$$\left(\frac{\Gamma(a_0 + p/2)^\beta}{\Gamma(a_0)^\beta (2b_0\pi)^{\beta p/2} |\boldsymbol{I} + \boldsymbol{X}\boldsymbol{\Sigma}_0\boldsymbol{X}^T|^{\beta/2}} \left(\frac{\Gamma(a_0 + p/2)\Gamma(\beta a_0 + \beta p/2 + a_0)}{\Gamma(a_0)\Gamma(\beta a_0 + \beta p/2 + a_0 + p/2)} \frac{1}{(1+\beta)} - \frac{1}{\beta}\right)\right. \tag{20}$$
$$\left. + \log\left(\frac{1 - H(r_t, r_{t+1})}{H(r_t, r_{t+1})}\right)\right) \frac{(1+\beta)(\pi)^{(\beta p)/2}}{((1 + \frac{p}{2a_0})^{\alpha(2a_0 + p - 1)/2}\exp(\beta(1/(6(2a_0 + p)) - p/2))}$$

We define the set defined by inequality (20) as $S\left(p, \beta, a_0, b_0, \mu_0, \boldsymbol{\Sigma}_0, |\boldsymbol{V}|_{\min}\right) = \{(\beta, H(r_t, r_{t+1})) : (\beta, H(r_t, r_{t+1}))$ satisfy (20) for $p, \beta, a_0, b_0, \mu_0, \boldsymbol{\Sigma}_0, |\boldsymbol{V}|_{\min}\}$. As a result we can see that for fixed of $a_0, b_0, \mu_0, \Sigma_0$ and $|\boldsymbol{V}| \geqslant |\boldsymbol{V}|_{\min}$ it is always possible to choose values of $\beta$ and $H(r_t, r_{t+1})$ such that this holds. To see this consider fixing $\beta$, the the upper bound is simply increasing in $\log\left(\frac{1 - H(r_t, r_{t+1})}{H(r_t, r_{t+1})}\right)$ which takes values in $\mathbb{R}$ and thus can be set large enough so that the inequality holds. $\square$

We note that in practice this result is likely to be stronger than is necessary. The observation that is most likely to generate a change-point will have 0 mass under the predictive associated with the current segment but also appears at the prior mode. While this was necessary to demonstrate this result for all situations this is incredibly unlikely to occur. The requirement for $|\boldsymbol{V}_{\min}|$ is a result of the beta-divergence loss function depending on $\int p(z|\boldsymbol{y}_{1:t})^{1+\beta} dz$. In the proof of this result we demonstrate that $f_{m_t}^\beta(\boldsymbol{y}_{t+1}|\mathcal{F}_t)/f_{m_t}^\beta(\boldsymbol{y}_{t+1}|\boldsymbol{y}_0)$ is increasing in $|\boldsymbol{V}|$ and as a result if it is allowed to get too small the inequality in equation (20) would not hold. This is an undesirable consequence of the beta-divergence score not being completely local, that is to say not solely depending on the predictive probability of the observation, thus the score under the prior can be quite a lot bigger than the score under the continuing run length independent of the observations seen and solely based on the predictive covariances.

## 3 Variational Bayes Approximation for $\beta$-divergence based General Bayesian Inference with the Bayesian Linear Model

For ease of notation, we use $\beta = \beta_{\mathrm{p}}$. We wish to approximate the posterior belief distribution $\pi_{\mathrm{DPD}}^\beta(\mu, \sigma^2|\boldsymbol{y})$ which for observations $\boldsymbol{y} = (\boldsymbol{y}_1, \boldsymbol{y}_2, \dots, \boldsymbol{y}_n)^T$ with $\boldsymbol{y}_i \in \mathbb{R}^d$, prior $\mathrm{NIG}^0(\mu, \sigma^2|a_0, b_0, \mu_0, \boldsymbol{\Sigma}_0)$, model likelihood $f$ and density power divergence (DPD) loss

$$\ell^\beta(\mu, \sigma^2|\boldsymbol{y}_i) = \frac{1}{\beta}f(\boldsymbol{y}_i|\mu, \sigma^2)^\beta - \frac{1}{1+\beta}\int_{\mathcal{Y}} f(\boldsymbol{y}_i|\mu, \sigma^2)^{1+\beta}d\boldsymbol{y} \tag{21}$$

is given by

$$\pi_{\mathrm{DPD}}^\beta(\mu, \sigma^2|\boldsymbol{y}) = \mathrm{NIG}^0(\mu, \sigma^2|a_0, b_0, \mu_0, \boldsymbol{\Sigma}_0) \cdot \exp\left\{-\sum_{i=1}^n \ell^\beta(\mu, \sigma^2|\boldsymbol{y}_i)\right\}. \tag{22}$$

In particular, we want to approximate it with a posterior $\mathrm{NIG}^{\mathrm{VB}}(\mu, \sigma^2|\widehat{a}_n, \widehat{b}_n, \widehat{\mu}_n, \widehat{\boldsymbol{\Sigma}}_n)$ via Variational Bayes. This can be done by minimizing the variational parameters in a Kullback-Leibler sense:

$$(a^*, b^*, \mu^*, \boldsymbol{\Sigma}^*) = \underset{(\widehat{a}_n, \widehat{b}_n, \widehat{\mu}_n, \widehat{\boldsymbol{\Sigma}}_n)}{\mathrm{argmin}} \left\{\mathrm{KL}\left(\pi_{\mathrm{DPD}}^\beta(\mu, \sigma^2|\boldsymbol{y}) \,\Big\|\, \mathrm{NIG}^{\mathrm{VB}}(\mu, \sigma^2|\widehat{a}_n, \widehat{b}_n, \widehat{\mu}_n, \widehat{\boldsymbol{\Sigma}}_n)\right)\right\}. \tag{23}$$

It is straightforward to rewrite the objective function for the above minimization as the Evidence Lower Bound (ELBO) induced by the DPD:

$$\text{ELBO}_{\text{DPD}} = -\underbrace{\text{KL}\left(\text{NIG}^{\text{VB}}(\boldsymbol{\mu}, \sigma^2 | \widehat{a}_n, \widehat{b}_n, \widehat{\boldsymbol{\mu}}_n, \widehat{\boldsymbol{\Sigma}}_n) \,\|\, \text{NIG}^0(\boldsymbol{\mu}, \sigma^2 | a_0, b_0, \boldsymbol{\mu}_0, \boldsymbol{\Sigma}_0)\right)}_{=Q_1}.$$

$$\underbrace{-\mathbb{E}_{\text{VB}}\left[-\sum_{i=1}^{n} \ell^\beta(\boldsymbol{\mu}, \sigma^2 | \boldsymbol{y}_i)\right]}_{=Q_2}. \tag{24}$$

In what follows, closed forms are derived for both $Q_1$ and $Q_2$. Some algebraic tricks will be applied multiple times, and will be referred to by the following symbols:

■ Completion of Squares, i.e. $\boldsymbol{u}'\boldsymbol{A}\boldsymbol{u} - 2\boldsymbol{v}'\boldsymbol{u} = (\boldsymbol{u} - \boldsymbol{A}^{-1}\boldsymbol{v})'\boldsymbol{A}(\boldsymbol{u} - \boldsymbol{A}^{-1}\boldsymbol{v}) - \boldsymbol{v}'\boldsymbol{A}^{-1}\boldsymbol{v}$;

$I(\mathcal{N})$ Integrating out the Normal density;

$I(\mathcal{IG})$ Integrating out the Inverse Gamma density.

Throughout, the dimensionality of $\boldsymbol{\mu}$ is $p \in \mathbb{N}$, $\mathcal{N}(\boldsymbol{\mu}|\boldsymbol{\mu}_0, \boldsymbol{\Sigma}_0)$ refers to a normal pdf in $\boldsymbol{\mu}$ with expectation $\boldsymbol{\mu}_0$, variance $\boldsymbol{\Sigma}_0$ and $\mathcal{IG}(\sigma^2|a, b)$ to an inverse gamma pdf in $\sigma^2$ with shape $a$ and scale $b$.

### 3.1 $Q_1$

First, note that by definition,

$$Q_1 = \int_{\boldsymbol{\mu},\sigma^2} \underbrace{\log\left(\frac{\text{NIG}^{\text{VB}}(\boldsymbol{\mu}, \sigma^2 | \widehat{a}_n, \widehat{b}_n, \widehat{\boldsymbol{\mu}}_n, \widehat{\boldsymbol{\Sigma}}_n)}{\text{NIG}^0(\boldsymbol{\mu}, \sigma^2 | a_0, b_0, \boldsymbol{\mu}_0, \boldsymbol{\Sigma}_0)}\right)}_{=Q_1^{\log}} \text{NIG}^{\text{VB}}(\boldsymbol{\mu}, \sigma^2 | \widehat{a}_n, \widehat{b}_n, \widehat{\boldsymbol{\mu}}_n, \widehat{\boldsymbol{\Sigma}}_n) d\boldsymbol{\mu} d\sigma^2. \tag{25}$$

Writing out $Q_1^{\log}$, one obtains a natural sum of three components $C_1, C_2(\sigma^2), C_3(\sigma^2, \boldsymbol{\mu})$:

$$Q_1^{\log} = \log\left(\frac{|\widehat{\boldsymbol{\Sigma}}_n|^{-0.5} \frac{\widehat{b}_n^{\widehat{a}_n}}{\Gamma(\widehat{a}_n)} (\sigma^2)^{-0.5p - \widehat{a}_n - 1} \exp\left\{-\frac{1}{2\sigma^2}\left[(\boldsymbol{\mu} - \widehat{\boldsymbol{\mu}}_n)'\widehat{\boldsymbol{\Sigma}}_n^{-1}(\boldsymbol{\mu} - \widehat{\boldsymbol{\mu}}_n) + 2\widehat{b}_n\right]\right\}}{|\boldsymbol{\Sigma}_0|^{-0.5} \frac{b_0^{a_0}}{\Gamma(a_0)} (\sigma^2)^{-0.5p - a_0 - 1} \exp\left\{-\frac{1}{2\sigma^2}\left[(\boldsymbol{\mu} - \boldsymbol{\mu}_0)'\boldsymbol{\Sigma}_0^{-1}(\boldsymbol{\mu} - \boldsymbol{\mu}_0) + 2b_0\right]\right\}}\right)$$

$$= \underbrace{\log\left(\frac{\widehat{b}_n^{\widehat{a}_n} \Gamma(a_0)}{b_0^{a_0} \Gamma(\widehat{a}_n)}\right) + 0.5 \log\left|\boldsymbol{\Sigma}_0 \widehat{\boldsymbol{\Sigma}}_n^{-1}\right|}_{=C_1} + \underbrace{(\widehat{a}_n - a_0)\log(\frac{1}{\sigma^2})}_{=C_2(\sigma^2)}$$

$$\underbrace{-\frac{1}{2\sigma^2}\left[(\boldsymbol{\mu} - \widehat{\boldsymbol{\mu}}_n)'\widehat{\boldsymbol{\Sigma}}_n^{-1}(\boldsymbol{\mu} - \widehat{\boldsymbol{\mu}}_n) - (\boldsymbol{\mu} - \boldsymbol{\mu}_0)'\boldsymbol{\Sigma}_0^{-1}(\boldsymbol{\mu} - \boldsymbol{\mu}_0) + 2(\widehat{b}_n - b_0)\right]}_{=C_3(\sigma^2,\boldsymbol{\mu})}. \tag{26}$$

Next, note that $C_3(\sigma^2, \boldsymbol{\mu})$ further decomposes into

$$\underbrace{\frac{1}{2\sigma^2}\left[\boldsymbol{\mu}'\left(\widehat{\boldsymbol{\Sigma}}_n^{-1} - \boldsymbol{\Sigma}_0^{-1}\right)\boldsymbol{\mu} - 2\boldsymbol{\mu}'\left(\widehat{\boldsymbol{\Sigma}}_n^{-1}\widehat{\boldsymbol{\mu}}_n - \boldsymbol{\Sigma}_0^{-1}\boldsymbol{\mu}_0\right)\right]}_{=C_4(\sigma^2,\boldsymbol{\mu})} + \underbrace{\frac{1}{\sigma^2}\underbrace{\left[\frac{1}{2}\widehat{\boldsymbol{\mu}}_n'\widehat{\boldsymbol{\Sigma}}_n^{-1}\widehat{\boldsymbol{\mu}}_n - \frac{1}{2}\boldsymbol{\mu}_0'\boldsymbol{\Sigma}_0^{-1}\boldsymbol{\mu}_0 + (\widehat{b}_n - b_0)\right]}_{=C_5}}_{=C_6(\sigma^2)}.$$

Notice that we have isolated the random variable $\boldsymbol{\mu}$ inside $C_4(\sigma^2, \boldsymbol{\mu})$ and that by definition, $\text{NIG}^{\text{VB}}(\boldsymbol{\mu}, \sigma^2 | \widehat{a}_n, \widehat{b}_n, \widehat{\boldsymbol{\mu}}_n, \widehat{\boldsymbol{\Sigma}}_n) = \mathcal{N}^{\text{VB}}(\boldsymbol{\mu}|\widehat{\boldsymbol{\mu}}_n, \sigma^2 \widehat{\boldsymbol{\Sigma}}_n) \cdot \mathcal{IG}^{\text{VB}}(\sigma^2 | \widehat{a}_n, \widehat{b}_n)$, meaning that

$$Q_1 = C_1 + \int_{\sigma^2} \left\{C_2(\sigma^2) - C_6(\sigma^2)\right\} \mathcal{IG}^{\text{VB}}(\sigma^2 | \widehat{a}_n, \widehat{b}_n) d\sigma^2$$

$$- \int_{\sigma^2} \left\{\underbrace{\int_{\boldsymbol{\mu}} C_4(\sigma^2, \boldsymbol{\mu}) \mathcal{N}^{\text{VB}}(\boldsymbol{\mu}|\widehat{\boldsymbol{\mu}}_n, \sigma^2 \widehat{\boldsymbol{\Sigma}}_n) d\boldsymbol{\mu}}_{=C_7(\sigma^2)}\right\} \mathcal{IG}^{\text{VB}}(\sigma^2 | \widehat{a}_n, \widehat{b}_n) d\sigma^2. \tag{27}$$

The inner integral is available in closed form, and naturally decomposes as

$$
\begin{aligned}
C_7(\sigma^2) &= \frac{1}{2\sigma^2}\mathbb{E}_{\mathcal{N}^{\text{VB}}}\left[\boldsymbol{\mu}'\left(\widehat{\boldsymbol{\Sigma}}_n^{-1}-\boldsymbol{\Sigma}_0^{-1}\right)\boldsymbol{\mu}\right]-\frac{2}{2\sigma^2}\mathbb{E}_{\mathcal{N}^{\text{VB}}}\left[\boldsymbol{\mu}'\right]\left(\widehat{\boldsymbol{\Sigma}}_n^{-1}\widehat{\boldsymbol{\mu}}_n-\boldsymbol{\Sigma}_0^{-1}\boldsymbol{\mu}_0\right)\\
&= \frac{1}{2\sigma^2}\mathbb{E}_{\mathcal{N}^{\text{VB}}}\left[\text{tr}\left(\left(\widehat{\boldsymbol{\Sigma}}_n^{-1}-\boldsymbol{\Sigma}_0^{-1}\right)\boldsymbol{\mu}\boldsymbol{\mu}'\right)\right]-\frac{1}{\sigma^2}\widehat{\boldsymbol{\mu}}_n'\left(\widehat{\boldsymbol{\Sigma}}_n^{-1}\widehat{\boldsymbol{\mu}}_n-\boldsymbol{\Sigma}_0^{-1}\boldsymbol{\mu}_0\right)\\
&= \frac{1}{2\sigma^2}\text{tr}\left(\left(\widehat{\boldsymbol{\Sigma}}_n^{-1}-\boldsymbol{\Sigma}_0^{-1}\right)\mathbb{E}_{\mathcal{N}^{\text{VB}}}\left[\boldsymbol{\mu}\boldsymbol{\mu}'\right]\right)-\frac{1}{\sigma^2}\widehat{\boldsymbol{\mu}}_n'\left(\widehat{\boldsymbol{\Sigma}}_n^{-1}\widehat{\boldsymbol{\mu}}_n-\boldsymbol{\Sigma}_0^{-1}\boldsymbol{\mu}_0\right)\\
&= \frac{1}{2\sigma^2}\text{tr}\left(\left(\widehat{\boldsymbol{\Sigma}}_n^{-1}-\boldsymbol{\Sigma}_0^{-1}\right)\left[\sigma^2\widehat{\boldsymbol{\Sigma}}_n-\widehat{\boldsymbol{\mu}}_n\widehat{\boldsymbol{\mu}}_n'\right]\right)-\frac{1}{\sigma^2}\widehat{\boldsymbol{\mu}}_n'\left(\widehat{\boldsymbol{\Sigma}}_n^{-1}\widehat{\boldsymbol{\mu}}_n-\boldsymbol{\Sigma}_0^{-1}\boldsymbol{\mu}_0\right)\\
&= \underbrace{\frac{1}{2}\text{tr}\left(I-\boldsymbol{\Sigma}_0^{-1}\widehat{\boldsymbol{\Sigma}}_n\right)}_{=C_8}-\frac{1}{\sigma^2}\underbrace{\left[\frac{1}{2}\widehat{\boldsymbol{\mu}}_n'(\widehat{\boldsymbol{\Sigma}}_n^{-1}-\boldsymbol{\Sigma}_0^{-1})\widehat{\boldsymbol{\mu}}_n-\widehat{\boldsymbol{\mu}}_n'\left(\widehat{\boldsymbol{\Sigma}}_n^{-1}\widehat{\boldsymbol{\mu}}_n-\boldsymbol{\Sigma}_0^{-1}\boldsymbol{\mu}_0\right)\right]}_{=C_9}.\quad (28)
\end{aligned}
$$

$$\underbrace{\phantom{\frac{1}{2}\text{tr}\left(I-\boldsymbol{\Sigma}_0^{-1}\widehat{\boldsymbol{\Sigma}}_n\right)-\frac{1}{\sigma^2}\left[\frac{1}{2}\widehat{\boldsymbol{\mu}}_n'(\widehat{\boldsymbol{\Sigma}}_n^{-1}-\boldsymbol{\Sigma}_0^{-1})\widehat{\boldsymbol{\mu}}_n-\widehat{\boldsymbol{\mu}}_n'\left(\widehat{\boldsymbol{\Sigma}}_n^{-1}\widehat{\boldsymbol{\mu}}_n-\boldsymbol{\Sigma}_0^{-1}\boldsymbol{\mu}_0\right)\right]}}_{=C_{10}(\sigma^2)}$$

We may now rewrite $Q_1$ so as to integrate out $\sigma^2$ next:

$$
Q_1 = C_1 - C_8 + \int_{\sigma^2}\left\{C_2(\sigma^2)-C_6(\sigma^2)-C_{10}(\sigma^2)\right\}\mathcal{IG}^{\text{VB}}(\sigma^2|\widehat{a}_n,\widehat{b}_n)d\sigma^2.
$$

Using the additivity of integrals, we consider its three components separately and then add them up together afterwards. For $C_2(\sigma^2)$, (I) apply a change of variable with $z=\frac{\sigma^2}{\widehat{b}_n}$ and then use (II) that $\frac{d}{dx}a^{-x}=-a^x\cdot\log(a)=a^x\cdot\log(a^{-1})$ together with Fubini's Theorem (III) to find that

$$
\begin{aligned}
C_{11} &= \int_{\sigma^2}C_2(\sigma^2)\mathcal{IG}^{\text{VB}}(\sigma^2|\widehat{a}_n,\widehat{b}_n)d\sigma^2\\
&= (\widehat{a}_n-a_0)\int_{\sigma^2}\log\left(\frac{1}{\sigma^2}\right)\frac{\widehat{b}_n^{\widehat{a}_n}}{\Gamma(\widehat{a}_n)}(\sigma^2)^{-\widehat{a}_n-1}\exp\left\{-\frac{\widehat{b}_n}{\sigma^2}\right\}d\sigma^2\\
&\overset{(I)}{=} (\widehat{a}_n-a_0)\int_z\log\left(\frac{1}{z\widehat{b}_n}\right)\frac{\widehat{b}_n^{\widehat{a}_n+1}}{\Gamma(\widehat{a}_n)}\left(z\widehat{b}_n\right)^{-\widehat{a}_n-1}\exp\left\{-\frac{1}{z}\right\}dz\\
&= (\widehat{a}_n-a_0)\frac{1}{\Gamma(\widehat{a}_n)}\int_z\left(-\log(z)-\log(\widehat{b}_n)\right)z^{-\widehat{a}_n-1}\exp\left\{-\frac{1}{z}\right\}dz\\
&\overset{I(\mathcal{IG})}{=} (\widehat{a}_n-a_0)\left[\frac{1}{\Gamma(\widehat{a}_n)}\int_z\left(-\log(z)\right)z^{-\widehat{a}_n-1}\exp\left\{-\frac{1}{z}\right\}dz-\log(\widehat{b}_n)\right]\\
&\overset{(II)}{=} (\widehat{a}_n-a_0)\left[\frac{1}{\Gamma(\widehat{a}_n)}\int_z\frac{d}{d\widehat{a}_n}\left\{z^{-\widehat{a}_n-1}\exp\left\{-\frac{1}{z}\right\}\right\}dz-\log(\widehat{b}_n)\right]\\
&\overset{(III)}{=} (\widehat{a}_n-a_0)\left[\frac{1}{\Gamma(\widehat{a}_n)}\frac{d}{d\widehat{a}_n}\underbrace{\left\{\int_z z^{-\widehat{a}_n-1}\exp\left\{-\frac{1}{z}\right\}dz\right\}}_{I(\mathcal{IG})\overset{}{=}\Gamma(\widehat{a}_n)}-\log(\widehat{b}_n)\right]\\
&= (\widehat{a}_n-a_0)\left(\frac{\Gamma'(\widehat{a}_n)}{\Gamma(\widehat{a}_n)}-\log(\widehat{b}_n)\right)\\
&= (\widehat{a}_n-a_0)\left(\Psi(\widehat{a}_n)-\log(\widehat{b}_n)\right),
\end{aligned}
$$

where $\Psi$ is the digamma function. For $C_6(\sigma^2)$, one obtains the closed form as

$$
\begin{aligned}
C_{12} &= \int_{\sigma^2} C_6(\sigma^2) \mathcal{IG}^{\text{VB}}(\sigma^2|\widehat{a}_n, \widehat{b}_n) d\sigma^2 \\
&= C_5 \int_{\sigma^2} \frac{\widehat{b}_n^{\widehat{a}_n}}{\Gamma(\widehat{a}_n)} (\sigma^2)^{-\widehat{a}_n - 1 - 1} \exp\left\{ -\frac{\widehat{b}_n}{\sigma^2} \right\} d\sigma^2 \\
&\overset{I(\mathcal{IG})}{=} C_5 \frac{\Gamma(\widehat{a}_n + 1)}{\widehat{b}_n \Gamma(\widehat{a}_n)}.
\end{aligned}
\tag{29}
$$

Using the exact same steps for $C_{10}(\sigma^2)$, one finds

$$
\begin{aligned}
C_{13} &= \int_{\sigma^2} C_{10}(\sigma^2) \mathcal{IG}^{\text{VB}}(\sigma^2|\widehat{a}_n, \widehat{b}_n) d\sigma^2 \\
&\overset{I(\mathcal{IG})}{=} C_9 \frac{\Gamma(\widehat{a}_n + 1)}{\widehat{b}_n \Gamma(\widehat{a}_n)},
\end{aligned}
\tag{30}
$$

finally yielding

$$
\begin{aligned}
Q_1 &= C_1 - C_8 + C_{11} - C_{12} - C_{13} \\
&= \log\left( \frac{\widehat{b}_n^{\widehat{a}_n} \Gamma(a_0)}{b_0^{a_0} \Gamma(\widehat{a}_n)} \right) + 0.5 \log\left| \boldsymbol{\Sigma}_0 \widehat{\boldsymbol{\Sigma}}_n^{-1} \right| - \frac{1}{2} \text{tr}\left( I - \boldsymbol{\Sigma}_0^{-1} \widehat{\boldsymbol{\Sigma}}_n \right) + (\widehat{a}_n - a_0)\left( \Psi(\widehat{a}_n) - \log(\widehat{b}_n) \right) \\
&\quad - \left[ \frac{1}{2} \widehat{\boldsymbol{\mu}}_n' \widehat{\boldsymbol{\Sigma}}_n^{-1} \widehat{\boldsymbol{\mu}}_n - \frac{1}{2} \boldsymbol{\mu}_0' \boldsymbol{\Sigma}_0^{-1} \boldsymbol{\mu}_0 + (\widehat{b}_n - b_0) \right] \cdot \frac{\Gamma(\widehat{a}_n + 1)}{\widehat{b}_n \Gamma(\widehat{a}_n)} \\
&\quad - \left[ \frac{1}{2} \widehat{\boldsymbol{\mu}}_n' (\widehat{\boldsymbol{\Sigma}}_n^{-1} - \boldsymbol{\Sigma}_0^{-1}) \widehat{\boldsymbol{\mu}}_n - \widehat{\boldsymbol{\mu}}_n' \left( \widehat{\boldsymbol{\Sigma}}_n^{-1} \widehat{\boldsymbol{\mu}}_n - \boldsymbol{\Sigma}_0^{-1} \boldsymbol{\mu}_0 \right) \right] \cdot \frac{\Gamma(\widehat{a}_n + 1)}{\widehat{b}_n \Gamma(\widehat{a}_n)} \\
&= \log\left( \frac{\widehat{b}_n^{\widehat{a}_n} \Gamma(a_0)}{b_0^{a_0} \Gamma(\widehat{a}_n)} \right) + 0.5 \log\left| \boldsymbol{\Sigma}_0 \widehat{\boldsymbol{\Sigma}}_n^{-1} \right| - \frac{1}{2} \text{tr}\left( I - \boldsymbol{\Sigma}_0^{-1} \widehat{\boldsymbol{\Sigma}}_n \right) + (\widehat{a}_n - a_0)\left( \Psi(\widehat{a}_n) - \log(\widehat{b}_n) \right) \\
&\quad + \frac{1}{2} \left[ (\boldsymbol{\mu}_0 - \widehat{\boldsymbol{\mu}}_n)' \boldsymbol{\Sigma}_0^{-1} (\boldsymbol{\mu}_0 - \widehat{\boldsymbol{\mu}}_n) + 2(b_0 - \widehat{b}_n) \right] \cdot \frac{\Gamma(\widehat{a}_n + 1)}{\widehat{b}_n \Gamma(\widehat{a}_n)}
\end{aligned}
\tag{31}
$$

$$
\tag{32}
$$

### 3.2 $Q_2$

Noting that one can write $Q_2$ as

$$
\begin{aligned}
&= E_{\text{VB}}\left[ \sum_{i=1}^{n} \ell^\beta(\boldsymbol{\mu}, \sigma^2 | \boldsymbol{y}_i) \right] \\
&= \int_{\boldsymbol{\mu}, \sigma^2} \left\{ \sum_{i=1}^{n} \left[ \frac{1}{\beta} f(\boldsymbol{y}_i | \boldsymbol{\mu}, \sigma^2)^\beta - \frac{1}{1+\beta} \int_{\mathcal{Y}} f(\boldsymbol{y}|\boldsymbol{\mu}, \sigma^2)^{1+\beta} d\boldsymbol{y} \right] \text{NIG}^{\text{VB}}(\boldsymbol{\mu}, \sigma^2|\widehat{a}_n, \widehat{b}_n, \widehat{\boldsymbol{\mu}}_n, \widehat{\boldsymbol{\Sigma}}_n) \right\} d\sigma^2 d\boldsymbol{\mu} \\
&= \sum_{i=1}^{n} \left[ \int_{\boldsymbol{\mu}, \sigma^2} \left\{ \frac{1}{\beta} f(\boldsymbol{y}_i | \boldsymbol{\mu}, \sigma^2)^\beta - \frac{1}{1+\beta} \int_{\mathcal{Y}} f(\boldsymbol{y}|\boldsymbol{\mu}, \sigma^2)^{1+\beta} d\boldsymbol{y} \right\} \text{NIG}^{\text{VB}}(\boldsymbol{\mu}, \sigma^2|\widehat{a}_n, \widehat{b}_n, \widehat{\boldsymbol{\mu}}_n, \widehat{\boldsymbol{\Sigma}}_n) d\sigma^2 d\boldsymbol{\mu} \right]
\end{aligned}
\tag{33}
$$

The last equation implies that it is sufficient to concern ourselves with the integral for a single term. To this end, observe that the likelihood for a single observation $\boldsymbol{y}_i$ with regressor matrix $\boldsymbol{X}_i$ is given by

$$
f(\boldsymbol{y}_i | \boldsymbol{\mu}, \sigma^2) = \mathcal{N}(\boldsymbol{y}_i | \boldsymbol{X}_i' \boldsymbol{\mu}, \sigma^2 I_d),
\tag{34}
$$

where $I_d$ is the identity matrix of dimension $d$. Looking at the likelihood terms inside $\ell^\beta$, the $\beta$-exponentiated likelihood term can be rewritten as

$$\frac{1}{\beta}f(\boldsymbol{y}_i|\boldsymbol{\mu},\sigma^2)^\beta = \underbrace{\frac{1}{\beta}(2\pi)^{-0.5d\beta}(\sigma^2)^{-0.5d\beta}}_{=D_1(\sigma^2)} \cdot \exp\left\{-\frac{\beta}{2\sigma^2}\left[(\boldsymbol{y}_i-\boldsymbol{X}_i'\boldsymbol{\mu})'(\boldsymbol{y}_i-\boldsymbol{X}_i'\boldsymbol{\mu})\right]\right\}$$

$$= D_1(\sigma^2)\cdot\exp\left\{-\frac{\beta}{2\sigma^2}\left[\boldsymbol{y}_i'\boldsymbol{y}_i + \boldsymbol{\mu}'\underbrace{(\boldsymbol{X}_i\boldsymbol{X}_i')}_{=\check{\boldsymbol{\Sigma}}_i^{-1}}\boldsymbol{\mu} - 2(\boldsymbol{y}_i'\boldsymbol{X}_i)\boldsymbol{\mu}\right]\right\}$$

$$\stackrel{\blacksquare}{=} D_1(\sigma^2)\cdot\exp\left\{-\frac{1}{2\sigma^2}\left[\beta(\boldsymbol{\mu}-\underbrace{\check{\boldsymbol{\Sigma}}_i(\boldsymbol{X}_i'\boldsymbol{y}_i)}_{=\check{\boldsymbol{\mu}}_i})'\check{\boldsymbol{\Sigma}}_i^{-1}(\boldsymbol{\mu}-\check{\boldsymbol{\mu}}_i) + \beta\underbrace{\left[\boldsymbol{y}_i'\boldsymbol{y}_i - (\boldsymbol{y}_i\boldsymbol{X}_i')\check{\boldsymbol{\Sigma}}_i(\boldsymbol{X}_i\boldsymbol{y}_i')\right]}_{=D_{2,i}}\right]\right\}$$

$$= D_1(\sigma^2)\cdot\exp\left\{-\frac{1}{2\sigma^2}\left[\underbrace{\beta(\boldsymbol{\mu}-\check{\boldsymbol{\mu}}_i)'\check{\boldsymbol{\Sigma}}_i^{-1}(\boldsymbol{\mu}-\check{\boldsymbol{\mu}}_i)}_{=D_{3,i}(\boldsymbol{\mu})} + D_{2,i}\right]\right\}$$

$$= D_1(\sigma^2)\cdot\exp\left\{-\frac{1}{2\sigma^2}\left[D_{3,i}(\boldsymbol{\mu}) + D_{2,i}\right]\right\}, \tag{35}$$

while the integral is available in closed form as

$$\frac{1}{1+\beta}\int_{\mathcal{Y}}f(\boldsymbol{y}|\boldsymbol{\mu},\sigma^2)^{1+\beta}d\boldsymbol{y} \stackrel{I(\mathcal{N})}{=} (\sigma^2)^{-0.5p\beta}\underbrace{(2\pi)^{-0.5d\beta}(1+\beta)^{-0.5d-1}}_{=D_4} \tag{36}$$

One can see a neat separation between terms involving $\sigma^2$ and terms involving $\boldsymbol{\mu}$ again, allowing us to rewrite the integral in equation (33) such as to exploit the conditional structure of the normal inverse-gamma distribution in Eqs. (36), (35). Looking at integrating out $\sigma^2$ from (35) first, note that

$$\begin{aligned} L_1 &= \int_{\sigma^2}\left\{\frac{1}{1+\beta}\int_{\mathcal{Y}}f(\boldsymbol{y}|\boldsymbol{\mu},\sigma^2)^{1+\beta}d\boldsymbol{y}\right\}\mathcal{IG}^{\text{VB}}(\sigma^2|\hat{a}_n,\hat{b}_n)d\sigma^2\\ &= D_4\int_{\sigma^2}(\sigma^2)^{-0.5d\beta-\hat{a}_n-1}\frac{\hat{b}_n^{\hat{a}_n}}{\Gamma(\hat{a}_n)}\exp\left\{-\frac{\hat{b}_n}{\sigma^2}\right\}d\sigma^2\\ &\stackrel{I(\mathcal{N})}{=} D_4\cdot\frac{\Gamma(\hat{a}_n+0.5d\beta)}{\Gamma(\hat{a}_n)\hat{b}_n^{0.5d\beta}}. \end{aligned} \tag{37}$$

For the $\beta$-exponentiated likelihood term, one finds that

$$\begin{aligned} L_{2,i} &= \int_{\sigma^2,\boldsymbol{\mu}}\frac{1}{\beta}f(\boldsymbol{y}_i|\boldsymbol{\mu},\sigma^2)^\beta\text{NIG}^{\text{VB}}(\boldsymbol{\mu},\sigma^2|\hat{a}_n,\hat{b}_n,\hat{\boldsymbol{\mu}}_n,\hat{\boldsymbol{\Sigma}}_n)d\sigma^2d\boldsymbol{\mu}\\ &= \int_{\sigma^2}D_1(\sigma^2)\cdot\exp\left\{-\frac{1}{2\sigma^2}D_{2,i}\right\}\underbrace{\left[\int_{\boldsymbol{\mu}}\exp\left\{-\frac{1}{2\sigma^2}D_{3,i}(\boldsymbol{\mu})\right\}\mathcal{N}^{\text{VB}}(\boldsymbol{\mu}|\hat{\boldsymbol{\mu}}_n,\sigma^2\hat{\boldsymbol{\Sigma}}_n)d\boldsymbol{\mu}\right]}_{=D_{5,i}(\sigma^2)}\mathcal{IG}^{\text{VB}}(\sigma^2|\hat{a}_n,\hat{b}_n)d\sigma^2, \end{aligned}$$

where we have again exploited the conditional structure of our assumed posterior. The inner integral equals

$$D_{5,i}(\sigma^2) = (2\pi)^{-0.5p}\left|\sigma^2\hat{\boldsymbol{\Sigma}}_n\right|^{-0.5}\underbrace{\int_{\boldsymbol{\mu}}\exp\left\{-\frac{1}{2\sigma^2}\underbrace{\left[D_{3,i}(\boldsymbol{\mu}) + (\boldsymbol{\mu}-\hat{\boldsymbol{\mu}}_n)'\hat{\boldsymbol{\Sigma}}_n^{-1}(\boldsymbol{\mu}-\hat{\boldsymbol{\mu}}_n)\right]}_{=D_{6,i}(\boldsymbol{\mu})}\right\}}_{=D_{7,i}(\sigma^2)} \tag{38}$$

indicating that the closed form for the integral is available if one rewrites it as a normal density. To this end, one can use completion of squares to rewrite

$$
\begin{aligned}
D_{6,i}(\boldsymbol{\mu}) &= \beta(\boldsymbol{\mu} - \boldsymbol{\check{\mu}}_i)'\boldsymbol{\check{\Sigma}}_i^{-1}(\boldsymbol{\mu} - \boldsymbol{\check{\mu}}_i) + (\boldsymbol{\mu} - \boldsymbol{\hat{\mu}}_n)'\boldsymbol{\hat{\Sigma}}_n^{-1}(\boldsymbol{\mu} - \boldsymbol{\hat{\mu}}_n) \\
&= \boldsymbol{\mu}'\underbrace{\left[\boldsymbol{\hat{\Sigma}}_n^{-1} + \beta\boldsymbol{\check{\Sigma}}_i^{-1}\right]}_{=\boldsymbol{\tilde{\Sigma}}_i^{-1}}\boldsymbol{\mu} - 2\left[\hat{b}_n'\boldsymbol{\hat{\Sigma}}_n^{-1} + \beta\boldsymbol{\check{\mu}}_i'\boldsymbol{\check{\Sigma}}_i^{-1}\right]\boldsymbol{\mu} + \left[\boldsymbol{\hat{\mu}}_n'\boldsymbol{\hat{\Sigma}}_n^{-1}\boldsymbol{\hat{\mu}}_n + \beta\boldsymbol{\check{\mu}}_i'\boldsymbol{\check{\Sigma}}_i^{-1}\boldsymbol{\check{\mu}}_i\right] \\
&\stackrel{\blacksquare}{=} \left(\boldsymbol{\mu} - \underbrace{\boldsymbol{\tilde{\Sigma}}_i\left[\boldsymbol{\hat{\Sigma}}_n^{-1}\hat{b}_n + \beta\boldsymbol{\check{\Sigma}}_i^{-1}\boldsymbol{\check{\mu}}_i\right]}_{=\boldsymbol{\tilde{\mu}}_i}\right)'\boldsymbol{\tilde{\Sigma}}_i^{-1}(\boldsymbol{\mu} - \boldsymbol{\tilde{\mu}}_i)+ \\
&\quad \underbrace{\boldsymbol{\hat{\mu}}_n'\boldsymbol{\hat{\Sigma}}_n^{-1}\boldsymbol{\hat{\mu}}_n + \beta\boldsymbol{\check{\mu}}_i'\boldsymbol{\check{\Sigma}}_i^{-1}\boldsymbol{\check{\mu}}_i - \left(\boldsymbol{\hat{\Sigma}}_n^{-1}\boldsymbol{\hat{\mu}}_n + \beta\boldsymbol{\check{\Sigma}}_i^{-1}\boldsymbol{\check{\mu}}_i\right)'\boldsymbol{\tilde{\Sigma}}_i\left(\boldsymbol{\hat{\Sigma}}_n^{-1}\boldsymbol{\hat{\mu}}_n + \beta\boldsymbol{\check{\Sigma}}_i^{-1}\boldsymbol{\check{\mu}}_i\right)}_{=D_{8,i}} \\
&= (\boldsymbol{\mu} - \boldsymbol{\tilde{\mu}}_i)'\boldsymbol{\tilde{\Sigma}}_i^{-1}(\boldsymbol{\mu} - \boldsymbol{\tilde{\mu}}_i) + D_{8,i},
\end{aligned}
\tag{39}
$$

which then allows integrating out $\boldsymbol{\mu}$ from $D_{7,i}(\sigma^2)$ using the density of a normal random variable:

$$
\begin{aligned}
D_{7,i}(\sigma^2) &= \exp\left\{-\frac{1}{2\sigma^2}D_{8,i}\right\}\int_{\boldsymbol{\mu}}\exp\left\{-\frac{1}{2\sigma^2}(\boldsymbol{\mu} - \boldsymbol{\tilde{\mu}}_i)'\boldsymbol{\tilde{\Sigma}}_i^{-1}(\boldsymbol{\mu} - \boldsymbol{\tilde{\mu}}_i)\right\}d\boldsymbol{\mu} \\
&\stackrel{I(\mathcal{N})}{=} \exp\left\{-\frac{1}{2\sigma^2}D_{8,i}\right\}(2\pi)^{0.5p}|\sigma^2\boldsymbol{\tilde{\Sigma}}_i|^{0.5},
\end{aligned}
\tag{40}
$$

so we can finally rewrite the entire integral as

$$
D_{5,i}(\sigma^2) = |\boldsymbol{\hat{\Sigma}}_n^{-1}\boldsymbol{\tilde{\Sigma}}_i|^{0.5}\exp\left\{-\frac{1}{2\sigma^2}D_{8,i}\right\},
\tag{41}
$$

which enables rewriting $L_{2,i}$ as

$$
\begin{aligned}
L_{2,i} &= \underbrace{\frac{1}{\beta}(2\pi)^{-0.5d\beta}|\boldsymbol{\hat{\Sigma}}_n^{-1}\boldsymbol{\tilde{\Sigma}}_i|^{0.5}}_{=D_{9,i}}\int_{\sigma^2}(\sigma^2)^{-0.5d\beta}\exp\left\{-\frac{1}{\sigma^2}\cdot\frac{1}{2}\left[D_{2,i} + D_{8,i}\right]\right\}\mathcal{IG}^{\mathrm{VB}}(\sigma^2|\hat{a}_n, \hat{b}_n)d\sigma^2 \\
&\stackrel{I(\mathcal{IG})}{=} \frac{D_{9,i}\cdot\Gamma(\hat{a}_n + 0.5d\beta)\cdot\hat{b}_n^{\hat{a}_n}}{\Gamma(\hat{a}_n)\cdot\left[\hat{b}_n + 0.5(D_{2,i} + D_{7,i})\right]^{(\hat{a}_n + 0.5d\beta)}},
\end{aligned}
\tag{42}
$$

finally implying that one may write

$$
\begin{aligned}
Q_2 &= \sum_{i=1}^{n}L_{2,i} - nL_1 \\
&= \sum_{i=1}^{n}\left\{\frac{D_{9,i}\cdot\Gamma(\hat{a}_n + 0.5d\beta)\cdot\hat{b}_n^{\hat{a}_n}}{\Gamma(\hat{a}_n)\cdot\left[\hat{b}_n + 0.5(D_{2,i} + D_{8,i})\right]^{(\hat{a}_n + 0.5d\beta)}}\right\} - nD_4\cdot\frac{\Gamma(\hat{a}_n + 0.5d\beta)}{\Gamma(\hat{a}_n)\hat{b}_n^{0.5d\beta}} \\
&= \sum_{i=1}^{n}\left\{\frac{\frac{1}{\beta}(2\pi)^{-0.5d\beta}\left|\boldsymbol{\hat{\Sigma}}_n^{-1}\left[\boldsymbol{\hat{\Sigma}}_n^{-1} + \beta(\boldsymbol{X}_i\boldsymbol{X}_i)\right]^{-1}\right|^{0.5}\cdot\Gamma(\hat{a}_n + 0.5d\beta)\cdot\hat{b}_n^{\hat{a}_n}}{\Gamma(\hat{a}_n)\cdot\left[\hat{b}_n + 0.5\left(D_{2,i} + D_{8,i}\right)\right]^{(\hat{a}_n + 0.5d\beta)}}\right\} \\
&\quad - n\cdot\frac{(2\pi)^{-0.5d\beta}(1 + \beta)^{-0.5d-1}\cdot\Gamma(\hat{a}_n + 0.5d\beta)}{\Gamma(\hat{a}_n)\hat{b}_n^{0.5d\beta}}.
\end{aligned}
$$

We further simplify this expression by observing that

$$
\begin{aligned}
D_{2,i} + D_{8,i} &= \beta\left[\boldsymbol{y}_i'\boldsymbol{y}_i - (\boldsymbol{y}_i\boldsymbol{X}_i')\breve{\boldsymbol{\Sigma}}_i(\boldsymbol{X}_i\boldsymbol{y}_i')\right] + \widehat{\boldsymbol{\mu}}_n'\widehat{\boldsymbol{\Sigma}}_n^{-1}\widehat{\boldsymbol{\mu}}_n + \beta\breve{\boldsymbol{\mu}}_i'\breve{\boldsymbol{\Sigma}}_i^{-1}\breve{\boldsymbol{\mu}}_i \\
&\quad - \left(\widehat{\boldsymbol{\Sigma}}_n^{-1}\widehat{\boldsymbol{\mu}}_n + \beta\breve{\boldsymbol{\Sigma}}_i^{-1}\breve{\boldsymbol{\mu}}_i\right)' \tilde{\boldsymbol{\Sigma}}_i \left(\widehat{\boldsymbol{\Sigma}}_n^{-1}\widehat{\boldsymbol{\mu}}_n + \beta\breve{\boldsymbol{\Sigma}}_i^{-1}\breve{\boldsymbol{\mu}}_i\right) \\
&= \beta\boldsymbol{y}_i'\boldsymbol{y}_i - \beta(\boldsymbol{y}_i\boldsymbol{X}_i')\breve{\boldsymbol{\Sigma}}_i(\boldsymbol{X}_i\boldsymbol{y}_i') + \widehat{\boldsymbol{\mu}}_n'\widehat{\boldsymbol{\Sigma}}_n^{-1}\widehat{\boldsymbol{\mu}}_n + \beta(\boldsymbol{y}_i\boldsymbol{X}_i')\breve{\boldsymbol{\Sigma}}_i(\boldsymbol{X}_i\boldsymbol{y}_i') \\
&\quad - \left(\widehat{\boldsymbol{\Sigma}}_n^{-1}\widehat{\boldsymbol{\mu}}_n + \beta(\boldsymbol{X}_i'\boldsymbol{y}_i)\right)' \tilde{\boldsymbol{\Sigma}}_i \left(\widehat{\boldsymbol{\Sigma}}_n^{-1}\widehat{\boldsymbol{\mu}}_n + \beta(\boldsymbol{X}_i'\boldsymbol{y}_i)\right) \\
&= \beta\boldsymbol{y}_i'\boldsymbol{y}_i + \widehat{\boldsymbol{\mu}}_n'\widehat{\boldsymbol{\Sigma}}_n^{-1}\widehat{\boldsymbol{\mu}}_n - \left(\widehat{\boldsymbol{\Sigma}}_n^{-1}\widehat{\boldsymbol{\mu}}_n + \beta(\boldsymbol{X}_i'\boldsymbol{y}_i)\right)'\left[\widehat{\boldsymbol{\Sigma}}_n^{-1} + \beta(\boldsymbol{X}_i\boldsymbol{X}_i)\right]^{-1}\left(\widehat{\boldsymbol{\Sigma}}_n^{-1}\widehat{\boldsymbol{\mu}}_n + \beta(\boldsymbol{X}_i'\boldsymbol{y}_i)\right),
\end{aligned}
$$

leaving us with

$$
Q_2 = \frac{\Gamma(\widehat{a}_n + 0.5d\beta)\cdot\widehat{b}_n^{\widehat{a}_n}\cdot|\widehat{\boldsymbol{\Sigma}}_n^{-1}|^{0.5}}{\beta(2\pi)^{0.5d\beta}\Gamma(\widehat{a}_n)}\times
$$

$$
\sum_{i=1}^{n}\left\{\frac{\left|\left[\widehat{\boldsymbol{\Sigma}}_n^{-1} + \beta(\boldsymbol{X}_i\boldsymbol{X}_i)\right]^{-1}\right|^{0.5}}{\left[\widehat{b}_n + 0.5\left(\beta\boldsymbol{y}_i'\boldsymbol{y}_i + \widehat{\boldsymbol{\mu}}_n'\widehat{\boldsymbol{\Sigma}}_n^{-1}\widehat{\boldsymbol{\mu}}_n - \left(\widehat{\boldsymbol{\Sigma}}_n^{-1}\widehat{\boldsymbol{\mu}}_n + \beta(\boldsymbol{X}_i'\boldsymbol{y}_i)\right)'\left[\widehat{\boldsymbol{\Sigma}}_n^{-1} + \beta(\boldsymbol{X}_i\boldsymbol{X}_i)\right]^{-1}\left(\widehat{\boldsymbol{\Sigma}}_n^{-1}\widehat{\boldsymbol{\mu}}_n + \beta(\boldsymbol{X}_i'\boldsymbol{y}_i)\right)\right)\right]^{(\widehat{a}_n+0.5d\beta)}}\right\}
$$

$$
-n\cdot\frac{\Gamma(\widehat{a}_n + 0.5d\beta)}{\Gamma(\widehat{a}_n)\widehat{b}_n^{0.5d\beta}(2\pi)^{0.5d\beta}(1 + \beta)^{0.5d+1}}.
$$

### 3.3 ELBO

Putting together the results of the two previous sections, the ELBO is obtained as

$$
\begin{aligned}
ELBO &= -Q_1 + Q_2 \\
&= -\log\left(\frac{\widehat{b}_n^{\widehat{a}_n}\Gamma(a_0)}{b_0^{a_0}\Gamma(\widehat{a}_n)}\right) - 0.5\log\left|\boldsymbol{\Sigma}_0\widehat{\boldsymbol{\Sigma}}_n^{-1}\right| + \frac{1}{2}\mathrm{tr}\left(I - \boldsymbol{\Sigma}_0^{-1}\widehat{\boldsymbol{\Sigma}}_n\right) - (\widehat{a}_n - a_0)\left(\Psi(\widehat{a}_n) - \log(\widehat{b}_n)\right) \\
&\quad - \left[\frac{1}{2}(\boldsymbol{\mu}_0 - \widehat{\boldsymbol{\mu}}_n)'\boldsymbol{\Sigma}_0^{-1}(\boldsymbol{\mu}_0 - \widehat{\boldsymbol{\mu}}_n) + (b_0 - \widehat{b}_n)\right]\cdot\frac{\Gamma(\widehat{a}_n + 1)}{\widehat{b}_n\Gamma(\widehat{a}_n)} \\
&\quad + \frac{\Gamma(\widehat{a}_n + 0.5d\beta)\cdot\widehat{b}_n^{\widehat{a}_n}\cdot|\widehat{\boldsymbol{\Sigma}}_n^{-1}|^{0.5}}{\beta(2\pi)^{0.5d\beta}\Gamma(\widehat{a}_n)}\times
\end{aligned}
$$

$$
\sum_{i=1}^{n}\left\{\frac{\left|\widehat{\boldsymbol{\Sigma}}_n^{-1} + \beta\left(\boldsymbol{X}_i'\boldsymbol{X}_i\right)\right|^{-0.5}}{\left[\widehat{b}_n + 0.5\left(\beta\boldsymbol{y}_i'\boldsymbol{y}_i + \widehat{\boldsymbol{\mu}}_n'\widehat{\boldsymbol{\Sigma}}_n^{-1}\widehat{\boldsymbol{\mu}}_n - \left(\widehat{\boldsymbol{\Sigma}}_n^{-1}\widehat{\boldsymbol{\mu}}_n + \beta(\boldsymbol{X}_i'\boldsymbol{y}_i)\right)'\left[\widehat{\boldsymbol{\Sigma}}_n^{-1} + \beta(\boldsymbol{X}_i'\boldsymbol{X}_i)\right]^{-1}\left(\widehat{\boldsymbol{\Sigma}}_n^{-1}\widehat{\boldsymbol{\mu}}_n + \beta(\boldsymbol{X}_i'\boldsymbol{y}_i)\right)\right)\right]^{(\widehat{a}_n+0.5d\beta)}}\right\}
$$

$$
-n\cdot\frac{\Gamma(\widehat{a}_n + 0.5d\beta)}{\Gamma(\widehat{a}_n)\widehat{b}_n^{0.5d\beta}(2\pi)^{0.5d\beta}(1 + \beta)^{0.5d+1}} \tag{43}
$$

### 3.4 Differentiation

In this section, we take derivatives of the ELBO with respect to each variational parameter, i.e. $\widehat{a}_n, \widehat{b}_n, \widehat{\boldsymbol{\mu}}_n, \widehat{\boldsymbol{\Sigma}}_n$. Observing that differentiation with respect to $\widehat{\boldsymbol{\Sigma}}_n^{-1}$ is easier than with respect to $\widehat{\boldsymbol{\Sigma}}_n$, parametrize the optimization using the Cholesky decomposition, i.e. $\widehat{\boldsymbol{\Sigma}}_n^{-1} = \boldsymbol{L}\boldsymbol{L}'$, where $L$ is a lower triangular matrix and is unique if $\widehat{\boldsymbol{\Sigma}}_n$ (equivalently $\widehat{\boldsymbol{\Sigma}}_n^{-1}$) is positive definite[1].

### 3.4.1 Derivative with respect to $L$

In what follows, we differentiate the ELBO term by term with respect to the $p(p-1)\frac{1}{2}$ entries in the lower triangular part of $\boldsymbol{L}$ that can be summarized in the vector $\mathrm{vech}\,(\boldsymbol{L})$. To this end, define

$$E_1 = -0.5 \log\left|\boldsymbol{\Sigma}_0 \widehat{\boldsymbol{\Sigma}}_n^{-1}\right| + \frac{1}{2}\mathrm{tr}\left(I - \boldsymbol{\Sigma}_0^{-1}\widehat{\boldsymbol{\Sigma}}_n\right) \tag{44}$$

$$E_2 = \underbrace{\frac{\Gamma(\widehat{a}_n + 0.5 d\beta)\cdot \widehat{b}_n^{\widehat{a}_n}}{\beta(2\pi)^{0.5 d\beta}\Gamma(\widehat{a}_n)}}_{=F}\,|\widehat{\boldsymbol{\Sigma}}_n^{-1}|^{0.5} \tag{45}$$

$$E_{3,i} = \left|\widehat{\boldsymbol{\Sigma}}_n^{-1} + \beta\left(\boldsymbol{X}_i'\boldsymbol{X}_i\right)\right|^{-0.5} \tag{46}$$

$$E_4 = \widehat{\boldsymbol{\mu}}_n'\widehat{\boldsymbol{\Sigma}}_n^{-1}\widehat{\boldsymbol{\mu}}_n \tag{47}$$

$$E_{5,i} = -\widehat{\boldsymbol{\mu}}_n'\widehat{\boldsymbol{\Sigma}}_n^{-1}\left[\widehat{\boldsymbol{\Sigma}}_n^{-1} + \beta\left(\boldsymbol{X}_i'\boldsymbol{X}_i\right)\right]^{-1}\widehat{\boldsymbol{\Sigma}}_n^{-1}\widehat{\boldsymbol{\mu}}_n \tag{48}$$

$$E_{6,i} = -\beta^2(\boldsymbol{y}_i'\boldsymbol{X}_i)\left[\widehat{\boldsymbol{\Sigma}}_n^{-1} + \beta\left(\boldsymbol{X}_i'\boldsymbol{X}_i\right)\right]^{-1}(\boldsymbol{X}_i'\boldsymbol{y}_i), \tag{49}$$

$$E_{7,i} = -2\beta\widehat{\boldsymbol{\mu}}_n'\widehat{\boldsymbol{\Sigma}}_n^{-1}\left[\widehat{\boldsymbol{\Sigma}}_n^{-1} + \beta\left(\boldsymbol{X}_i'\boldsymbol{X}_i\right)\right]^{-1}(\boldsymbol{X}_i'\boldsymbol{y}_i). \tag{50}$$

Obtaining the derivative of the ELBO is equivalent to obtaining the derivatives of these newly defined quantities, as

$$\frac{\partial}{\partial\mathrm{vech}\,(\boldsymbol{L})}\{ELBO\} = \frac{\partial}{\partial\mathrm{vech}\,(\boldsymbol{L})}\{E_1\}$$

$$+ \frac{\partial}{\partial\mathrm{vech}\,(\boldsymbol{L})}\{E_2\}\cdot\sum_{i=1}^{n}\left\{\frac{E_{3,i}}{\left[\widehat{b}_n + 0.5\left(\beta\boldsymbol{y}_i'\boldsymbol{y}_i + E_4 + E_{5,i} + E_{6,i} + E_{7,i}\right)\right]^{\widehat{a}_n + 0.5 d\beta}}\right\}$$

$$+ E_2\cdot\sum_{i=1}^{n}\left\{\frac{\frac{\partial}{\partial\mathrm{vech}(\boldsymbol{L})}\{E_{3,i}\}}{\left[\widehat{b}_n + 0.5\left(\beta\boldsymbol{y}_i'\boldsymbol{y}_i + E_4 + E_{5,i} + E_{6,i} + E_{7,i}\right)\right]^{\widehat{a}_n + 0.5 d\beta}}\right\}$$

$$+ E_2\cdot\sum_{i=1}^{n}\left\{E_{3,i}\cdot\frac{\partial}{\partial\mathrm{vech}\,(\boldsymbol{L})}\left\{\left[\widehat{b}_n + 0.5\left(\beta\boldsymbol{y}_i'\boldsymbol{y}_i + E_4 + E_{5,i} + E_{6,i} + E_{7,i}\right)\right]^{-\widehat{a}_n - 0.5 d\beta}\right\}\right\} \tag{51}$$

where the chain and sum rule imply that

$$\frac{\partial}{\partial\mathrm{vech}\,(\boldsymbol{L})}\left\{\left[\widehat{b}_n + 0.5\left(\beta\boldsymbol{y}_i'\boldsymbol{y}_i + E_4 + E_{5,i} + E_{6,i} + E_{7,i}\right)\right]^{-\widehat{a}_n - 0.5 d\beta}\right\}$$

$$= (-\widehat{a}_n - 0.5 d\beta)\cdot\left[\widehat{b}_n + 0.5\left(\beta\boldsymbol{y}_i'\boldsymbol{y}_i + E_4 + E_{5,i} + E_{6,i} + E_{7,i}\right)\right]^{-\widehat{a}_n - 0.5 d\beta - 1}\times$$

$$0.5\cdot\frac{\partial}{\partial\mathrm{vech}\,(\boldsymbol{L})}\{E_4 + E_{5,i} + E_{6,i} + E_{7,i}\}, \tag{52}$$

For convenience and simplified notation when taking the derivatives of the expressions defined in Eqs (44) - (50), also define the following matrices:

$$\boldsymbol{R} = \left[\widehat{\boldsymbol{\Sigma}}_n^{-1} + \beta\left(\boldsymbol{X}_i'\boldsymbol{X}_i\right)\right] \tag{53}$$

$$\boldsymbol{B} = \widehat{\boldsymbol{\mu}}_n\widehat{\boldsymbol{\mu}}_n'. \tag{54}$$

Define also the following symbols to mark operations used in the derivations:

- $\partial$ Switching from differential notation $\partial\boldsymbol{L}$ to the derivative $\frac{\partial}{\partial\mathrm{vech}(\boldsymbol{L})}$;

- tr Properties of the trace like invariance under cyclic permutations, invariance under the transpose, additivity, and the fact that for $c$ a scalar, $\mathrm{tr}(c) = c$.

Note than when the differential operator $\partial$ is used, its scope is always limited to the next term only, unless brackets are used. Hence $\partial \boldsymbol{LL}'$ uses the differential only with respect to $\boldsymbol{L}$, while $\partial \left(\boldsymbol{LL}'\right)^{-1}$ uses it with respect to the entire expression $\left(\boldsymbol{LL}'\right)^{-1}$. It is also worth noting that $\partial \boldsymbol{L}' = \left(\partial \boldsymbol{L}\right)'$ for any matrix $\boldsymbol{L}$, as this will be used in conjunction with the transpose invariance of the trace throughout to simplify terms. Using these symbols and the differential notation, proceed by noting the following:

$$\partial(\boldsymbol{LL}') = \partial \boldsymbol{R} = \partial \boldsymbol{LL}' + \boldsymbol{L}\partial \boldsymbol{L}' = \partial \boldsymbol{LL}' + \boldsymbol{L}\partial \boldsymbol{L}' = \partial \boldsymbol{LL}' + \left(\partial \boldsymbol{LL}'\right)' \tag{55}$$

$$\partial(\boldsymbol{LL}')^{-1} = -(\boldsymbol{LL}')^{-1}\left[\partial(\boldsymbol{LL}')\right](\boldsymbol{LL}')^{-1} \tag{56}$$

$$\partial|\boldsymbol{LL}'| = |\boldsymbol{LL}'| \cdot \mathrm{tr}\left((\boldsymbol{LL}')^{-1}\left[\partial \boldsymbol{LL}' + \left(\partial \boldsymbol{LL}'\right)'\right]\right)$$

$$\overset{\mathrm{tr}}{=} 2|\boldsymbol{LL}'| \cdot \mathrm{tr}\left(\boldsymbol{L}'(\boldsymbol{LL}')^{-1}\partial \boldsymbol{L}\right) \tag{57}$$

$$\partial \boldsymbol{R}^{-1} = -\boldsymbol{R}^{-1}\partial \boldsymbol{R}\boldsymbol{R}^{-1} = -\boldsymbol{R}^{-1}\partial \boldsymbol{LL}'\boldsymbol{R}^{-1} - \left[\boldsymbol{R}^{-1}\partial \boldsymbol{LL}'\boldsymbol{R}^{-1}\right]'. \tag{58}$$

With this in place, the derivatives of the quantities defined before are obtained as

$$\partial E_1 = -\frac{1}{2}\partial\left\{\log|\boldsymbol{\Sigma}_0| + \log|\boldsymbol{LL}'|\right\} - \frac{1}{2}\partial\left\{\mathrm{tr}\left(\boldsymbol{\Sigma}_0^{-1}\widehat{\boldsymbol{\Sigma}}_n\right)\right\}$$

$$= -\frac{1}{2} \cdot |\boldsymbol{LL}'|^{-1} \cdot \partial|\boldsymbol{LL}'| - \frac{1}{2}\mathrm{tr}\left(\boldsymbol{\Sigma}_0^{-1}\partial\left(\boldsymbol{LL}'\right)^{-1}\right)$$

$$= -\frac{1}{2}\mathrm{tr}\left(\boldsymbol{L}'(\boldsymbol{LL}')^{-1}\partial \boldsymbol{L}\right) + \frac{1}{2}\mathrm{tr}\left(\boldsymbol{\Sigma}_0^{-1}(\boldsymbol{LL}')^{-1}\left[\partial(\boldsymbol{LL}')\right](\boldsymbol{LL}')^{-1}\right)$$

$$\overset{\mathrm{tr}}{=} -\mathrm{tr}\left(\boldsymbol{L}'(\boldsymbol{LL}')^{-1}\partial \boldsymbol{L}\right) + \mathrm{tr}\left(\boldsymbol{L}'\left(\boldsymbol{LL}'\right)^{-1}\boldsymbol{\Sigma}_0^{-1}\left(\boldsymbol{LL}'\right)^{-1}\partial \boldsymbol{L}\right)$$

$$\partial E_2 = F \cdot \partial|\boldsymbol{LL}'|^{0.5}$$

$$= \frac{F}{2} \cdot |\boldsymbol{LL}'|^{-0.5} \cdot 2|\boldsymbol{LL}'| \cdot \mathrm{tr}\left(\boldsymbol{L}'(\boldsymbol{LL}')^{-1}\partial \boldsymbol{L}\right)$$

$$= F \cdot |\boldsymbol{LL}'|^{0.5}\mathrm{tr}\left(\boldsymbol{L}'(\boldsymbol{LL}')^{-1}\partial \boldsymbol{L}\right) \tag{59}$$

$$\partial E_{3,i} = \partial \boldsymbol{R}^{-0.5} = -\frac{1}{2}|\boldsymbol{R}|^{-1.5}\partial \boldsymbol{R}$$

$$= -\frac{1}{2}|\boldsymbol{R}|^{-0.5}\mathrm{tr}\left(\boldsymbol{R}^{-1}\partial\left(\boldsymbol{LL}'\right)\right)$$

$$\overset{\mathrm{tr}}{=} -|\boldsymbol{R}|^{-0.5}\mathrm{tr}\left(\boldsymbol{L}'\boldsymbol{R}^{-1}\partial \boldsymbol{L}\right) \tag{60}$$

$$\partial E_4 \overset{\mathrm{tr}}{=} \mathrm{tr}\left(\widehat{\boldsymbol{\mu}}_n'\partial(\boldsymbol{LL}')\widehat{\boldsymbol{\mu}}_n\right)$$

$$= \mathrm{tr}\left(\widehat{\boldsymbol{\mu}}_n'\left[\partial \boldsymbol{LL}' + \boldsymbol{L}\partial \boldsymbol{L}'\right]\widehat{\boldsymbol{\mu}}_n\right)$$

$$\overset{\mathrm{tr}}{=} 2 \cdot \mathrm{tr}\left(\boldsymbol{L}'\boldsymbol{B}\partial \boldsymbol{L}\right) \tag{61}$$

$$\partial E_{5,i} \overset{\mathrm{tr}}{=} -\mathrm{tr}\left(\widehat{\boldsymbol{\mu}}_n'\partial\left(\boldsymbol{LL}'\right)\boldsymbol{R}^{-1}\left(\boldsymbol{LL}'\right)\widehat{\boldsymbol{\mu}}_n\right) - \mathrm{tr}\left(\widehat{\boldsymbol{\mu}}_n'\left(\boldsymbol{LL}'\right)\partial \boldsymbol{R}^{-1}\left(\boldsymbol{LL}'\right)\widehat{\boldsymbol{\mu}}_n\right)$$

$$\quad -\mathrm{tr}\left(\widehat{\boldsymbol{\mu}}_n'\left(\boldsymbol{LL}'\right)\boldsymbol{R}^{-1}\partial\left(\boldsymbol{LL}'\right)\widehat{\boldsymbol{\mu}}_n\right)$$

$$\overset{\mathrm{tr}}{=} -2 \cdot \mathrm{tr}\left(\widehat{\boldsymbol{\mu}}_n'\partial \boldsymbol{LL}'\boldsymbol{R}^{-1}\left(\boldsymbol{LL}'\right)\widehat{\boldsymbol{\mu}}_n\right) + 2 \cdot \mathrm{tr}\left(\widehat{\boldsymbol{\mu}}_n'\left(\boldsymbol{LL}'\right)\boldsymbol{R}^{-1}\partial \boldsymbol{LL}'\boldsymbol{R}^{-1}\left(\boldsymbol{LL}'\right)\widehat{\boldsymbol{\mu}}_n\right)$$

$$\quad -2 \cdot \mathrm{tr}\left(\widehat{\boldsymbol{\mu}}_n'\left(\boldsymbol{LL}'\right)\boldsymbol{R}^{-1}\partial \boldsymbol{LL}'\widehat{\boldsymbol{\mu}}_n\right)$$

$$\overset{\mathrm{tr}}{=} -2 \cdot \mathrm{tr}\left(\boldsymbol{L}'\boldsymbol{R}^{-1}\left(\boldsymbol{LL}'\right)\boldsymbol{B}\partial \boldsymbol{L}\right) + 2 \cdot \mathrm{tr}\left(\boldsymbol{L}'\boldsymbol{R}^{-1}\left(\boldsymbol{LL}'\right)\boldsymbol{B}\left(\boldsymbol{LL}'\right)\boldsymbol{R}^{-1}\partial \boldsymbol{L}\right)$$

$$\quad -2 \cdot \mathrm{tr}\left(\boldsymbol{L}'\boldsymbol{B}\left(\boldsymbol{LL}'\right)\boldsymbol{R}^{-1}\partial \boldsymbol{L}\right)$$

$$\partial E_{6,i} \overset{\mathrm{tr}}{=} -\beta^2\mathrm{tr}\left(\left(\boldsymbol{y}_i'\boldsymbol{X}_i\right)\partial \boldsymbol{R}^{-1}\left(\boldsymbol{X}_i'\boldsymbol{y}_i\right)\right)$$

$$\overset{\mathrm{tr}}{=} 2\beta^2\mathrm{tr}\left(\left(\boldsymbol{y}_i'\boldsymbol{X}_i\right)\boldsymbol{R}^{-1}\partial \boldsymbol{LL}'\boldsymbol{R}^{-1}\left(\boldsymbol{X}_i'\boldsymbol{y}_i\right)\right)$$

$$\overset{\mathrm{tr}}{=} 2\beta^2\mathrm{tr}\left(\boldsymbol{L}'\boldsymbol{R}^{-1}\left(\boldsymbol{X}_i'\boldsymbol{y}_i\right)\left(\boldsymbol{y}_i'\boldsymbol{X}_i\right)\boldsymbol{R}^{-1}\partial \boldsymbol{L}\right) \tag{62}$$

$$\partial E_{7,i} \overset{\mathrm{tr}}{=} -2\beta \cdot \left[\mathrm{tr}\left(\widehat{\boldsymbol{\mu}}_n'\partial\left(\boldsymbol{LL}'\right)\boldsymbol{R}^{-1}\left(\boldsymbol{X}_i\boldsymbol{y}_i\right)\right) + \mathrm{tr}\left(\widehat{\boldsymbol{\mu}}_n'\left(\boldsymbol{LL}'\right)\partial \boldsymbol{R}^{-1}\left(\boldsymbol{X}_i\boldsymbol{y}_i\right)\right)\right]$$

$$\overset{\mathrm{tr}}{=} -2\beta \cdot \left[\mathrm{tr}\left(\widehat{\boldsymbol{\mu}}_n'\partial \boldsymbol{LL}'\boldsymbol{R}^{-1}\left(\boldsymbol{X}_i\boldsymbol{y}_i\right)\right) + \mathrm{tr}\left(\widehat{\boldsymbol{\mu}}_n'\boldsymbol{L}\partial \boldsymbol{L}'\boldsymbol{R}^{-1}\left(\boldsymbol{X}_i\boldsymbol{y}_i\right)\right)\right.$$

$$\left. -\mathrm{tr}\left(\widehat{\boldsymbol{\mu}}_n'\left(\boldsymbol{LL}'\right)\boldsymbol{R}^{-1}\partial \boldsymbol{LL}'\boldsymbol{R}^{-1}\left(\boldsymbol{X}_i\boldsymbol{y}_i\right)\right) - \mathrm{tr}\left(\widehat{\boldsymbol{\mu}}_n'\left(\boldsymbol{LL}'\right)\boldsymbol{R}^{-1}\boldsymbol{L}\partial \boldsymbol{L}'\boldsymbol{R}^{-1}\left(\boldsymbol{X}_i\boldsymbol{y}_i\right)\right)\right]$$

$$\overset{\mathrm{tr}}{=} -2\beta \cdot \left[\mathrm{tr}\left(\boldsymbol{L}'\boldsymbol{R}^{-1}\left(\boldsymbol{X}_i\boldsymbol{y}_i\right)\widehat{\boldsymbol{\mu}}_n'\partial \boldsymbol{L}\right) + \mathrm{tr}\left(\boldsymbol{L}'\widehat{\boldsymbol{\mu}}_n\left(\boldsymbol{y}_i'\boldsymbol{X}_i\right)\boldsymbol{R}^{-1}\partial \boldsymbol{L}\right)\right.$$

$$-\operatorname{tr}\left(\boldsymbol{L}'\boldsymbol{R}^{-1}\left(\boldsymbol{X}_i\boldsymbol{y}_i\right)\widehat{\boldsymbol{\mu}}_n'\left(\boldsymbol{L}\boldsymbol{L}'\right)\boldsymbol{R}^{-1}\partial\boldsymbol{L}\right)-\operatorname{tr}\left(\boldsymbol{L}'\boldsymbol{R}^{-1}\left(\boldsymbol{L}\boldsymbol{L}'\right)\widehat{\boldsymbol{\mu}}_n\left(\boldsymbol{y}_i'\boldsymbol{X}_i\right)\boldsymbol{R}^{-1}\partial\boldsymbol{L}\right)\Bigg] \quad (63)$$

This can now be converted into derivative notation and simplified. To this end, first note that for any $p\times\partial$ matrix $A$ which is not a function of $\boldsymbol{L}$,

$$\operatorname{tr}(Ad\boldsymbol{L})=\sum_{i=1}^{p}A_{1i}dL_{i1}+\sum_{i=2}^{p}A_{2i}dL_{i2}+\cdots=\sum_{j=1}^{p}\left\{\sum_{i=j}^{p}A_{ji}dL_{ji}\right\}, \quad (64)$$

implying in particular that

$$\frac{\partial}{\partial\operatorname{vech}\left(\boldsymbol{L}\right)}\operatorname{tr}(Ad\boldsymbol{L})=\operatorname{vech}(A^T) \quad (65)$$

and use this by defining $\operatorname{vech}^T(A)=\operatorname{vech}(A^T)$ to note that

$$\begin{aligned}
\frac{\partial}{\partial\operatorname{vech}\left(\boldsymbol{L}\right)}E_1 &\stackrel{\partial}{=} \operatorname{vech}^T\left(-\left[\boldsymbol{L}'(\boldsymbol{L}\boldsymbol{L}')^{-1}\right]+\left[\boldsymbol{L}'\left(\boldsymbol{L}\boldsymbol{L}'\right)^{-1}\boldsymbol{\Sigma}_0^{-1}\left(\boldsymbol{L}\boldsymbol{L}'\right)^{-1}\right]\right)\\
&=\operatorname{vech}^T\left(\boldsymbol{L}'\left(\boldsymbol{L}\boldsymbol{L}'\right)^{-1}\left[\boldsymbol{\Sigma}_0^{-1}\left(\boldsymbol{L}\boldsymbol{L}'\right)^{-1}-I_p\right]\right)\\
&=\operatorname{vech}^T\left(\boldsymbol{L}^{-1}\left[\boldsymbol{\Sigma}_0^{-1}\left(\boldsymbol{L}\boldsymbol{L}'\right)^{-1}-I_p\right]\right)\\
&=\operatorname{vech}\left(\left[\left(\boldsymbol{L}\boldsymbol{L}'\right)^{-1}\boldsymbol{\Sigma}_0^{-1}-I_p\right]\boldsymbol{L}^{-T}\right)
\end{aligned} \quad (66)$$

$$\begin{aligned}
\frac{\partial}{\partial\operatorname{vech}\left(\boldsymbol{L}\right)}E_2 &\stackrel{\partial}{=} F\cdot|\boldsymbol{L}\boldsymbol{L}'|^{0.5}\cdot\operatorname{vech}^T\left(\boldsymbol{L}'(\boldsymbol{L}\boldsymbol{L}')^{-1}\right)\\
&=F\cdot|\boldsymbol{L}\boldsymbol{L}'|^{0.5}\cdot\operatorname{vech}\left(\boldsymbol{L}^{-T}\right)
\end{aligned} \quad (67)$$

$$\frac{\partial}{\partial\operatorname{vech}\left(\boldsymbol{L}\right)}E_{3,i} \stackrel{\partial}{=} -|\boldsymbol{R}|^{-0.5}\cdot\operatorname{vech}\left(\boldsymbol{R}^{-1}\boldsymbol{L}\right) \quad (68)$$

$$\frac{\partial}{\partial\operatorname{vech}\left(\boldsymbol{L}\right)}E_4 \stackrel{\partial}{=} 2\cdot\operatorname{vech}\left(\boldsymbol{B}\boldsymbol{L}\right) \quad (69)$$

$$\begin{aligned}
\frac{\partial}{\partial\operatorname{vech}\left(\boldsymbol{L}\right)}E_{5,i} &\stackrel{\partial}{=} \operatorname{vech}^T\left(-2\boldsymbol{L}'\boldsymbol{R}^{-1}\left(\boldsymbol{L}\boldsymbol{L}'\right)\boldsymbol{B}+2\boldsymbol{L}'\boldsymbol{R}^{-1}\left(\boldsymbol{L}\boldsymbol{L}'\right)\boldsymbol{B}\left(\boldsymbol{L}\boldsymbol{L}'\right)\boldsymbol{R}^{-1}-2\boldsymbol{L}'\boldsymbol{B}\left(\boldsymbol{L}\boldsymbol{L}'\right)\boldsymbol{R}^{-1}\right)\\
&=2\cdot\operatorname{vech}^T\left(\left[\boldsymbol{L}'\boldsymbol{R}^{-1}\left(\boldsymbol{L}\boldsymbol{L}'\right)\boldsymbol{B}\left[\left(\boldsymbol{L}\boldsymbol{L}'\right)\boldsymbol{R}^{-1}-I_p\right]\right]-\left[\boldsymbol{L}'\boldsymbol{B}\left(\boldsymbol{L}\boldsymbol{L}'\right)\boldsymbol{R}^{-1}\right]\right)\\
&=2\cdot\operatorname{vech}^T\left(\boldsymbol{L}'\left[\boldsymbol{R}^{-1}\left(\boldsymbol{L}\boldsymbol{L}'\right)\boldsymbol{B}\left[\left(\boldsymbol{L}\boldsymbol{L}'\right)\boldsymbol{R}^{-1}-I_p\right]-\boldsymbol{B}\left(\boldsymbol{L}\boldsymbol{L}'\right)\boldsymbol{R}^{-1}\right]\right)\\
&=2\cdot\operatorname{vech}\left(\left[\left[\boldsymbol{R}^{-1}\left(\boldsymbol{L}\boldsymbol{L}'\right)-I_p\right]\boldsymbol{B}\left(\boldsymbol{L}\boldsymbol{L}'\right)\boldsymbol{R}^{-1}-\boldsymbol{R}^{-1}\left(\boldsymbol{L}\boldsymbol{L}'\right)\boldsymbol{B}\right]\boldsymbol{L}\right)
\end{aligned} \quad (70)$$

$$\frac{\partial}{\partial\operatorname{vech}\left(\boldsymbol{L}\right)}E_{6,i} \stackrel{\partial}{=} 2\beta^2\cdot\operatorname{vech}\left(\boldsymbol{R}^{-1}\left(\boldsymbol{X}_i'\boldsymbol{y}_i\right)\left(\boldsymbol{y}_i'\boldsymbol{X}_i\right)\boldsymbol{R}^{-1}\boldsymbol{L}\right) \quad (71)$$

$$\begin{aligned}
\frac{\partial}{\partial\operatorname{vech}\left(\boldsymbol{L}\right)}E_{7,i} &\stackrel{\partial}{=} -2\beta\cdot\operatorname{vech}^T\Bigg(\boldsymbol{L}'\boldsymbol{R}^{-1}\left(\boldsymbol{X}_i\boldsymbol{y}_i\right)\widehat{\boldsymbol{\mu}}_n'+\boldsymbol{L}'\widehat{\boldsymbol{\mu}}_n\left(\boldsymbol{y}_i'\boldsymbol{X}_i\right)\boldsymbol{R}^{-1}\\
&\qquad\qquad -\boldsymbol{L}'\boldsymbol{R}^{-1}\left(\boldsymbol{X}_i\boldsymbol{y}_i\right)\widehat{\boldsymbol{\mu}}_n'\left(\boldsymbol{L}\boldsymbol{L}'\right)\boldsymbol{R}^{-1}-\boldsymbol{L}'\boldsymbol{R}^{-1}\left(\boldsymbol{L}\boldsymbol{L}'\right)\widehat{\boldsymbol{\mu}}_n\left(\boldsymbol{y}_i'\boldsymbol{X}_i\right)\boldsymbol{R}^{-1}\Bigg)\\
&=-2\beta\cdot\operatorname{vech}^T\Bigg(\boldsymbol{L}'\boldsymbol{R}^{-1}\left(\boldsymbol{X}_i'\boldsymbol{y}_i\right)\widehat{\boldsymbol{\mu}}_n'\left[I_p-\left(\boldsymbol{L}\boldsymbol{L}'\right)\boldsymbol{R}^{-1}\right]\\
&\qquad\qquad +\left[I_p-\boldsymbol{L}'\boldsymbol{R}^{-1}\boldsymbol{L}\right]\boldsymbol{L}'\widehat{\boldsymbol{\mu}}_n\left(\boldsymbol{y}_i'\boldsymbol{X}_i\right)\boldsymbol{R}^{-1}\Bigg)\\
&=-2\beta\cdot\operatorname{vech}\Bigg(\left[I_p-\boldsymbol{R}^{-1}\left(\boldsymbol{L}\boldsymbol{L}'\right)\right]\widehat{\boldsymbol{\mu}}_n\left(\boldsymbol{y}_i'\boldsymbol{X}_i\right)\boldsymbol{R}^{-1}\boldsymbol{L}\\
&\qquad\qquad +\boldsymbol{R}^{-1}\left(\boldsymbol{X}_i'\boldsymbol{y}_i\right)\widehat{\boldsymbol{\mu}}_n'\boldsymbol{L}\left[I_p-\boldsymbol{L}'\boldsymbol{R}^{-1}\boldsymbol{L}\right]\Bigg)
\end{aligned} \quad (72)$$

### 3.4.2 Derivative with respect to $\widehat{\boldsymbol{\mu}}_n$

Differentiating with respect to $\widehat{\boldsymbol{\mu}}_n$ is trivial. One proceeds by the same logic as in the section before, to which end one additionally needs to define the new term

$$E_8 = -\frac{1}{2}\left[(\boldsymbol{\mu}_0-\widehat{\boldsymbol{\mu}}_n)'\boldsymbol{\Sigma}_0^{-1}(\boldsymbol{\mu}_0-\widehat{\boldsymbol{\mu}}_n)+2(b_0-\widehat{b}_n)\right]\cdot\frac{\Gamma(\widehat{a}_n+1)}{\widehat{b}_n\Gamma(\widehat{a}_n)}, \quad (73)$$

allowing us to write

$$\frac{\partial}{\partial \widehat{\boldsymbol{\mu}}_n} \{ELBO\} = \frac{\partial}{\partial \widehat{\boldsymbol{\mu}}_n} \{E_8\} +$$

$$E_2 \cdot \sum_{i=1}^{n} \left\{ E_{3,i} \cdot \frac{\partial}{\partial \widehat{\boldsymbol{\mu}}_n} \left\{ \left[ \widehat{b}_n + 0.5 \left( \beta \boldsymbol{y}_i' \boldsymbol{y}_i + E_4 + E_{5,i} + E_{6,i} + E_{7,i} \right) \right]^{-\widehat{a}_n - 0.5 d\beta} \right\} \right\} \tag{74}$$

where

$$\frac{\partial}{\partial \widehat{\boldsymbol{\mu}}_n} \left\{ \left[ \widehat{b}_n + 0.5 \left( \beta \boldsymbol{y}_i' \boldsymbol{y}_i + E_4 + E_{5,i} + E_{6,i} + E_{7,i} \right) \right]^{-\widehat{a}_n - 0.5 d\beta} \right\}$$

$$= (-\widehat{a}_n - 0.5 d\beta) \cdot \left[ \widehat{b}_n + 0.5 \left( \beta \boldsymbol{y}_i' \boldsymbol{y}_i + E_4 + E_{5,i} + E_{6,i} + E_{7,i} \right) \right]^{-\widehat{a}_n - 0.5 d\beta - 1} \times$$

$$0.5 \cdot \frac{\partial}{\partial \widehat{\boldsymbol{\mu}}_n} \{ E_4 + E_{5,i} + E_{7,i} \}, \tag{75}$$

so that obtaining the derivative is achieved by finding $\frac{\partial}{\partial \widehat{\boldsymbol{\mu}}_n} E_4$, $\frac{\partial}{\partial \widehat{\boldsymbol{\mu}}_n} E_{5,i}$, $\frac{\partial}{\partial \widehat{\boldsymbol{\mu}}_n} E_{7,i}$ and $\frac{\partial}{\partial \widehat{\boldsymbol{\mu}}_n} E_8$:

$$\frac{\partial}{\partial \widehat{\boldsymbol{\mu}}_n} E_4 = 2 \cdot \widehat{\boldsymbol{\mu}}_n' \widehat{\boldsymbol{\Sigma}}_n^{-1} \tag{76}$$

$$\frac{\partial}{\partial \widehat{\boldsymbol{\mu}}_n} E_{5,i} = -2 \cdot \widehat{\boldsymbol{\mu}}_n' \widehat{\boldsymbol{\Sigma}}_n^{-1} \boldsymbol{R}^{-1} \widehat{\boldsymbol{\Sigma}}_n^{-1} \tag{77}$$

$$\frac{\partial}{\partial \widehat{\boldsymbol{\mu}}_n} E_{7,i} = -2\beta \cdot \left( \boldsymbol{y}_i' \boldsymbol{X}_i \right) \boldsymbol{R}^{-1} \widehat{\boldsymbol{\Sigma}}_n^{-1} \tag{78}$$

$$\frac{\partial}{\partial \widehat{\boldsymbol{\mu}}_n} E_8 = -\frac{1}{2} \cdot \frac{\Gamma(\widehat{a}_n + 1)}{\widehat{b}_n \Gamma(\widehat{a}_n)} \left[ \frac{\partial}{\partial \widehat{\boldsymbol{\mu}}_n} \left( \widehat{\boldsymbol{\mu}}_n' \boldsymbol{\Sigma}_0^{-1} \widehat{\boldsymbol{\mu}}_n \right) - 2 \frac{\partial}{\partial \widehat{\boldsymbol{\mu}}_n} \left( \widehat{\boldsymbol{\mu}}_n \boldsymbol{\Sigma}_0^{-1} \boldsymbol{\mu}_0 \right) \right]$$

$$= -\frac{1}{2} \cdot \frac{\Gamma(\widehat{a}_n + 1)}{\widehat{b}_n \Gamma(\widehat{a}_n)} \left[ 2 \widehat{\boldsymbol{\mu}}_n' \boldsymbol{\Sigma}_0^{-1} - 2 \boldsymbol{\mu}_0' \boldsymbol{\Sigma}_0^{-1} \right]$$

$$= \frac{\Gamma(\widehat{a}_n + 1)}{\widehat{b}_n \Gamma(\widehat{a}_n)} \left[ (\boldsymbol{\mu}_0 - \widehat{\boldsymbol{\mu}}_n)' \boldsymbol{\Sigma}_0^{-1} \right] \tag{79}$$

### 3.4.3 Derivative with respect to $\widehat{a}_n$

We proceed again by the same logic. Define

$$E_9 = -\log \left( \frac{\widehat{b}_n^{\widehat{a}_n} \Gamma(a_0)}{b_0^{a_0} \Gamma(\widehat{a}_n)} \right) \tag{80}$$

$$E_{10} = -(\widehat{a}_n - a_0) \left( \Psi(\widehat{a}_n) - \log(\widehat{b}_n) \right) \tag{81}$$

$$E_{11} = -n \cdot \frac{\Gamma(\widehat{a}_n + 0.5 d\beta)}{\Gamma(\widehat{a}_n) \widehat{b}_n^{0.5 d\beta} (2\pi)^{0.5 d\beta} (1 + \beta)^{0.5 d + 1}}. \tag{82}$$

Use this to write

$$\frac{\partial}{\partial \widehat{a}_n} \{ELBO\} = \frac{\partial}{\partial \widehat{a}_n} \{E_8\} + \frac{\partial}{\partial \widehat{a}_n} \{E_9\} + \frac{\partial}{\partial \widehat{a}_n} \{E_{10}\} + \frac{\partial}{\partial \widehat{a}_n} \{E_{11}\} +$$

$$+ \frac{\partial}{\partial \widehat{a}_n} \{E_2\} \sum_{i=1}^{n} \left\{ \frac{E_{3,i}}{\left[ \widehat{b}_n + 0.5 \left( \beta \boldsymbol{y}_i' \boldsymbol{y}_i + E_4 + E_{5,i} + E_{6,i} + E_{7,i} \right) \right]^{\widehat{a}_n + 0.5 d\beta}} \right\}$$

$$+ E_2 \cdot \sum_{i=1}^{n} \left\{ E_{3,i} \cdot \frac{\partial}{\partial \widehat{a}_n} \left\{ \left[ \widehat{b}_n + 0.5 \left( \beta \boldsymbol{y}_i' \boldsymbol{y}_i + E_4 + E_{5,i} + E_{6,i} + E_{7,i} \right) \right]^{-\widehat{a}_n - 0.5 d\beta} \right\} \right\} \tag{83}$$

where for $\widehat{a}_n$, the inner term equals

$$\frac{\partial}{\partial \widehat{a}_n} \left\{ \left[ \underbrace{\widehat{b}_n + 0.5 \left( \beta \boldsymbol{y}_i' \boldsymbol{y}_i + E_4 + E_{5,i} + E_{6,i} + E_{7,i} \right)}_{=K} \right]^{-\widehat{a}_n - 0.5 d\beta} \right\} = -\log (K) \cdot K^{-\widehat{a}_n - 0.5 d\beta}, \tag{84}$$

so that the differentiation with respect to $\widehat{a}_n$ requires obtaining the following terms:

$$\frac{\partial}{\partial \widehat{a}_n} E_2 = \frac{|\widehat{\boldsymbol{\Sigma}}_n^{-1}|^{0.5}}{\beta(2\pi)^{0.5d\beta}} \left[ \frac{\frac{\partial}{\partial \widehat{a}_n}\{\Gamma(\widehat{a}_n + 0.5d\beta)\}\widehat{b}_n^{\widehat{a}_n}}{\Gamma(\widehat{a}_n)} + \frac{\frac{\partial}{\partial \widehat{a}_n}\{\widehat{b}_n^{\widehat{a}_n}\}\Gamma(\widehat{a}_n + 0.5d\beta)}{\Gamma(\widehat{a}_n)} \right.$$
$$\left. + \frac{\partial}{\partial \widehat{a}_n}\{\Gamma(\widehat{a}_n)^{-1}\} \cdot \widehat{b}_n^{\widehat{a}_n}\Gamma(\widehat{a}_n + 0.5d\beta) \right]$$
$$= \frac{|\widehat{\boldsymbol{\Sigma}}_n^{-1}|^{0.5}\widehat{b}_n^{\widehat{a}_n}\Gamma(\widehat{a}_n + 0.5d\beta)}{\beta(2\pi)^{0.5d\beta}\Gamma(\widehat{a}_n)} \left[ \Psi(\widehat{a}_n + 0.5d\beta) + \log(\widehat{b}_n) - \Psi(\widehat{a}_n) \right] \quad (85)$$

$$\frac{\partial}{\partial \widehat{a}_n} E_8 = -\frac{1}{2}\left[(\boldsymbol{\mu}_0 - \widehat{\boldsymbol{\mu}}_n)'\boldsymbol{\Sigma}_0^{-1}(\boldsymbol{\mu}_0 - \widehat{\boldsymbol{\mu}}_n) + 2(b_0 - \widehat{b}_n)\right] \cdot \left[ \frac{\frac{\partial}{\partial \widehat{a}_n}\{\Gamma(\widehat{a}_n + 1)\}}{\widehat{b}_n\Gamma(\widehat{a}_n)} - \frac{\frac{\partial}{\partial \widehat{a}_n}\{\Gamma(\widehat{a}_n)\}\Gamma(\widehat{a}_n + 1)}{\Gamma(\widehat{a}_n)^2\widehat{b}_n} \right]$$
$$= -\frac{1}{2}\left[(\boldsymbol{\mu}_0 - \widehat{\boldsymbol{\mu}}_n)'\boldsymbol{\Sigma}_0^{-1}(\boldsymbol{\mu}_0 - \widehat{\boldsymbol{\mu}}_n) + 2(b_0 - \widehat{b}_n)\right] \cdot \frac{\Gamma(\widehat{a}_n + 1)}{\widehat{b}_n\Gamma(\widehat{a}_n)} \cdot \left[\Psi(\widehat{a}_n + 1) - \Psi(\widehat{a}_n)\right] \quad (86)$$

$$\frac{\partial}{\partial \widehat{a}_n} E_9 = -\frac{\partial}{\partial \widehat{a}_n}\left\{\widehat{a}_n \log(\widehat{b}_n)\right\} + \frac{\partial}{\partial \widehat{a}_n}\{\log(\Gamma(\widehat{a}_n))\}$$
$$= -\log(\widehat{b}_n) + \Psi(\widehat{a}_n) \quad (87)$$

$$\frac{\partial}{\partial \widehat{a}_n} E_{10} = \frac{\partial}{\partial \widehat{a}_n}\left\{\widehat{a}_n \log(\widehat{b}_n)\right\} - \frac{\partial}{\partial \widehat{a}_n}\{(\widehat{a}_n - a_0)\Psi(\widehat{a}_n)\}$$
$$= \log(\widehat{b}_n) - \Psi(\widehat{a}_n) - (\widehat{a}_n - a_0)\Psi^{(1)}(\widehat{a}_n) \quad (88)$$

$$\frac{\partial}{\partial \widehat{a}_n} E_{11} = -\frac{n}{\widehat{b}_n^{0.5d\beta}(2\pi)^{0.5d\beta}(1 + \beta)^{0.5d+1}} \cdot \frac{\partial}{\partial \widehat{a}_n}\left\{\frac{\Gamma(\widehat{a}_n + 0.5d\beta)}{\Gamma(\widehat{a}_n)}\right\}$$
$$= -\frac{n}{\widehat{b}_n^{0.5d\beta}(2\pi)^{0.5d\beta}(1 + \beta)^{0.5d+1}} \cdot \frac{\Gamma(\widehat{a}_n + 0.5d\beta)}{\Gamma(\widehat{a}_n)} \cdot \left[\Psi(\widehat{a}_n + 0.5d\beta) - \Psi(\widehat{a}_n)\right], \quad (89)$$

where $\Psi^{(1)}$ denotes the trigamma function.

### 3.4.4 Derivative with respect to $\widehat{b}_n$

As for the other variational parameters, note that

$$\frac{\partial}{\partial \widehat{b}_n}\{ELBO\} = \frac{\partial}{\partial \widehat{b}_n}\{E_8\} + \frac{\partial}{\partial \widehat{b}_n}\{E_9\} + \frac{\partial}{\partial \widehat{b}_n}\{E_{10}\} + \frac{\partial}{\partial \widehat{b}_n}\{E_{11}\} +$$
$$+ \frac{\partial}{\partial \widehat{b}_n}\{E_2\}\sum_{i=1}^{n}\left\{\frac{E_{3,i}}{\left[\widehat{b}_n + 0.5\left(\beta \boldsymbol{y}_i'\boldsymbol{y}_i + E_4 + E_{5,i} + E_{6,i} + E_{7,i}\right)\right]^{\widehat{a}_n + 0.5d\beta}}\right\}$$
$$+ E_2 \cdot \sum_{i=1}^{n}\left\{E_{3,i} \cdot \frac{\partial}{\partial \widehat{b}_n}\left\{\left[\widehat{b}_n + 0.5\left(\beta \boldsymbol{y}_i'\boldsymbol{y}_i + E_4 + E_{5,i} + E_{6,i} + E_{7,i}\right)\right]^{-\widehat{a}_n - 0.5d\beta}\right\}\right\} \quad (90)$$

where the chain rule implies that

$$\frac{\partial}{\partial \widehat{b}_n}\left\{\left[\underbrace{\widehat{b}_n + 0.5\left(\beta \boldsymbol{y}_i'\boldsymbol{y}_i + E_4 + E_{5,i} + E_{6,i} + E_{7,i}\right)}_{=K}\right]^{-\widehat{a}_n - 0.5d\beta}\right\} = (-\widehat{a}_n - 0.5d\beta) \cdot K^{-\widehat{a}_n - 0.5d\beta - 1}. \quad (91)$$

Thus one proceeds by the same logic as before.

$$\frac{\partial}{\partial \widehat{b}_n} E_2 = \frac{\widehat{a}_n \Gamma(\widehat{a}_n + 0.5d\beta) \cdot |\widehat{\boldsymbol{\Sigma}}_n^{-1}|^{0.5}}{\beta(2\pi)^{0.5d\beta}\Gamma(\widehat{a}_n)} \cdot \widehat{b}_n^{\widehat{a}_n - 1} \tag{92}$$

$$\frac{\partial}{\partial \widehat{b}_n} E_8 = \frac{1}{2}\left[(\boldsymbol{\mu}_0 - \widehat{\boldsymbol{\mu}}_n)'\boldsymbol{\Sigma}_0^{-1}(\boldsymbol{\mu}_0 - \widehat{\boldsymbol{\mu}}_n) + 2b_0\right]\frac{\Gamma(\widehat{a}_n + 1)}{\Gamma(\widehat{a}_n)} \cdot \frac{1}{\widehat{b}_n^2} \tag{93}$$

$$\frac{\partial}{\partial \widehat{b}_n} E_9 = -\frac{\widehat{a}_n}{\widehat{b}_n} \tag{94}$$

$$\frac{\partial}{\partial \widehat{b}_n} E_{10} = \frac{\widehat{a}_n - a_0}{\widehat{b}_n} \tag{95}$$

$$\frac{\partial}{\partial \widehat{b}_n} E_{11} = \frac{nd\beta \cdot \Gamma(\widehat{a}_n + 0.5d\beta)}{2 \cdot \Gamma(\widehat{a}_n)(2\pi)^{0.5d\beta}(1 + \beta)^{0.5d+1}} \cdot \widehat{b}_n^{-0.5d\beta - 1} \tag{96}$$

## 4  Timing and performance comparisons: Markov Chain Monte Carlo vs Structured Variational Bayes

We ran timing comparisons of SVI with MCMC for several subsets of the well-log data set. We ran the $\beta$-D BOCPD algorithm implementing an MCMC inference regime using *stan* [Carpenter et al., 2016] and compared this with our SVI inference regime. The two inference schemes were then run on 3 datasets of different time-length; the first 200 observations of the well-log, the first 500 observations of the well log and the full well-log, in order to show the impact changing the number of observations has on the timings for the algorithm. For the SVI used to produce these timings, we perform *full* optimization at every step, which is significantly slower than the SGD-variant that we present in the paper and that can be found in our repository at `https://github.com/alan-turing-institute/rbocpdms`. In spite of this, the SVI is still orders of magnitudes faster.

Table 1: Table of times to run the $\beta$-D BOCPD algorithm under the MCMC and SVI with full optimization on the first 200 observations, the first 500 observations and the full well log dataset.

|  | T=200 | T=500 | T=4050 |
|---|---|---|---|
| MCMC | 7615.2 | 20388.7 | 106073.0 |
| SVI (full optimization) | 102.8 | 328.5 | 3240.0 |

Another question of interest is how much the Stochastic Gradient Descent (SGD) inside our inference procedure provides robustness and how much the $\beta$-D itself is responsile for this. To put this question to the test, we ran full vs SGD-based optimization on the well-log data. As shown in Fig. 3, the results are very close to identical: No CPs are declared under one that are not declared in the other, and the run-length distribution's maximum coincides throughout.

Figure 3: MAP segmentation and run-length distributions of the well-log data. SGD inference outcomes in blue, outcomes under full optimization in red. The corresponding run-length distributions for SGD (middle) and full (bottom) optimization are shown in grayscale with dashed maximum.

# 5 Initialization for $\beta_\mathrm{p}$

The initialization procedure described in the paper is illustrated in Fig. 4. Here, the yellow dashed line gives a standard normal density corresponding to our model for the data. The gray dotted vertical line gives the amount of standard deviations from the posterior mean where one wishes to maximize the influence. We have chosen to maximize the influence at observations with $2.75$ standard deviations away from our posterior mean. In the first picture, $\beta_\mathrm{p} = 0$ and thus the influence function corresponds to the Kullback-Leibler Divergence. Concordantly, it has no maximum and observations have more influence the further in the tail of our model they occur. Thus, one needs to increase $\beta_\mathrm{p}$ slightly. This is done in the second picture. While observations in the tail get smaller influence now than before, the influence of observations is still increasing beyond $2.75$ standard deviations. So one needs to increase $\beta_\mathrm{p}$ two more times, until one finally obtains the desired outcome for $\beta_\mathrm{p} = 0.25$ in the fourth picture. Notice that the influence does not immediately drop to $0$ for observations further in the tails than $2.75$ standard deviations away from the posterior mean, but it does decay. In some sense, we have set $\beta_\mathrm{p}$ such that we think of observations occuring $2.75$ standard deviations away from the posterior mean as being *most informative*. This is significantly different from what is implied by the Kullback-Leibler divergence, where an observation is most informative if it agrees least with the fitted model. It is intuitive why this produces good inferences if one is in the M-closed world and similarly intuitive why it does not in the M-open world.

Figure 4: Illustration of the initialization procedure, from left to right.

# 6 Recursive Optimization for $\beta_\mathrm{rlm}$

Recall that

$$\widehat{\boldsymbol{y}}_t(\boldsymbol{\beta}) = \sum_{r_t, m_t} \mathbb{E}\left(\boldsymbol{y}_t | \boldsymbol{y}_{1:(t-1)}, r_{t-1}, m_{t-1}, \beta_\mathrm{p}\right) p(r_{t-1}, m_{t-1} | \boldsymbol{y}_{1:(t-1)}, \beta_\mathrm{rlm}). \quad (97)$$

the issue reduces to finding the partial derivatives $\nabla_{\beta_\mathrm{rlm}} \widehat{\boldsymbol{y}}_t(\boldsymbol{\beta})$ and $\nabla_{\beta_\mathrm{p}} \widehat{\boldsymbol{y}}_t(\boldsymbol{\beta})$. Notice that for $\nabla_{\beta_\mathrm{rlm}} \widehat{\boldsymbol{y}}_t(\boldsymbol{\beta})$, one finds that

$$\nabla_{\beta_\mathrm{rlm}} \widehat{\boldsymbol{y}}_t(\boldsymbol{\beta}) = \sum_{r_t, m_t} \mathbb{E}\left(\boldsymbol{y}_t | \boldsymbol{y}_{1:(t-1)}, r_{t-1}, m_{t-1}, \beta_\mathrm{p}\right) \nabla_{\beta_\mathrm{rlm}} p(r_{t-1}, m_{t-1} | \boldsymbol{y}_{1:(t-1)}, \beta_\mathrm{rlm}). \quad (98)$$

Observe now that for $p(\boldsymbol{y}_{1:t}) = \sum_{r_t, m_t} p(r_t, m_t, \boldsymbol{y}_{1:t} | \beta_\mathrm{rlm})$,

$$\begin{aligned} &\nabla_{\beta_\mathrm{rlm}} p(r_t, m_t | \boldsymbol{y}_{1:t}, \beta_\mathrm{rlm}) \\ =&\nabla_{\beta_\mathrm{rlm}} \left\{ \frac{p(r_t, m_t, \boldsymbol{y}_{1:t} | \beta_\mathrm{rlm})}{\sum_{r_t, m_t} p(r_t, m_t, \boldsymbol{y}_{1:t} | \beta_\mathrm{rlm})} \right\} \\ =& \frac{\nabla_{\beta_\mathrm{rlm}} p(r_t, m_t, \boldsymbol{y}_{1:t} | \beta_\mathrm{rlm})}{p(\boldsymbol{y}_{1:t})} - \frac{p(r_t, m_t, \boldsymbol{y}_{1:t} | \beta_\mathrm{rlm})}{p(\boldsymbol{y}_{1:t})^2} \cdot \sum_{r_t, m_t} \nabla_{\beta_\mathrm{rlm}} p(r_t, m_t, \boldsymbol{y}_{1:t} | \beta_\mathrm{rlm}). \end{aligned} \quad (99)$$

Thus we have reduced the problem to finding $\nabla_{\beta_{\mathrm{rlm}}} p(r_t, m_t, \boldsymbol{y}_{1:t}|\beta_{\mathrm{rlm}})$. Defining for a predictive posterior distribution $f_{m_t}(\boldsymbol{y}_t|\mathcal{F}_{t-1})$ its $\beta$-divergence analogue as

$$f_{m_t}^{\beta_{\mathrm{rlm}}}(\boldsymbol{y}_t|\mathcal{F}_{t-1}) = \exp\left\{\frac{1}{\beta_{\mathrm{rlm}}} f_{m_t}(\boldsymbol{y}_t|\mathcal{F}_{t-1})^{\beta_{\mathrm{rlm}}} - \frac{1}{1+\beta_{\mathrm{rlm}}}\int_{\mathcal{Y}} f_{m_t}(\boldsymbol{y}_t|\mathcal{F}_{t-1})^{1+\beta_{\mathrm{rlm}}} d\boldsymbol{y}_t\right\} \quad (100)$$

and uppressing the conditioning on $\beta_{\mathrm{rlm}}$ for convenience, one can using the recursion

$$p(\boldsymbol{y}_{1:t}, r_t, m_t) = \sum_{m_{t-1}, r_{t-1}}\left\{f_{m_t}^{\beta_{\mathrm{rlm}}}(\boldsymbol{y}_t|\mathcal{F}_{t-1}) q(m_t|\mathcal{F}_{t-1}, m_{t-1}) H(r_t, r_{t-1}) p(\boldsymbol{y}_{1:(t-1)}, r_{t-1}, m_{t-1})\right\} \quad (101a)$$

compute $\nabla_{\beta_{\mathrm{rlm}}} p(r_t, m_t, \boldsymbol{y}_{1:t})$ from $\nabla_{\beta_{\mathrm{rlm}}} p(r_{t-1}, m_{t-1}, \boldsymbol{y}_{1:(t-1)}|\beta_{\mathrm{rlm}})$ for $r_t = r_{t-1} + 1$ as

$$\nabla_{\beta_{\mathrm{rlm}}} p(\boldsymbol{y}_{1:t}, r_t, m_t)$$
$$= \left\{\nabla_{\beta_{\mathrm{rlm}}} f_{m_t}^{\beta_{\mathrm{rlm}}}(\boldsymbol{y}_t|\mathcal{F}_{t-1}) q(m_t|\mathcal{F}_{t-1}, m_{t-1}) H(r_t, r_{t-1}) p(\boldsymbol{y}_{1:(t-1)}, r_{t-1}, m_{t-1})\right\} +$$
$$\left\{f_{m_t}^{\beta_{\mathrm{rlm}}}(\boldsymbol{y}_t|\mathcal{F}_{t-1}) \nabla_{\beta_{\mathrm{rlm}}} q(m_t|\mathcal{F}_{t-1}, m_{t-1}) H(r_t, r_{t-1}) p(\boldsymbol{y}_{1:(t-1)}, r_{t-1}, m_{t-1})\right\} +$$
$$\left\{f_{m_t}^{\beta_{\mathrm{rlm}}}(\boldsymbol{y}_t|\mathcal{F}_{t-1}) q(m_t|\mathcal{F}_{t-1}, m_{t-1}) H(r_t, r_{t-1}) \nabla_{\beta_{\mathrm{rlm}}} p(\boldsymbol{y}_{1:(t-1)}, r_{t-1}, m_{t-1})\right\}. \quad (102)$$

Similarly, for $r_t = 0$ the expression becomes

$$\nabla_{\beta_{\mathrm{rlm}}} p(\boldsymbol{y}_{1:t}, r_t, m_t)$$
$$= \nabla_{\beta_{\mathrm{rlm}}} f_{m_t}^{\beta_{\mathrm{rlm}}}(\boldsymbol{y}_t|\mathcal{F}_{t-1}) \cdot q(m_t) \sum_{r_{t-1}, m_{t-1}} H(0, r_{t-1}) p(\boldsymbol{y}_{1:(t-1)}, r_{t-1}, m_{t-1}) +$$
$$f_{m_t}^{\beta_{\mathrm{rlm}}}(\boldsymbol{y}_t|\mathcal{F}_{t-1}) \cdot q(m_t) \sum_{r_{t-1}, m_{t-1}} H(0, r_{t-1}) \nabla_{\beta_{\mathrm{rlm}}} p(\boldsymbol{y}_{1:(t-1)}, r_{t-1}, m_{t-1}). \quad (103)$$

This implies that if $f_{m_t}^{\beta_{\mathrm{rlm}}}(\boldsymbol{y}_t|\mathcal{F}_{t-1})$ and $q(m_t|\mathcal{F}_{t-1}, m_{t-1})$ are differentiable with respect to $\beta_{\mathrm{rlm}}$, then the entire expression can be updated recursively. For most exponential family likelihoods (and in particular the normal likelihood of the Bayesian Linear Regression), $\nabla_{\beta_{\mathrm{rlm}}} f_{m_t}^{\beta_{\mathrm{rlm}}}(\boldsymbol{y}_t|\mathcal{F}_{t-1})$ is available analytically. In particular, as long as $\int_{\mathcal{Y}} f_{m_t}(\boldsymbol{y}_t|\mathcal{F}_{t-1})^{1+\beta_{\mathrm{rlm}}} d\boldsymbol{y}_t$ has a closed form, $\nabla_{\beta_{\mathrm{rlm}}} f_{m_t}^{\beta_{\mathrm{rlm}}}(\boldsymbol{y}_t|\mathcal{F}_{t-1})$ can be found in analytic form. In the case of Bayesian Linear Regression where the $d$-dimensional posterior predictive takes the shape of a student-$t$ distribution with $\nu$ degrees of freedom and posterior covariance $\frac{\nu}{\nu-2}\Sigma$, one finds that

$$\nabla_{\beta_{\mathrm{rlm}}} f_{m_t}^{\beta_{\mathrm{rlm}}}(\boldsymbol{y}_t|\mathcal{F}_{t-1}) = \nabla_{\beta_{\mathrm{rlm}}} g_1(\beta_{\mathrm{rlm}}) g_2(\beta_{\mathrm{rlm}}) g_3(\beta_{\mathrm{rlm}}) +$$
$$g_1(\beta_{\mathrm{rlm}}) \nabla_{\beta_{\mathrm{rlm}}} g_2(\beta_{\mathrm{rlm}}) g_3(\beta_{\mathrm{rlm}}) +$$
$$g_1(\beta_{\mathrm{rlm}}) g_2(\beta_{\mathrm{rlm}}) \nabla_{\beta_{\mathrm{rlm}}} g_3(\beta_{\mathrm{rlm}}), \quad (104)$$

where for $\eta = \nu d + d\beta_{\mathrm{rlm}} + \nu$,

$$g_1(\beta_{\mathrm{rlm}}) = \left(\frac{\Gamma(0.5[\nu+d])}{\Gamma(0.5\nu)}\right)^{1+\beta_{\mathrm{rlm}}}$$

$$g_2(\beta_{\mathrm{rlm}}) = \frac{\Gamma(0.5\eta)}{\Gamma(0.5[\eta+p])}$$

$$g_3(\beta_{\mathrm{rlm}}) = (\nu\pi)^{-0.5p\cdot\beta_{\mathrm{rlm}}} \cdot |\Sigma|^{-\beta_{\mathrm{rlm}}},$$

so that their derivatives are given by

$$\nabla_{\beta_{\mathrm{rlm}}} g_1(\beta_{\mathrm{rlm}}) = -(\beta_{\mathrm{rlm}} + 1) \cdot \log(g_1(\beta_{\mathrm{rlm}})) \cdot g_2(\beta_{\mathrm{rlm}})$$
$$\nabla_{\beta_{\mathrm{rlm}}} g_1(\beta_{\mathrm{rlm}}) = 0.5(\nu + p)\left[\cdot\frac{\Gamma(0.5\eta)\Psi(0.5\eta)}{\Gamma([0.5[\eta+p])} - \frac{\Gamma(0.5[\eta])\Psi(0.5[p+\eta])}{\Gamma([0.5[\eta+p])}\right]$$
$$\nabla_{\beta_{\mathrm{rlm}}} g_3(\beta_{\mathrm{rlm}}) = -g_3(\beta_{\mathrm{rlm}}) \cdot \log(g_3(\beta_{\mathrm{rlm}})) \cdot \frac{1}{\beta_{\mathrm{rlm}}}$$

$$(105)$$

As for $\nabla_{\beta_{\mathrm{rlm}}} q(m_t|\mathcal{F}_{t-1}, m_{t-1})$, one can again obtain it recursively, since for $r_t > 0$,

$$\nabla_{\beta_{\mathrm{rlm}}} q(m_t|\mathcal{F}_{t-1}, m_{t-1})$$
$$= \nabla_{\beta_{\mathrm{rlm}}}\left\{\frac{p(\boldsymbol{y}_{1:(t-1)}, r_{t-1}, m_{t-1})}{\sum_{m_{t-1}} p(\boldsymbol{y}_{1:(t-1)}, m_{t-1})}\right\}$$
$$= \frac{\nabla_{\beta_{\mathrm{rlm}}} p(\boldsymbol{y}_{1:(t-1)}, r_{t-1}, m_{t-1})}{\sum_{m_{t-1}} p(\boldsymbol{y}_{1:(t-1)}, r_{t-1}, m_{t-1})} - \frac{\sum_{m_{t-1}} \nabla_{\beta_{\mathrm{rlm}}} p(\boldsymbol{y}_{1:(t-1)}, r_{t-1}, m_{t-1})}{\left(\sum_{m_{t-1}} p(\boldsymbol{y}_{1:(t-1)}, r_{t-1}, m_{t-1})\right)^2}. \quad (106)$$

# 7 Proof of Theorem 2

*Proof.* For ease of notation, we use $\beta_{\mathrm{p}} = \beta$. The model used for the inference is an exponential family model of the form

$$f(x;\theta) = \exp\left(\eta(\theta)^T T(x)\right) g(\eta(\theta)) A(x), \tag{107}$$

where $g(\eta(\theta)) := \left(\int \exp\left(\eta(\theta)^T T(x)\right) A(x)dx\right)^{-1}$. Now under our SVI routine the $\beta$-D posterior originating from this model and its conjugate prior is approximated by a member of the conjugate prior family. As a result the conjugate prior and variational posterior to the above model have the form

$$\pi_0(\theta|\nu_0, \mathcal{X}_0) = g(\eta(\theta))^{\nu_0} \exp\left(\nu_0\eta(\theta)^T \mathcal{X}_0\right) h(\mathcal{X}_0, \nu_0) \tag{108}$$

$$\pi_n^{VB}(\theta|\nu_n, \mathcal{X}_n) = g(\eta(\theta))^{\nu_n} \exp\left(\nu_n\eta(\theta)^T \mathcal{X}_n\right) h(\mathcal{X}_n, \nu_n), \tag{109}$$

where $(\nu_0, \mathcal{X}_0)$ are the prior hyperparameters, $(\nu_n, \mathcal{X}_n)$ represent the variational parameters and $h(\mathcal{X}_i, \nu_i) := \left(\int g(\eta(\theta))^{\nu_i} \exp\left(\nu_i\eta(\theta)^T \mathcal{X}_i\right) d\theta\right)^{-1}$. The resulting ELBO objective function under GBI has the form

$$ELBO(\nu_n, \mathcal{X}_n) =$$
$$\mathbb{E}_{\pi_n^{VB}}\left[\log\left(\exp\left(\sum_{i=1}^n -\ell^D(x;\theta)\right)\right)\right] - d_{KL}\left(\pi_n^{VB}(\theta|\nu_n, \mathcal{X}_n), \pi_0(\theta|\nu_0, \mathcal{X}_0)\right), \tag{110}$$

where for the $\beta$-D posterior

$$-\ell^\beta(x;\theta) = \frac{1}{\beta}\left(\exp\left(\eta(\theta)^T T(x)\right) g(\eta(\theta)) A(x)\right)^\beta -$$
$$\frac{1}{\beta+1}\int\left(\exp\left(\eta(\theta)^T T(z)\right) g(\eta(\theta)) A(x)\right)^{1+\beta} dz \tag{111}$$
$$= \frac{1}{\beta}\exp\left(\beta\eta(\theta)^T T(x)\right) g(\eta(\theta))^\beta A(x)^\beta -$$
$$\frac{1}{\beta+1}\int\exp\left((1+\beta)\eta(\theta)^T T(z)\right) g(\eta(\theta))^{1+\beta} A(x)^{1+\beta} dz. \tag{112}$$

Therefore the $ELBO(\nu_n, \mathcal{X}_n)$ has three integrals that need evaluating

$$B_1 = \sum_{i=1}^n \int \frac{1}{\beta}\exp\left(\beta\eta(\theta)^T T(x_i)\right) g(\eta(\theta))^\beta A(x_i)^\beta \pi_n^{VB}(\theta|\nu_n, \mathcal{X}_n) d\theta \tag{113}$$

$$B_2 = \frac{n}{\beta+1}\int\left\{\int\exp\left((1+\beta)\eta(\theta)^T T(z)\right) g(\eta(\theta))^{1+\beta} A(z))^{1+\beta} dz\right\} \pi_n^{VB}(\theta|\nu_n, \mathcal{X}_n) d\theta \tag{114}$$

$$B_3 = d_{KL}\left(\pi_n^{VB}(\theta|\nu_n, \mathcal{X}_n), \pi_0(\theta|\nu_0, \mathcal{X}_0)\right). \tag{115}$$

Now firstly for the term $B_1$ in equation (113)

$$B_1 = \sum_{i=1}^n \int \frac{1}{\beta}\exp\left(\beta\eta(\theta)^T T(x_i)\right) g(\eta(\theta))^\beta A(x_i)^\beta g(\eta(\theta))^{\nu_n} \exp\left(\nu_n\eta(\theta)^T \mathcal{X}_n\right) h(\mathcal{X}_n, \nu_n) d\theta \tag{116}$$

$$= \sum_{i=1}^n \frac{1}{\beta} A(x_i)^\beta h(\mathcal{X}_n, \nu_n) \int g(\eta(\theta))^{\beta+\nu_n} \exp\left(\eta(\theta)^T (\beta T(x_i) + \nu_n \mathcal{X}_n)\right) d\theta \tag{117}$$

$$= \sum_{i=1}^n \frac{1}{\beta} A(x_i)^\beta h(\mathcal{X}_n, \nu_n) \frac{1}{h\left(\frac{\beta T(x_i) + \nu_n \mathcal{X}_n}{\beta+\nu_n}, \beta+\nu_n\right)}. \tag{118}$$

Where we know that $h(\frac{\beta T(x_i)+\nu_n \mathcal{X}_n}{\beta+\nu_n}, \beta + \nu_n) = \int g(\eta(\theta))^{\beta+\nu_n} \exp\left(\eta(\theta)^T\left(\beta T(x_i) + \nu_n \mathcal{X}_n\right)\right) d\theta$
is integrable and closed form as it represents the normalising constant of the same exponential family as the prior and the variational posterior. Next we look at $B_2$ in equation (114). The whole integral is the product of two densities which must be positive and in order for the $ELBO(\nu_n, \mathcal{X}_n)$ to be defined it must also be integrable. Therefore we can use Fubini's theorem to switch the order of integration

$$B_2 = \frac{n}{\beta+1} \int \left\{ \int \exp\left((1+\beta)\eta(\theta)^T T(z)\right) g(\eta(\theta))^{1+\beta} \pi_n^{VB}(\theta|\nu_n, \mathcal{X}_n) d\theta \right\} A(z)^{1+\beta} dz \quad (119)$$

$$= \frac{n}{\beta+1} h(\mathcal{X}_n, \nu_n) \int \left\{ \int \exp\left(\eta(\theta)^T\left((1+\beta)T(z) + \nu_n\mathcal{X}_n\right)\right) g(\eta(\theta))^{1+\beta+\nu_n} d\theta \right\} A(z)^{1+\beta} dz \quad (120)$$

$$= \frac{n}{\beta+1} h(\mathcal{X}_n, \nu_n) \int \frac{A(z)^{1+\beta}}{h(\frac{(1+\beta)T(z)+\nu_n \mathcal{X}_n}{1+\beta+\nu_n}, 1+\beta+\nu_n)} dz. \quad (121)$$

once again $h(\frac{(1+\beta)T(z)+\nu_n\mathcal{X}_n}{1+\beta+\nu_n}, 1+\beta+\nu_n) = \int \exp\left(\eta(\theta)^T\left((1+\beta)T(z) + \nu_n\mathcal{X}_n\right)\right) g(\eta(\theta))^{1+\beta+\nu_n} d\theta$
is the normalisisng constant of the same exponential family as the prior and the variational posterior and is thus closed form. Lastly we look at $B_3$ in equation (115)

$$B_3 = \int \pi_n^{VB}(\theta|\nu_n, \mathcal{X}_n) \log \frac{g(\eta(\theta))^{\nu_n} \exp\left(\nu_n \eta(\theta)^T \mathcal{X}_n\right) h(\mathcal{X}_n, \nu_n)}{g(\eta(\theta))^{\nu_0} \exp\left(\nu_0 \eta(\theta)^T \mathcal{X}_0\right) h(\mathcal{X}_0, \nu_0)} \quad (122)$$

$$= \log \frac{h(\mathcal{X}_n, \nu_n)}{h(\mathcal{X}_0, \nu_0)} \int \pi_n^{VB}(\theta|\nu_n, \mathcal{X}_n) \left\{(\nu_n - \nu_0)\log g(\eta(\theta)) + \left(\eta(\theta)^T\left(\nu_n\mathcal{X}_n - \nu_0\mathcal{X}_0\right)\right)\right\} \quad (123)$$

$$= \log \frac{h(\mathcal{X}_n, \nu_n)}{h(\mathcal{X}_0, \nu_0)} \left\{(\nu_n - \nu_0)\lambda_n^{VB} + \left((\mu_n^{VB})^T\left(\nu_n\mathcal{X}_n - \nu_0\mathcal{X}_0\right)\right)\right\}, \quad (124)$$

where $\mu_n^{VB} = \mathbb{E}_{\pi_n^{VB}}\left[\eta(\theta)\right]$ and $\lambda_n^{VB} = \mathbb{E}_{\pi_n^{VB}}\left[\log g(\eta(\theta))\right]$.

As a result we get that

$$ELBO(\nu_n, \mathcal{X}_n) = B_1 - B_2 - B_3 \quad (125)$$

$$= \sum_{i=1}^{n} \frac{1}{\beta} A(x_i)^{\beta} h(\mathcal{X}_n, \nu_n) \frac{1}{h(\frac{\beta T(x_i)+\nu_n \mathcal{X}_n}{\beta+\nu_n}, \beta + \nu_n)}$$

$$- \frac{n}{\beta+1} h(\mathcal{X}_n, \nu_n) \int \frac{A(z)^{1+\beta}}{h(\frac{(1+\beta)T(z)+\nu_n \mathcal{X}_n}{1+\beta+\nu_n}, 1+\beta+\nu_n)} dz \quad (126)$$

$$- \log \frac{h(\mathcal{X}_n, \nu_n)}{h(\mathcal{X}_0, \nu_0)} \left\{(\nu_n - \nu_0)\lambda_n^{VB} + \left((\mu_n^{VB})^T\left(\nu_n\mathcal{X}_n - \nu_0\mathcal{X}_0\right)\right)\right\}.$$

$\square$

# 8 Complexity Analysis of Inference

**Time complexity:** Our SVRG method crucially hinges on the complexity of the gradient evaluations. For BLR, we note that evaluating the complete ELBO gradient derived above for $n$ observations has complexity $\mathcal{O}(np^3)$, where $p$ is the number of regressors. We proceed by defining $g$ as the (generic) complexity of a gradient evaluation, so for BLR $g = p^3$. Clearly, an SGD step using $b$ observations is of order $\mathcal{O}(bg)$. Similarly, the computation of the anchors is $\mathcal{O}(Bg)$. Next, let the optimization routine used for full optimization have complexity $\mathcal{O}(m(n, \dim(\boldsymbol{\theta})))$. Most standard (quasi-) Newton optimization routines such as BFGS or LBFGSB (used in our implementation) are polynomial in $n$ and $\dim(\boldsymbol{\theta})$. For such methods, since it holds that at most $W \geqslant n$ observations are evaluated in the full optimization, and since $\dim(\boldsymbol{\theta})$ is time-constant, $m(n, \dim(\boldsymbol{\theta}))$ is also constant in time. Thus, though these constants can be substantial, all optimization steps (whether SVRG steps

or full optimization steps) are $\mathcal{O}(1)$ in time. Since one performs $T$ of them for $T$ observations, the computational complexity (in time) is $\mathcal{O}(T)$.

**Space complexity**: One needs to store observations $\boldsymbol{y}_t$ as well as gradient evaluations. Storing one of them takes $\mathcal{O}(d)$ and $\mathcal{O}(\dim(\boldsymbol{\theta}))$ space, respectively. Since we only keep a window $W$ of the most recent observations (and gradients), this means that the space requirement is of order $\mathcal{O}(W(d + \dim(\boldsymbol{\theta})))$ and in particular constant in time.

## 9    Additional Details on Experiments

For all experiment, constrained Limited Memory Broyden–Fletcher–Goldfarb–Shannon is used for the full optimization step, where the constraints are $\widehat{a}_n > 1, \widehat{b}_n > 1$. We use Python's `scipy.optimize` wrapper, which calls a Fortran implementation. We also tested whether inference is sensitive to different initializations of $\beta_{\mathrm{p}}$ and found that it is fairly stable as long as $\beta_{\mathrm{p}}$ is chosen reasonably. For example, for the Air Pollution data, we could recover the same changepoint ($\pm 5$ days) for initializations of $\beta_{\mathrm{p}}$ ranging from $0.005$ up to $0.1$. All experiments were performed on a 2017 MacBook Pro with 16 GB 2133 MHz LPDDR3 and 3.1 GHz Intel Core i7.

### 9.1    Well-log data

**Hyperparameters:**    We set the hyperparameters for standard Bayesian On-line Changepoint Detection slightly differently, the reason being that due to the robustness guarantee of Theorem 1, we can use much less informative priors with the robust version than we can with the standard version: If priors are too flat, the standard version declares far too many changepoints. Thus, for the standard version, we use a constant CP prior (hazard) $H(r_t = r_{t-1} + 1 | r_{t-1}) = 0.01$, $a_0 = 1$, $b_0 = 10^4$, $\Sigma_0 = 0.25$, $\boldsymbol{\mu}_0 = 1.15 \cdot 10^4$, while for the robust version we can use a less informative prior by instead setting $b_0 = 10^7$. By virtue of our initialization procedure for $\beta_p$, this implies setting $\beta_{p,0} \approx 0.05$. To start out close to the KLD, we initialize $\beta_{\mathrm{rld},0} = 0.0001$.

**Inferential procedure:**    For the robust version, we set $W = 360$, $B = 25$, $b = 10$, $m = 20$, $K = 1$. For both versions, only the $50$ most likely run-lengths are kept. For the robust version, the average processing time was $0.487$ per observation.

### 9.2    Air Pollution data

**Preprocessing & Model Setup:** The air pollution data is observed every 15 minutes across 29 stations for 365 days. We average the 96 observations made over 24 hours. This is done to move the observed data closer to a normal distribution, as the measurements have significant daily volatility variations. To account for weekly cycles, we also calculate for each station the mean for each weekday and subtract it from the raw data.. Yearly seasonality is not accounted for. Afterwards, the data is normalized station-wise. This is done only for numerical stability, because the internal mechanisms of the used VAR models perform matrix operations (QR-decompositions and matrix multiplications in particular) that can adversely affect numerical stability for observations with large absolute value. Fig. 5 shows some of the station's data after these preprocessing steps have been taken.

The autoregressive models and spatially structured vector autoregressive models (VARs) are chosen to have lag lengths $1, 2, 3$. These short lag lengths are chosen to explicitly disadvantage the robust model universe: The non-robust run we compare against uses more than 20 models, with lag lengths $1, 5, 6, 7$, meaning that it is much more expressive and should be able to cope with outliers better. In spite of this, it not only declares more CPs, but also does worse than the robust version in terms of predictive performance. For both the robust and non-robust model, two spatially structured VARs are included as in Knoblauch and Damoulas [2018].

**Hyperparameters:**    We set $H(r_t = r_{t-1} + 1 | r_{t-1}) = 0.001$, $a_0 = 1$, $b_0 = 25$, $\boldsymbol{\mu}_0 = \mathbf{0}$, $\Sigma_0 = I \cdot 20$, which yields initialization $\beta_{\mathrm{p}} \approx 0.005$, $\beta_{\mathrm{rlm}} = 0.1$. The non-robust results are directly taken from Knoblauch and Damoulas [2018] and can be replicated running the code available from `https://github.com/alan-turing-institute/bocpdms/`

Figure 5: Some of the stations after preprocessing steps. $x$-axis gives NOX level, $y$-axis the day.

**Inferential procedure:** We set $W = 300$, $m = 50$, $B = 20$ and $b = 10$, $K = 25$ and retain the 50 most likely run-lengths. Processing times are more volatile than for the well-log because the full optimization procedure is significantly more expensive to perform. Most observations take significantly less than 20 seconds to process, but some take over a minute (depending on how many of the retained run-lengths are divisible by $m$ at each time point).

## 9.3 Optimizing $\beta$

Lastly, we investigate the trajectories for $\beta$ as it is being optimized. For all trajectories, a bounded predictive absolute loss was used with threshold $\tau$, i.e. $L(x) = \max\{|x|, \tau\}$. For $\beta_{\mathrm{rld}}$, $\tau = 5/T$ (where $T$ is the length of the time series) while for $\beta_{\mathrm{p}}$, $\tau = 0.1$. The results are not sensitive to these thresholds, and they are picked with the intent that (1) a single observation should not affect $\beta_{\mathrm{p}}$ by more than 0.1 and (2) that overall, $\beta_{\mathrm{rld}}$ should not change by more than 5 in absolute magnitude. As the initialization procedure for $\beta_{\mathrm{p}}$ works very well for predictive performance, the on-line optimization never even comes close to making a step with size $\tau$. The picture is rather different for $\beta_{\mathrm{rld}}$, which reaches $\tau$ rather often. We note that this is because the estimated gradients for $\beta_{\mathrm{rld}}$ can be very extreme, which is why the implementation averages 50 consecutive gradients before performing a step. Overall, we note that for the well log data whose trajectories are depicted in Fig. 6, the degrees of robustness do not change much relative to their starting points at $\beta_{\mathrm{p}} = 0.05$ and $\beta_{\mathrm{rld}} = 0.001$. In particular, the absolute change over more than $4,000$ observations is $< 0.002$ for $\beta_{\mathrm{p}}$ and $< 0.015$ for $\beta_{\mathrm{rld}}$. Step sizes are $1/t$ at time $t$.

For the Air Pollution Data, the story is slightly different: Here, $\beta_{\mathrm{p}}$ does not change after the first iteration, where it jumps from 0.005 directly to $10^{-10}$. While this seems odd, it is mainly due to the fact that for numerical stability reasons[2] , one needs to ensure that $\beta_{\mathrm{p}} > \varepsilon$ for some $\varepsilon > 0$; and in our implementation, $\varepsilon = 10^{-10}$. The interpretation of the trace graph is thus that the optimization continuously suggests less robust values for $\beta_{\mathrm{p}}$, but that we cannot admit them due to numerical stability. The downward trend also holds for $\beta_{\mathrm{rld}}$, which is big enough to not endanger numerical stability and hence can drift downwards.

Fig. 6 also shows that the optimization technique used for $\beta$ needs further investigation and research. For starters, the outcomes suggest that a second order method could yield better results than using a first-order SGD technique. In the future, we would like to explore this in greater detail and also

Figure 6: $\beta$ trajectories for the well-log data. For $\beta_{\mathrm{rld}}$, steps are only taken every 50 observations to average gradient noise

explore more advanced optimization methods like line search or trust region optimization methods for this problem.

## Footnotes

[1]Note that $\boldsymbol{L}$ need not be unique if $\widehat{\boldsymbol{\Sigma}}_n$ is positive semi-definite, but this is of no concern for us here: Since we implicitly impose that $\widehat{\boldsymbol{\Sigma}}_n$ is non-singular (so that $\widehat{\boldsymbol{\Sigma}}_n^{-1}$ is unique and well-defined), all covariance matrices $\widehat{\boldsymbol{\Sigma}}_n$ considered have to be positive definite.

[2] In particular, working with the $\beta$-D implies that one takes the exponential of a density, i.e. $e^{f^{\beta}}$. So even working on a log scale now means working with the densities $f^{\beta}$ *directly*. It should be clear that these quantities become numerically unstable for $\beta$ too large or too small.