[Reviews · NeurIPS 2018]

Reviewer 1



The paper provided a doubly robust Bayesian inference method for online change-point detection, through the application of General Bayesian Inference with \beta-divergences. Theoretical analysis and empirical results demonstrate that the proposed method is robust to outliers and can adapt to non-stationary data via online parameter update. Overall, this is a paper rigorously written and a valuable contribution to the change-point detection community. Here are some additional comments about the paper: 1) In the algorithm description in Section 3.2, there is an operation called FullOpt, however I could not find the exact definition of this operator. 2) In the same section, the authors mentioned that "an MCMC implementation in Stan takes 10^5 times longer". I was wondering how this conclusion is obtained. It would be better if the authors give concrete running times for both approaches. 3) The main content of the paper is about change-point detection, but the title of the paper is about Robust Bayesian inference in general. I would recommend to have a title that more accurately reflects the content of the paper. Note: The authors agreed in the response to include more details of computational time in the final version, to make the comparison more clear.

Reviewer 2



Overview The paper introduces a robust online change point detection algorithm for non-stationary time-series data. Robustness comes as a by product of minimizing \beta-divergence between data and fitted model as opposed to using KL divergence as in standard Bayesian inference. \beta-Divergence uses Tsallis loss function which assigns less influence to samples in the tails and as a result the inference that relies on \beta-divergence becomes less sensitive to outliers and the model is less likely to call a random spike as a change point. In the generalized Bayesian inference the posteriors are intractable. The paper mitigate this problem by resorting to structural variational approximation, which is proved to be exact as \beta converges to zero. The paper also discusses systematic approaches to initialize \beta and refine it online. A heuristic stochactic gradient descent is proposed to make the algorithm scalable for streaming data. The main idea is to achieve a trade-off between accuracy and scalability by anchoring stochastic gradient near an optimum. Technical Quality The paper uses a similar idea as in Fearnhead and Rigaill to quantify robustness when studying the odds of r_{t+1} \in {0, r+1} but some motivation as to why these odds are important and why r+1 is selected for robustness would be useful. The paper argues that MCMC applications for online CP detection has been sparse mainly because MCMC are not very scalable. This argument is somewhat unjustified. Similar problems have been dealt with from a stochastic inference point of view using sequential Monte carlo (SMC) samplers and sequential importance resampling. For example Dundar et al. (Bayesian nonexhaustive learning for online discovery and modeling of emerging classes, ICML 2012) uses particle filters to classify non-stationary data, where emerging classes can be considered in a way similar to change points in the time series data. Also triggering kernels in self-exciting point processes such as Hawkes model can also be considered similar to change points. Some of the existing Hawkes model also uses SMC samplers. Given that MCMC samplers has been proved quite scalable and effective in a variety of similar problems it is not very convincing to think that they will not be scalable for BOCPD problem studied in the paper. Theorem 2 proves that the analytical form of the evidence lower bound can be obtained if three quantities have closed forms. Paper states that closed forms of these quantities can be obtained for many exponential models. The significance and relevance of this theorem is not very clear. For example, if one has to use Normal-Inverse-Gamma model the posterior predictive distribution can be obtained in a closed-form as student-t. In which case one is inclined to think that MCMC sampling based on this posterior can also achieve robustness. The paper offers three reasons as to why student-t is not ideal in this case but considering student-t based MCMC as one of the benchmark techniques and demonstrate its limitations on real-world datasets would offer a more compelling case to support the three arguments. Clarity The paper reads well. Most of the presented ideas are easy to follow. Originality Using \beta-divergence to achieve robustness in online change point detection can be considered an original idea. Significance The main contribution of the paper is somewhat incremental and limited. The key idea is to replace KL divergence with \beta-divergence for online CP detection to more effectively deal with outliers. Interesting additional work is done especially for initializing and refining \beta online and using SGD anchoring to achieve scalability but the significances of these different pieces of work are difficult to judge because their role on the overall performance of the proposed model are not discussed in greater detail. Although the comparison against KL divergence is well justified no comparison is offered against MCMC techniques. For example potential improvement over sequential Monte carlo samplers is not clear. The thoroughness of experimental discussion and analysis is below NIPS standards. Given these limitations the impact of the proposed work in the ML literature would likely be limited. Other comments: The experimental analysis compares performances of robust vs. standard Bayesian online change point detection on two data sets. Details are quite scarce especially with regard to the selection of hyperparameters in Bayesian priors, characteristics of data sets, additional benchmark techniques etc. For example one straightforward solution to eliminate outliers in univariate data would be to preprocess the signal by a 1-dimensional median filter. Would such a basic preprocessing technique be competitive against more complicated solutions? If yes then it could have been considered for benchmarking. If not, then arguments as to why it would not be effective would be useful to eliminate such naive ideas and get an intuitive understanding of the characteristics of the data sets being studied. In page 7 "Replacing expected ..." may read better if "expected" is replaced by "expectation". Note: I have reviewed author's feedback and glad to see that it addresses some of my concerns about MCMC not being used as a benchmark. Accordingly, I upgraded my score from a 5 to a 6.

Reviewer 3



This paper builds on recent work in the area of “General Bayesian Inference” (GBI). GBI has been proposed to deal with the “M-open world”, where the class of models we choose does not capture the actual sampling distribution. A remarkable fact is that in this setting standard Bayesian updating can be seen as a method which learns a model by minimising the predictive KL-divergence from the model from which the data were sampled. The problem with the KL-divergence in the case where the model is mis-specified the penalties are much worse in the tails, meaning that the posteriors that are derived can be unsatisfactory for e.g. decision making. Hence, the use of “robust” divergence measures are proposed, which focus instead on the areas of greatest mass. $\beta$-divergences have been proposed to do exactly this. The $\beta$-divergence can be derived from the more general $\alpha,\beta$-divergence by setting $\alpha$ to 1, or from the Bregman divergence family. There are two nice properties of the $\beta$-divergence: posterior inference does not require estimating the data generating density; and it results in robust inference, since observations with low predicted probability under the model are automatically down weighted. This is not for free, and is at the cost of statistical efficiency. The paper’s first contribution is to take GBI, and apply it to sequential Bayesian inference, where the prior at time t is the posterior from t - 1, to produce a general online inference method that is naturally robust to model misspecification. I feel that some of this could be made clearer. For example, M-open worlds are cited but not explained, and to the general reader it is probably not at all obvious why standard Bayesian inference corresponds to the KL between the fitted model and the data generating mechanism. I also think that the point that using $\beta$-divergence gives your robustness *irrespective of the model* is quite subtle. It is mentioned in the paper (e.g. at the end of 2.1) but I think this point could get missed. The technical contribution here is then to take a specific model: the Product Partition Model (aka Bayesian Changepoint Detection), and derive the inference procedure for the $\beta$-divergence. This essentially boils down to a regression model that is computed over the period since the last changepoint (the run length), with a corresponding model for the probability of the run continuing from one time step to the next. Note that the (extremely long and detailed) derivations are given in the appendix. The second contribution is to improve the efficiency of GBI in this model using stochastic variance reduced gradient descent (SVRG). This is actually general to GBI, rather than specific to BOCPD, although it’s also a fairly trivial modification from SVRG with KL. There is an extra nuance where the gradient estimates are anchored every few steps, which presumably comes from an insight (failure?) whilst running vanilla SVRG. It would be interesting to know how much is lost (if anything) versus performing full optimisation of the variational parameters (as we know, sometimes SGD provides an additional regularisation effect). The final technical contribution is of optimisation of $\beta$, including doing so in an online fashion. There are a lot of moving parts in this, even for the static version. It would be interesting in the examples to see how $\beta$ ends up getting set, and how it evolves through the time course of the experiments. Without this is somewhat difficult to estimate the utility of this method of selection, and whether the online updating is necessary or not. The experiments themselves are two classical examples in CPD. Unsurprisingly the proposed method is more robust to outliers, so mission accomplished in this regard. However, I feel there are multiple ways that these could be improved. Firstly, both of these examples are 1-dimensional; whilst the methods are clearly applicable to higher dimensions, and it would be really interesting to see how they behave. Also, as described in the appendix, there are quite a few hyper parameters and preprocessing steps, and little intuition as to how these are reached. I fear that a lot of experimentation was performed to get to this stage, and that application to a new domain would remain a difficult task. Specific comments: Figure 1a is a bit unintuitive. It’s clear that the KL is focusing more on the tails of the distribution. However, the influence has a strange profile for the $\beta$-divergences - going up as you move away before falling away again. It’s hard to say what the practitioner should take from this: one might think to use as small a $\beta$ as possible. However when $\beta$->0 it reduces to the KL-Divergence (again, not obvious at all from the plot). I realise that section 4 discusses optimisation of $\beta$, but I think there is some intuition missing here In the definition of the product partition model, there is a $\Sigma_0$ which has no definition (it is multiplied by $\sigma^2$, which has a conjugate prior) In the algorithm, what is B vs b? What is Geom? Is the $\mathcal{I}$ sampled with or without replacement? In the for loop, it looks like the indexer i isn’t used, and clashes with the $i \in \mathcal{I}$ [22] looks like it has been published under the title "Principles of Bayesian Inference Using General Divergence Criteria" references missing in appendices (latex build) Note: I didn’t check all of the appendix, but the parts that I did check were all correct. Note: I have reviewed author's feedback, and glad to see that it addresses the concerns raised both by myself also by R2. I had a misconception of the data being used in the 2nd example. Accordingly I have raised my score from a 7 to an 8.